# GENERATIVE ADVERSARIAL EQUILIBRIUM SOLVERS

**Denizalp Goktas**[*]
Brown University
denizalp_goktas@brown.edu

**David C. Parkes**[†] **Ian Gemp, Luke Marris,**
**Georgios Piliouras, Romuald Elie, Guy Lever, Andrea Tacchetti**
Google DeepMind
{parkesd, imgemp, marris, parkesd, imgemp, marris,
gpil, relie, guylever, atacchet}@google.com

## ABSTRACT

We introduce the use of generative adversarial learning to compute equilibria in general game-theoretic settings, specifically the *generalized Nash equilibrium* (GNE) in *pseudo-games*, and its specific instantiation as the *competitive equilibrium* (CE) in Arrow-Debreu competitive economies. Pseudo-games are a generalization of games in which players' actions affect not only the payoffs of other players but also their feasible action spaces. Although the computation of GNE and CE is intractable in the worst-case, i.e., PPAD-hard, in practice, many applications only require solutions with high accuracy in expectation over a distribution of problem instances. We introduce *Generative Adversarial Equilibrium Solvers* (GAES): a family of generative adversarial neural networks that can learn GNE and CE from only a sample of problem instances. We provide computational and sample complexity bounds for Lipschitz-smooth function approximators in a large class of concave pseudo-games, and apply the framework to finding Nash equilibria in normal-form games, CE in Arrow-Debreu competitive economies, and GNE in an environmental economic model of the Kyoto mechanism.

## 1 INTRODUCTION

Economic models and equilibrium concepts are critical tools to solve practical problems, including capacity allocation in wireless and network communication (Han et al., 2011; Pang et al., 2008), energy resource allocation (Hobbs & Pang, 2007; Jing-Yuan & Smeers, 1999), and cloud computing (Gutman & Nisan, 2012; Lai et al., 2005; Zahedi et al., 2018; Ardagna et al., 2017). Many of these economic models are instances of what are known as *pseudo-games*, in which the actions taken by each player affect not only the other players' payoffs, as in games, but also the other players' strategy sets.[1] The formalism of pseudo-games was introduced by Arrow & Debreu (1954), who used it in studying their foundational microeconomic equilibrium model, the competitive economy.

The standard solution concept for pseudo-games is the *generalized Nash equilibrium (GNE)* (Arrow & Debreu, 1954; Facchinei & Kanzow, 2010a), which is an action profile from which no player can improve their payoff by unilaterally deviating to another action in the space of admissible actions determined by the actions of other players. Important economic models can often be formulated as a pseudo-game, with their set of solutions equal to the set of GNE of the pseudo-game: for instance, the set of *competitive equilibria (CE)* (Walras, 1896; Arrow & Debreu, 1954) of an Arrow-Debreu competitive economy corresponds to the set of GNE of an associated pseudo-game.

A large literature has been devoted to the computation of GNE in certain classes of pseudo-games but unfortunately many algorithms that are guaranteed to converge in theory have in practice been observed to converge slowly in ill-conditioned or large problems or fail numerically (Facchinei &

---

[*]Research conducted while the author was an intern at Google DeepMind.

[†]Also, School of Engineering and Applied Sciences, Harvard University.

[1]In many games, such as chess, the action taken by one player affects the actions available to the others, but these games are sequential, while in pseudo-games actions are chosen simultaneously. Additionally, even if one constructs a game with payoffs that penalize the players for actions that are not allowed, the NE of the ensuing game will in general not correspond to the GNE of the original pseudo-game and can often be trivial. We refer the reader to Appendix A for a mathematical example.

Kanzow, 2010b; Jordan et al., 2022; Goktas & Greenwald, 2022). Additionally, all known algorithms have hyperparameters that have to be optimized individually for every pseudo-game instance (Facchinei & Kanzow, 2010a), deteriorating the performance of these algorithms when used to solve multiple pseudo-games. These issues point to a need to develop methods to compute GNE, for a distribution of pseudo-games, reliably and quickly.

We reformulate the problem of computing GNE in pseudo-games (and CE in Arrow-Debreu competitive economies) as a learning problem for a generative adversarial network (GAN) called the *Generative Adversarial Equilibrium Solver (GAES)*, consisting of a generator and a discriminator network. The generator takes as input a parametric representation of a pseudo-game, and predicts a solution that consists of a tuple of actions, one per player. The discriminator takes as input both the pseudo-game and the output of the generator, and outputs a best-response for each player, seeking to find a useful unilateral deviation for all players; this also gives the sum of regrets, with which to evaluate the generator (see Figure 1). GAES predicts GNE and CE in batches and in order to minimize the expected exploitability, across a distribution of pseudo-games. GAES amortizes computational cost up-front in training, and allows for near constant evaluation time for inference. Our approach is inspired by previous methods that cast the computation of an equilibrium in normal-form games as an unsupervised learning problem (Duan et al., 2021a; Marris et al., 2022). These methods train a network to predict a strategy profile that minimizes the *exploitability* (i.e., the sum of the players' payoff-maximizing unilateral deviations w.r.t. a given strategy profile) over a distribution of games. These methods become inefficient in pseudo-games, since in contrast to regular games, the exploitability in pseudo-games (1) requires solving a non-linear optimization problem, (2) is not Lipschitz-continuous, in turn making it hard to learn from samples, and (3) has unbounded gradients, which might lead to exploding gradients in neighborhoods of GNE. Our GAN formulation circumvents all three of these issues.

Although the computation of GNE is intractable in the worst-case (Chen & Deng, 2006; Daskalakis et al., 2009; Chen & Teng, 2009; Vazirani & Yannakakis, 2011; Garg et al., 2017), in practice, applications may only require a solver that gives solutions with high accuracy in expectation over a realistic distribution of problem instances. In particular, a decision maker may need to compute a GNE for a sequence of pseudo-games from some family or *en masse* over a set of pseudo-games sampled from some distribution of interest. An example of such an application is the problem of resource allocation on cloud computing platforms (Hindman et al., 2011; Isard et al., 2009; Burns et al., 2016; Vavilapalli et al., 2013) where a significant number of methods make use of repeated computation of competitive equilibrium (Gutman & Nisan, 2012; Lai et al., 2005; Budish, 2011; Zahedi et al., 2018; Varian, 1973) and generalized Nash equilibrium (Ardagna et al., 2017; 2011b;a; Anselmi et al., 2014). In such settings, as consumers request resources from the platform, the platforms have to find a new equilibrium while handling all numerical failures within a given time frame. Another example is policy makers who often want to understand the equilibria induced by a policy for different distributions of agent preferences in a pseudo-game allowing them to study the impact of a policy for a distribution on different possible kinds of participants. For example, in studying the impact of a protocol such as the *Kyoto joint implementation mechanism* (see Section 5), one might be interested in understanding how the emission levels of countries would change based on their productivity levels (Jones et al., 2000). Other applications include computing competitive equilibria in stochastic market environments. For example, recently proposed algorithms work through a series of equilibrium problems, each of which has to be solved quickly (Liu et al., 2022).

**Contributions.** Earlier approaches Duan et al. (2021a); Marris et al. (2022) do not extend even to continuous (non pseudo-)games, since evaluating the expected exploitability and its gradient over a distribution of pseudo-games requires solving as many convex programs as examples in the data set. Additionally, in pseudo-games, the exploitability is not Lipschitz-continuous, and thus its gradient is unbounded (Appendix D), hindering the use of standard tools to prove sample complexity and convergence bounds, and making training hard due to exploding gradients. By delegating the task of computing a best-response to a discriminator, our method circumvents the issue of solving a convex

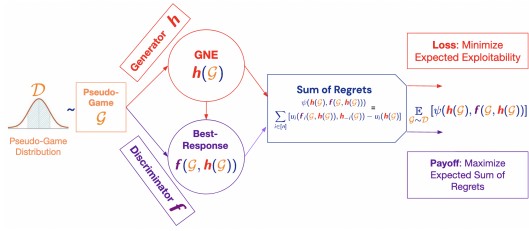

Figure 1: Summary of the Architecture of GAES.

program, yielding a training problem given by a min-max optimization problem whose objective is Lipschitz-continuous, for which gradients can be guaranteed to be bounded under standard assumptions on the discriminator and the payoffs of players.

Our approach also extends the class of (non pseudo-)games that can be solved through deep learning methods from normal-form games to simultaneous-move continuous action games, since the non-linear program involved in the computation of exploitability in previous methods makes them inefficient in application to continuous-action games. We prove polynomial-time convergence of our training algorithm for arbitrary Lipschitz-smooth function approximators with the joint action space dimensions larger than parameter space dimensions (Theorem 4.1, Section 4). and provide generalization bounds for arbitrary function approximators (Theorem 4.2, Section 4). Finally, we provide empirical evidence that GAES outperforms state of the art baselines in Arrow-Debreu competitive economies, and show that GAES can replicate existing qualitative analyses for pseudo-games, suggesting that GAES makes predictions that not only have low expected exploitability, but also are qualitatively correct, i.e., close to the true GNE in action space (Section 5).

**Additional related work.** We refer the reader to Appendix B for a survey of methods to compute GNEs, and to Appendix C for a survey of applications of GNE. Our contributions generally relate to a line of work on *differentiable economics*, which seeks to use methods of neural computation for problems of economic design and equilibrium computation. In regard to finding optimal economic designs, deep learning has been used for problems of auction design (Dütting et al., 2019; Curry et al., 2022c; Tacchetti et al., 2019; Curry et al., 2022a; Gemp et al., 2022; Rahme et al., 2020; Duan et al., 2022) and matching (Ravindranath et al., 2021). In regard to solving for equilibria, some recent works have tried to solve for Nash equilibria in auctions (Heidekrüger et al., 2019; Bichler et al., 2021), and dynamic stochastic general equilibrium models (Curry et al., 2022b; Chen et al., 2021; Hill et al., 2021).

## 2 PRELIMINARIES

**Notation.** All notation for variable types, e.g., vectors, are clear from context, if any confusions arise see Appendix E. We denote the set of integers $\{0, \ldots, n-1\}$ by $[n]$, the set of natural numbers by $\mathbb{N}$, the set of real numbers by $\mathbb{R}$, and the positive and strictly positive elements of a set by a subscript $+$ and $++$, e.g., $\mathbb{R}_+$ and $\mathbb{R}_{++}$. We denote by $\Delta_n = \{\boldsymbol{x} \in \mathbb{R}_+^n \mid \sum_{i=1}^n x_i = 1\}$, and by $\Delta(A)$, the set of probability measures on the set $A$.

**Pseudo-Games.** A *(concave) pseudo-game* (Arrow & Debreu, 1954) $\mathcal{G} \doteq (n, \mathcal{A}, \mathcal{X}^{\mathcal{G}}, \boldsymbol{h}^{\mathcal{G}}, \boldsymbol{u}^{\mathcal{G}})$, denoted $(n, \mathcal{A}, \mathcal{X}, \boldsymbol{h}, \boldsymbol{u})$ when clear from context, comprises $n \in \mathbb{N}_+$ players, where player $i \in [n]$ chooses an action $\boldsymbol{a}_i$ from a non-empty, compact, and convex *action space* $\mathcal{A}_i \subset \mathbb{R}^m$. We denote the players' joint action space by $\mathcal{A} = \bigtimes_{i \in [n]} \mathcal{A}_i \subset \mathbb{R}^{nm}$. Each player $i \in [n]$ aims to maximize their continuous *payoff*, $u_i : \mathcal{A} \to \mathbb{R}$, which is concave in $\boldsymbol{a}_i$, by choosing a feasible action from a set of actions, $\mathcal{X}_i(\boldsymbol{a}_{-i}) \subseteq \mathcal{A}_i$, this depending on the actions $\boldsymbol{a}_{-i} \in \mathcal{A}_{-i} \subset \mathbb{R}^{(n-1)m}$ of the other players. Here, $\mathcal{X}_i : \mathcal{A}_{-i} \rightrightarrows \mathcal{A}_i$ is a non-empty, continuous, compact- and convex-valued *(feasible) action correspondence*. It is this dependence on each others' actions that makes this a pseudo-game, and not just a game. In applications, we will represent $\mathcal{X}_i$ as $\mathcal{X}_i(\boldsymbol{a}_{-i}) = \{\boldsymbol{a}_i \in \mathcal{A}_i \mid h_{ip}(\boldsymbol{a}_i, \boldsymbol{a}_{-i}) \geq \boldsymbol{0}, \text{ for all } p \in [d]\}$, where for all $i \in [n]$, and $p \in [d]$, $h_{ip}$ is a continuous and quasi-concave function in $\boldsymbol{a}_i$, which defines the constraints. We denote the *product (feasible) action correspondence* by $\mathcal{X}(\boldsymbol{a}) = \bigtimes_{i \in [n]} \mathcal{X}_i(\boldsymbol{a}_{-i})$, which we note is guaranteed to be continuous, and non-empty-, compact-valued. We denote $\mathcal{X}$ the *set of jointly feasible strategies*, i.e., $\mathcal{X} = \{\boldsymbol{a} \in \mathcal{A} \mid h_{ip}(\boldsymbol{a}) \geq \boldsymbol{0}, \forall i \in [n], p \in [d]\}$. We denote the class of all pseudo-games by $\Gamma$.[2]

Given a pseudo-game $\mathcal{G}$, a *generalized Nash equilibrium (GNE)* is an action profile $\boldsymbol{a}^* \in \mathcal{X}$, s.t. for all $i \in [n]$ and $\boldsymbol{a}_i \in \mathcal{X}_i(\boldsymbol{a}^*_{-i})$, $u_i(\boldsymbol{a}^*) \geq u_i(\boldsymbol{a}_i, \boldsymbol{a}^*_{-i})$. An *equilibrium mapping*, $\boldsymbol{h} : \Gamma \to \mathcal{X}$ is a mapping that takes as input a pseudo-game $\mathcal{G} \in \Gamma$ and outputs a GNE, $\boldsymbol{h}(\mathcal{G})$, for that game. Given a pseudo-game $\mathcal{G}$, we define the *regret* for player $i \in [n]$ for action $\boldsymbol{a}_i$ as compared to another action $\boldsymbol{b}_i$, given the action profile $\boldsymbol{a}_{-i}$ of other players, as $\text{Regret}_i^{\mathcal{G}}(\boldsymbol{a}_i, \boldsymbol{b}_i; \boldsymbol{a}_{-i}) \doteq u_i^{\mathcal{G}}(\boldsymbol{b}_i, \boldsymbol{a}_{-i}) - u_i^{\mathcal{G}}(\boldsymbol{a}_i, \boldsymbol{a}_{-i})$. Additionally, the *cumulative regret*, between two action profiles $\boldsymbol{a} \in \mathcal{A}$ and $\boldsymbol{b} \in \mathcal{A}$ is given by

---

[2]A *game* (Nash, 1950) is a pseudo-game where, for all players $i \in [n]$, $\mathcal{X}_i$ is a constant correspondence with value $\mathcal{A}_i$. A *discrete action game* is a game where $\mathcal{A}_i = \Delta_m$ and $u_i$ is multilinear for all $i \in [n]$.

$\psi(\boldsymbol{a}, \boldsymbol{b}) \doteq \sum_{i \in [n]} \mathrm{Regret}_i^{\mathcal{G}}(\boldsymbol{a}_i, \boldsymbol{b}_i; \boldsymbol{a}_{-i})$. Further, the *exploitability* (Nikaido & Isoda, 1955), of an action profile $\boldsymbol{a}$ is defined as $\varphi^{\mathcal{G}}(\boldsymbol{a}) \doteq \sum_{i \in [n]} \max_{\boldsymbol{b}_i \in \mathcal{X}_i(\boldsymbol{a}_{-i})} \mathrm{Regret}_i^{\mathcal{G}}(\boldsymbol{a}_i, \boldsymbol{b}_i; \boldsymbol{a}_{-i})$. Note that an action profile $\boldsymbol{a}^*$ is a GNE iff $\varphi^{\mathcal{G}}(\boldsymbol{a}^*) = 0$.

**Mathematical preliminaries.** For any function $f : \mathcal{X} \to \mathcal{Y}$, we denote its Lipschitz-continuity constant by $\ell_f$. For two arbitrary sets $\mathcal{H}, \mathcal{H}' \subset \mathcal{F}$, the set $\mathcal{H}'$ *r-covers* $\mathcal{H}$ (w.r.t. some norm $\|\cdot\|$) if for any $\boldsymbol{h} \in \mathcal{H}$ there exists $\boldsymbol{h}' \in \mathcal{H}'$ such that $\|\boldsymbol{h} - \boldsymbol{h}'\| \leq r$. The *r-covering number*, $\rho(\boldsymbol{h}, r)$, of a set $\mathcal{H}$ is the cardinality of the smallest set $\mathcal{H}' \subset \mathcal{F}$ that $r$-covers $\mathcal{H}$. A set $\mathcal{H}$ is said to have a *bounded covering number*, if for all $r \in \mathbb{R}_+$, we have that the logarithm of its covering number is polynomially bounded in $1/r$, that is $\log(\rho(\boldsymbol{h}, r)) \leq \mathrm{poly}(1/r)$. Additional background can be found in Appendix E.

## 3 GENERATIVE ADVERSARIAL LEARNING OF EQUILIBRIUM MAPPINGS

In this section, we revisit previous formulations of the problem of learning an equilibrium mapping, discuss the computational difficulties associated with these formulations when used to learn GNE, and introduce our generative adversarial learning formulation.

As creating a sufficiently diverse sample of (pseudo-game, GNE) pairs, while performing adequate equilibrium selection, is intractable both theoretically and computationally, we forgo of supervised learning methods, and formulate the equilibrium mapping learning problem as an *unsupervised* learning problem, following the approach adopted by Marris et al. (2022); Duan et al. (2021b) for finding Nash equilibria. Given a hypothesis class $\mathcal{H} \subseteq \mathcal{X}^\Gamma$, and a distribution over pseudo-games $\mathcal{D} \in \Delta(\Gamma)$, *the unsupervised learning problem* for an equilibrium mapping consists of finding a hypothesis $\boldsymbol{h}^* \in \arg\min_{\boldsymbol{h} \in \mathcal{H}} \mathbb{E}_{\mathcal{G} \sim \mathcal{D}} [\ell(\mathcal{G}, \boldsymbol{h}(\mathcal{G}))]$ where $\ell : \Gamma \times \mathcal{A} \to \mathbb{R}$ is a loss function that outputs the distance of $\boldsymbol{a} \in \mathcal{A}$ from a GNE, such that for any pseudo-game $\mathcal{G} \in \Gamma$, $\ell(\mathcal{G}, \boldsymbol{a}^*) = 0$ iff $\boldsymbol{a}^*$ is a GNE of $\mathcal{G}$. In particular, Marris et al. (2022); Duan et al. (2021b) suggest to use exploitability as the loss function. However, a number of issues arise when trying to minimize the expected exploitability over a distribution of pseudo-games:

(1) Computing the gradient of the exploitability, when it exists, for even only one pseudo-game requires solving a concave maximization problem (this reasoning also applies to continuous games).

(2) The exploitability in pseudo-games, is in general not Lipschitz-continuous (unlike in regular games), even when payoffs are Lipschitz-continuous, since the inputs of the exploitability parameterize the constraints in the optimization problem defining each player's maximal regret computation. This makes it unclear how to efficiently approximate $\mathbb{E}_{\mathcal{G} \sim \mathcal{D}} [\varphi^{\mathcal{G}}(\boldsymbol{a})]$ from samples, without knowledge of the distribution.

(3) The exploitability in pseudo-games is absolutely continuous and hence differentiable almost everywhere Afriat (1971), but in contrast to games, the gradients cannot be bounded. This in turn precludes the convergence of first-order methods.[3]

To address the aforementioned issues, we propose a generative adversarial learning formulation of the associated unsupervised learning problem for equilibrium mappings. The formulation relies on the following observation, whose proof is deferred to Appendix F: the exploitability can be computed *ex post* after computing the expected cumulative regret by optimizing over the space of best-response functions from pseudo-games to actions, rather than the space of actions individually for every pseudo-game.

**Observation 1.** *For any* $\mathcal{D} \in \Delta(\Gamma)$, *we have* $\min_{\boldsymbol{h} \in \mathcal{X}^\Gamma} \mathbb{E}_{\mathcal{G} \sim \mathcal{D}} [\varphi^{\mathcal{G}}(\boldsymbol{h}(\mathcal{G}))] = \min_{\boldsymbol{h} \in \mathcal{X}^\Gamma} \max_{\boldsymbol{f} \in \mathcal{A}^\Gamma : \forall \mathcal{G} \in \Gamma, \boldsymbol{f}(\mathcal{G}) \in \mathcal{X}^{\mathcal{G}}(\boldsymbol{h}(\mathcal{G}))} \mathbb{E}_{\mathcal{G} \sim \mathcal{D}} [\psi^{\mathcal{G}}(\boldsymbol{h}(\mathcal{G}), \boldsymbol{f}(\mathcal{G}))]$

By Arrow-Debreu's lemma on abstract economies Arrow & Debreu (1954), $\boldsymbol{h}^*$ is guaranteed to exist and is an equilibrium mapping iff $\boldsymbol{h}^* \in \arg\min_{\boldsymbol{h} \in \mathcal{X}^\Gamma} \max_{\boldsymbol{f} \in \mathcal{A}^{\Gamma \times \mathcal{X}} : \forall \mathcal{G} \in \Gamma, \boldsymbol{f}(\mathcal{G}, \boldsymbol{h}(\mathcal{G})) \in \mathcal{X}^{\mathcal{G}}(\boldsymbol{h}(\mathcal{G}))} \mathbb{E}_{\mathcal{G} \sim \mathcal{D}} [\psi^{\mathcal{G}}(\boldsymbol{h}(\mathcal{G}), \boldsymbol{f}(\mathcal{G}))]$.

---

[3]We refer the reader to Appendix D for an example in which exploitability is not Lipschitz-continuous and has unbounded gradients.

This problem formulation allows us to overcome issues (1) and (2). For (1), rather than solve a concave program to compute the exploitability for each pseudo-game and action profile, we can learn a function that maps action profiles to their associated best-response profiles (see for example Lanctot et al. (2017) for training best-response oracles). For (2), the objective function in the Equation on the right hand side of Observation 1, i.e., the cumulative regret, is Lipschitz-continuous in the action profiles when payoff functions are, which opens the doors to use standard proof techniques to learn the objective from a polynomial sample of pseudo-games.

Still, the gradient of $\max_{\boldsymbol{f}} \mathbb{E}_{\mathcal{G} \sim \mathcal{D}} \left[ \psi^{\mathcal{G}}(\boldsymbol{h}(\mathcal{G}), \boldsymbol{f}(\mathcal{G}; \boldsymbol{h}(\mathcal{G}))) \right]$ with respect to $\boldsymbol{h}$ is in general unbounded even when it exists, due to the constraint $\forall \mathcal{G} \in \Gamma, \boldsymbol{f}(\mathcal{G}) \in \mathcal{X}(\boldsymbol{h}(\mathcal{G}))$. However, since any solution $\boldsymbol{f}^*(\mathcal{G}, \boldsymbol{h}(\mathcal{G}))$ to the inner optimization problem $\max_{\boldsymbol{f} \in \mathcal{A}^{\Gamma}: \forall \mathcal{G} \in \Gamma, \boldsymbol{f}(\mathcal{G}) \in \mathcal{X}(\boldsymbol{h}(\mathcal{G}))} \mathbb{E} \left[ \psi^{\mathcal{G}}(\boldsymbol{h}(\mathcal{G}), \boldsymbol{f}(\mathcal{G})) \right]$ is implicitly parameterized by the choice of equilibrium mapping $\boldsymbol{h}$, we can represent this dependence explicitly in the optimization problem, and restrict our selection of $\boldsymbol{f}$ to a continuously differentiable hypothesis class $\mathcal{F} \subset \mathcal{A}^{\Gamma \times \mathcal{X}}$, and overcome issue (3).

With these observations in mind, given hypothesis classes $\mathcal{H} \subset \mathcal{X}^{\Gamma}$, and $\mathcal{F} \subset \mathcal{A}^{\Gamma \times \mathcal{X}}$, the *generative adversarial learning problem* is to find a tuple $(\boldsymbol{h}^*, \boldsymbol{f}^*) \in \mathcal{H} \times \mathcal{F}$ that consists of a *generator* and *discriminator* to solve the following optimization problem:

$$\min_{\boldsymbol{h} \in \mathcal{H}} \max_{\boldsymbol{f} \in \mathcal{F}: \forall \mathcal{G} \in \Gamma, \boldsymbol{f}(\mathcal{G}; \boldsymbol{h}(\mathcal{G})) \in \mathcal{X}^{\mathcal{G}}(\boldsymbol{h}(\mathcal{G}))} \mathbb{E}_{\mathcal{G} \sim \mathcal{D}} \left[ \psi^{\mathcal{G}}(\boldsymbol{h}(\mathcal{G}), \boldsymbol{f}(\mathcal{G}; \boldsymbol{h}(\mathcal{G}))) \right]. \tag{1}$$

This problem can be interpreted as a zero-sum game between the generator and the discriminator. The generator takes as input a parametric representation of a pseudo-game, and predicts a solution that consists of an *action profile*, i.e., a tuple of actions, one per agent. The discriminator takes the game and the output of the generator as input, and outputs a best-response for each agent (Figure 1). The optimal mappings $(\boldsymbol{h}^*, \boldsymbol{f}^*)$ for Equation (1) are then called the *Generative Adversarial Equilibrium Solver (GAES)*.

## 4 CONVERGENCE AND SAMPLE COMPLEXITY

For training, we propose a stochastic variant of the multistep gradient descent ascent algorithm (Nouiehed et al., 2019; Sanjabi et al., 2018), which we call *stochastic exploitability descent*. Our algorithm computes the optimal generator and discriminator by estimating the gradient of the expected cumulative regret and exploitability on a training set of pseudo-games.

---

**Algorithm 1** Stochastic Exploitability Descent

**Inputs:** $\mathcal{B}, \eta_{\boldsymbol{h}}, \eta_{\boldsymbol{f}}, T_{\boldsymbol{h}}, T_{\boldsymbol{f}}, \boldsymbol{w}^{\boldsymbol{h},(0)}, \boldsymbol{w}^{\boldsymbol{f},(0)}$
**Outputs:** $\left( \boldsymbol{w}^{\boldsymbol{h},(t)}, \boldsymbol{w}^{\boldsymbol{f},(t)} \right)_{t=0}^{T_{\boldsymbol{h}}}$
1: **for** $t = 0, \ldots, T_{\boldsymbol{h}} - 1$ **do**
2:      Receive batch $\mathcal{B}^{(t)} \subset \mathcal{S}$.
3:      $\boldsymbol{w}^{\boldsymbol{h},(t+1)} = \boldsymbol{w}^{\boldsymbol{h},(t)} - \eta_{\boldsymbol{h}}^{(t)} \left( 1/|\mathcal{B}_{\boldsymbol{h}}^{(t)}| \sum_{\mathcal{G} \in \mathcal{B}_{\boldsymbol{h}}^{(t)}} \left[ \nabla_{\boldsymbol{w}^{\boldsymbol{h}}} \widehat{\psi}(\boldsymbol{w}^{\boldsymbol{h},(t)}, \boldsymbol{w}^{\boldsymbol{f},(t)}) \right] \right)$
4:      **for** $s = 0, \ldots, T_{\boldsymbol{f}} - 1$ **do**
5:          Receive batch $\mathcal{B}^{(s)} \subset \mathcal{S}$.
6:          $\boldsymbol{w}^{\boldsymbol{f}} = \boldsymbol{w}^{\boldsymbol{f}} + \eta_{\boldsymbol{f}}^{(s)} \left( 1/|\mathcal{B}_{\boldsymbol{f}}^{(s)}| \sum_{\mathcal{G} \in \mathcal{B}_{\boldsymbol{f}}^{(s)}} \nabla_{\boldsymbol{w}^{\boldsymbol{f}}} \widehat{\psi}(\boldsymbol{w}^{\boldsymbol{h},(t)}, \boldsymbol{w}^{\boldsymbol{f}}) \right)$
7:      **end for**
8:      $\boldsymbol{w}^{\boldsymbol{f},(t+1)} = \boldsymbol{w}^{\boldsymbol{f}}$
9: **end for**
10: Return $\left( \boldsymbol{w}^{\boldsymbol{h},(t)}, \boldsymbol{w}^{\boldsymbol{f},(t)} \right)_{t=0}^{T_{\boldsymbol{h}}}$

---

**Training Algorithm.** For purposes of applicability, going forward, we will assume that we have access to the distribution of pseudo-games $\mathcal{D}$ only indirectly through a *training set* $\mathcal{S} \sim \mathcal{D}$ of $k \in \mathbb{N}_+$ sampled pseudo-games. Additionally, we will assume that the generator $\boldsymbol{h} \in \mathcal{H}$ and discriminator

$\boldsymbol{f} \in \mathcal{F}$ are parameterized by parameter vectors, $\boldsymbol{w^h}, \boldsymbol{w^f} \in \mathbb{R}^\omega$ such that for all pseudo-games $\mathcal{G} \in \Gamma$, and weight vectors $\boldsymbol{w} \in \mathbb{R}^\omega$, $\boldsymbol{h}(\mathcal{G}; \boldsymbol{w^h}) \in \mathcal{X}$ and $\boldsymbol{f}(\mathcal{G}; \boldsymbol{h}(\mathcal{G}; \boldsymbol{w^h}), \boldsymbol{w^f}) \in \mathcal{A}$.

For notational simplicity, we define the *expected cumulative regret* and the *empirical cumulative regret*, respectively, as $\overline{\psi}(\boldsymbol{w^h}, \boldsymbol{w^f}) \doteq \mathbb{E}\left[\psi^{\mathcal{G}}(\boldsymbol{h}(\mathcal{G}; \boldsymbol{w^h}), \boldsymbol{f}(\mathcal{G}, \boldsymbol{h}(\mathcal{G}; \boldsymbol{w^h}); \boldsymbol{w^f}))\right]$, and $\widehat{\psi}(\boldsymbol{w^h}, \boldsymbol{w^f}) \doteq \mathbb{E}\left[\psi^{\mathcal{G}}(\boldsymbol{h}(\mathcal{G}; \boldsymbol{w^h}), \boldsymbol{f}(\mathcal{G}; \boldsymbol{h}(\mathcal{G}; \boldsymbol{w^h}), \boldsymbol{w^f}))\right]$ where the expectation is over the distribution of pseudo-games $\mathcal{G} \sim \mathcal{D}$ and the uniform distribution over the training set, $\mathcal{G} \sim \mathrm{unif}(\mathcal{S})$ respectively. Similarly, we define the *expected exploitability* and the *empirical exploitability* respectively as:

$$\overline{\varphi}(\boldsymbol{w^h}) \doteq \max_{\substack{\boldsymbol{w^f} \in \mathbb{R}^\omega : \forall \mathcal{G} \in \Gamma, \\ \boldsymbol{f}(\mathcal{G}; \boldsymbol{w^f}) \in \mathcal{X}(\boldsymbol{h}(\mathcal{G}, \boldsymbol{w^h}))}} \overline{\psi}(\boldsymbol{w^h}, \boldsymbol{w^f}) \quad \Big| \quad \widehat{\varphi}(\boldsymbol{w^h}) \doteq \max_{\substack{\boldsymbol{w^f} \in \mathbb{R}^\omega : \forall \mathcal{G} \in \Gamma, \\ \boldsymbol{f}(\mathcal{G}; \boldsymbol{w^f}) \in \mathcal{X}(\boldsymbol{h}(\mathcal{G}, \boldsymbol{w^h}))}} \widehat{\psi}(\boldsymbol{w^h}, \boldsymbol{w^f}),$$

Putting this together, our training problem becomes:

$$\min_{\boldsymbol{w^h} \in \mathbb{R}^\omega} \max_{\boldsymbol{w^f} \in \mathbb{R}^\omega : \forall \mathcal{G} \in \Gamma, \boldsymbol{f}(\mathcal{G}; \boldsymbol{h}(\mathcal{G}; \boldsymbol{w^h}), \boldsymbol{w^f}) \in \mathcal{X}(\boldsymbol{h}(\mathcal{G}; \boldsymbol{w^h}))} \widehat{\psi}(\boldsymbol{w^h}, \boldsymbol{w^f}). \tag{2}$$

We propose Algorithm 1 to solve this optimization problem. This is a nested stochastic gradient descent-ascent algorithm, which for each generator descent step runs multiple stochastic gradient ascent steps on the weights of the discriminator to approximate the empirical cumulative regret by processing the pseudo-games in the training set in *batches*, i.e., as mutually exclusive subsets of the training set. After the stochastic gradient ascent steps are done, the algorithm then takes a step of stochastic gradient descent on the empirical exploitability w.r.t. the weights of the generator using the discriminator's weights computed by the stochastic gradient ascent steps to compute the gradient. Note that when the hypothesis class of the discriminator is assumed to contain only differentiable functions, the gradient of the generator with respect to its weights exist, and they are given by the implicit function theorem.

**Convergence and generalization bounds.** We give assumptions under which our algorithm converges to a stationary point of the empirical exploitability in polynomial-time.

**Assumption 1.** *For any player $i \in [n]$ and $\mathcal{G} \in \mathrm{supp}(\mathcal{D})$: 1. (Lipschitz-smoothness) their payoff $u_i^{\mathcal{G}}$ is $\ell_{\nabla_{\boldsymbol{u}}}$-Lipschitz smooth, 2. (Strong concavity) their payoff $u_i^{\mathcal{G}}$ is $\mu_{u_i}$-strongly-concave in $\boldsymbol{a}_i$, and 3. (Lipschitz-smooth hypothesis classes) For all $\boldsymbol{h} \in \mathcal{H} \subset \mathcal{X}^{\Gamma \times \mathbb{R}^\omega}$, $\boldsymbol{f} \in \mathcal{F} \subset \mathcal{A}^{\Gamma \times \mathcal{X} \times \mathbb{R}^\omega}$, $\boldsymbol{h}(\mathcal{G}; \cdot)$, $\boldsymbol{f}(\mathcal{G}; \cdot)$ are injective, and Lipschitz-smooth functions.*

We note that strong concavity can further be weakened to concavity, and our results can directly extend to any concave pseudo-game by replacing the cumulative regret $\psi$ with its regularized counterpart $\psi_\alpha^{\mathcal{G}}(\boldsymbol{a}, \boldsymbol{b}) \doteq \psi_\alpha(\boldsymbol{a}, \boldsymbol{b}) - \alpha/2 \|\boldsymbol{a} - \boldsymbol{b}\|_2^2$, which is strongly-concave in $\boldsymbol{b}$ without modifying the solutions of the optimization (Von Heusinger & Kanzow, 2009). However, as in our experiments non-regularized cumulative regret performed better, for consistency, we present our theoretical results under the strong concavity assumption. That said, strong concavity in each player's action is a much weaker assumption than strong monotonicity of the pseudo-game which is commonly used in the GNE literature (Jordan et al., 2022). For omitted definitions, and proofs/results we refer the reader to Appendix E and Appendix F respectively.

Theorem 4.1 tells us that our algorithm converges to a stationary point of the empirical exploitability at a $\tilde{O}(1/\sqrt{T_h})$ rate, up to an error term that depends linearly on the distance, $\varepsilon$, of the discriminator computed by the algorithm w.r.t. to the optimal discriminator. A smaller $\varepsilon$ results in higher accuracy, at the expense of a longer run time. [4]

**Theorem 4.1** (Convergence to Stationary Point). *Suppose that Assumption 1 holds. Let $\varepsilon > 0$. If Algorithm 1 is run with inputs satisfying $\eta_{\boldsymbol{h}}^{(t)} \in \Theta(1/\sqrt{t})$, $\eta_{\boldsymbol{f}}^{(s)} \in \Theta(1/s)$, $T_{\boldsymbol{h}} \in \mathbb{N}_{++}$, and $T_{\boldsymbol{f}} \in O(1/\varepsilon)$, for all $t \in [T_{\boldsymbol{h}}], s \in [T_{\boldsymbol{f}}]$. Then, the outputs $(\boldsymbol{w^{h,(t)}}, \boldsymbol{w^{f,(t)}})_{t=0}^{T_{\boldsymbol{h}}}$ satisfy*

$$\mathbb{E}\left[\min_{t=0,\ldots,T_{\boldsymbol{h}}-1} \left\|\nabla_{\boldsymbol{a}} \widehat{\varphi}(\boldsymbol{w^{h,(t)}})\right\|_2^2\right] \in O\left(\frac{\log(T_{\boldsymbol{h}})}{\sqrt{T_{\boldsymbol{h}}}} + \varepsilon\right).$$

---

[4]Our min-max problem is non-convex-PL, for which single loop stochastic gradient descent ascent algorithms are not guaranteed to converge to stationary points of the exploitability (Lin et al., 2020; Daskalakis et al., 2009). Additionally, our algorithm's computational complexity $\tilde{O}(1/\epsilon^3)$ is orders of magnitude faster than the best known complexity of $\tilde{O}(1/\epsilon^6)$ for such problems (Lin et al., 2020).

We also give a sample complexity result showing how cumulative regret can be approximated with a sample of pseudo-games that is polynomial in the parameters of the game distribution, $1/\varepsilon$ and $1/\delta$. The novelty of the result comes from the context: we mentioned earlier that expected exploitability need not be Lipschitz-continuous, making it hard to use any standard and simple machinery to prove a sample complexity bound. However, by reframing this problem as one of learning the expected cumulative regret, which is Lipschitz-continuous, we can obtain the result.

**Theorem 4.2** (Sample Complexity of Expected Cumulative Regret). *Suppose that part 1. of Assumption 1 holds. Let $\varepsilon, \delta \in (0, 1)$, $r \in O\left(\varepsilon/\ell_\psi\right)$ and consider hypothesis classes $\mathcal{H}, \mathcal{F}$ given by the minimum $r$-covering sets of $\mathcal{X}^\Gamma, \mathcal{A}^{\mathcal{X} \times \Gamma}$ respectively. For any $\boldsymbol{h} \in \mathcal{X}^\Gamma$ and $\boldsymbol{f} \in \mathcal{A}^{\mathcal{X} \times \Gamma}$, take the closest hypotheses $\boldsymbol{h}^r \in \arg\min_{\boldsymbol{h}' \in \mathcal{H}} \|\boldsymbol{h} - \boldsymbol{h}'\|$ and $\boldsymbol{f}^r \in \arg\min_{\boldsymbol{f}' \in \mathcal{F}} \|\boldsymbol{f} - \boldsymbol{f}'\|$ to respectively. Then, for any pseudo-game distribution $\mathcal{D} \in \Delta(\Gamma)$ with compact support, with probability at least $1 - \delta$ over draws of the training set $\mathcal{S} \sim \mathcal{D}^k$ with $k \in \Omega\left(1/\varepsilon^2 \log\left(\delta^{-1}\rho(\mathcal{H}, \varepsilon/\ell_\psi)\rho(\mathcal{F}, \varepsilon/\ell_\psi)\right)\right)$:*

$$\left| \mathbb{E}_{\mathcal{G} \sim \mathrm{unif}(\mathcal{S})} \left[ \psi^{\mathcal{G}}(\boldsymbol{h}^r(\mathcal{G}), \boldsymbol{f}^r(\mathcal{G}, \boldsymbol{h}^r(\mathcal{G}))) \right] - \mathbb{E}_{\mathcal{G} \sim \mathcal{D}} \left[ \psi^{\mathcal{G}}(\boldsymbol{h}(\mathcal{G}), \boldsymbol{f}(\mathcal{G}, \boldsymbol{h}(\mathcal{G}))) \right] \right| \leq \varepsilon .$$

## 5 EXPERIMENTAL RESULTS

**Arrow-Debreu exchange economies.**[5] Our first set of experiments aim to solve CE in Arrow-Debreu exchange economies (Arrow & Debreu, 1954). The difficulty in solving the pseudo-game associated with Arrow-Debreu exchange economies—hereafter *exchange economies*—arises from the fact that it does not fit into any well-defined categories of pseudo-games, e.g., monotone or jointly convex, for which there are algorithms that converge to GNEs.

An *exchange economy* $(\boldsymbol{u}, \boldsymbol{E})$ consists of a finite set of $m \in \mathbb{N}_+$ goods and $n \in \mathbb{N}_+$ consumers (or traders). Every consumer $i \in [n]$ has a set of possible consumptions $\mathcal{X}_i \subseteq \mathbb{R}_+^m$, an endowment of goods $\boldsymbol{e}_i = (e_{i1}, \ldots, e_{im}) \in \mathbb{R}^m$ and a utility function $u_i : \mathbb{R}^m \to \mathbb{R}$. We denote $\boldsymbol{E} = (\boldsymbol{e}_1, \ldots, \boldsymbol{e}_n)^T$. Any exchange economy can be formulated as a pseudo-game whose set of GNE is equal to the set of competitive equilibria (CE)[6] of the original economy (Arrow & Debreu, 1954). This pseudo-game consists of $n + 1$ agents, who correspond to the $n$ buyers and a seller. The pseudo-game is given by the following optimization problem, for each buyer $i \in [n]$, and the seller, respectively:

$$\max_{\boldsymbol{x}_i \in \mathcal{X}_i : \boldsymbol{x}_i \cdot \boldsymbol{p} \leq \boldsymbol{e}_i \cdot \boldsymbol{p}} u_i(\boldsymbol{x}_i) \quad \Bigg| \quad \max_{\boldsymbol{p} \in \Delta_m} \boldsymbol{p} \cdot \left( \sum_{i \in [n]} \boldsymbol{x}_i - \sum_{i \in [n]} \boldsymbol{e}_i \right).$$

Let $\boldsymbol{v}_i \in \mathbb{R}_+^m$, $\boldsymbol{\rho} \in (-\infty, 1]^n$, be a vector of parameters for the utility function of buyer $i \in [n]$. In our experiments, we consider the following utility function classes: *Linear*: $u_i(\boldsymbol{x}_i) = \sum_{j \in [m]} v_{ij} x_{ij}$, *Cobb-Douglas*: $u_i(\boldsymbol{x}_i) = \prod_{j \in [m]} x_{ij}^{v_{ij}}$, *Leontief*: $u_i(\boldsymbol{x}_i) = \min_{j \in [m]} \{x_{ij}/v_{ij}\}$, and *constant elasticity of substitution (CES)*: $u_i(\boldsymbol{x}_i) = (\sum_{j \in [m]} v_{ij} x_{ij}^\rho)^{1/\rho}$. When we take $\rho = 1$, $\rho \to 0$, and $\rho \to -\infty$ for CES utilities, we obtain linear, Cobb-Douglas, and Leontief utilities respectively. We denote $\boldsymbol{V} = (\boldsymbol{v}_1, \ldots, \boldsymbol{v}_n)^T$. As is standard in the literature (Cheung et al., 2013; Brânzei et al., 2021), we assume that for all buyers $i \in [n]$, $\mathcal{X}_i = \mathbb{R}_+^m$. Once a utility function class is fixed, an exchange economy is referred to with the name of the utility function, and can be sampled as a tuple $(\boldsymbol{V}, \boldsymbol{E}) \in \mathbb{R}^{n \times m} \times \mathbb{R}^{n \times m}$ for linear, Cobb-Douglas, and Leontief exchange economies, and as a tuple $(\boldsymbol{V}, \boldsymbol{\rho}, \boldsymbol{E}) \in \mathbb{R}^{n \times m} \times \mathbb{R}^n \times \mathbb{R}^{n \times m}$ for CES exchange economies. For CES exchange economies, we have a *gross substitute (GS)* and *gross complements (GC)* CES economy, either when $\rho_i \geq 0$ for all buyers or $\rho_i < 0$ for all buyers, respectively. Otherwise, this is a mixed CES economy.

*Baselines.* For special cases of exchange economies, the computation of CE is well-studied (e.g., Bei et al. (2015)), allowing us to compare the performance of GAES to known specialized methods. We benchmark GAES to *tâtonnement* (Walras, 1896), which is an auction-like algorithm that is guaranteed to converge for CES utilities with $\boldsymbol{\rho} \in [0, 1)^n$ (Bei et al., 2015), including Cobb-Douglas and excluding Linear utilities. We also benchmark to *exploitability descent* (Goktas & Greenwald, 2022). For each of these baselines, we run an extensive grid search over decreasing learning rates during validation (see Appendix G.2). Each baseline was run to convergence.

We report the results of two experiments. First, we run our algorithms in linear, Cobb-Douglas, Leontief, GS CES, GC CES, and mixed CES exchange economies (we defer the results from the GS

---

[5]We include experiments on normal-form games, as well as all missing additional implementation details and network architecture diagrams in Appendix G.

[6]We refer the reader to Appendix G.2 for a definition.

CES, and GC CES experiments to the Appendix). We report the distribution of the exploitability on the test set for GAES and the baselines in Figure 10 (additional plots can be found in Appendix G.2). We measure performance w.r.t. the *normalized exploitability*, which is the exploitability of the method divided by the average exploitability over the action space. We observe that in all economies, GAES outputs an action profile that is on average better than at least $99\%$ of the action profiles in terms of exploitability. In all four markets, GAES on average achieves lower exploitability than the baselines (see Figure 2). We also see in Figure 10 (Appendix G.2) that GAES outperforms the baselines, in distribution, in every economy except Cobb-Douglas. This is not surprising since tâtonnement is guaranteed to converge in Cobb-Douglas economies (Bei et al., 2015). That said, tâtonement does not outperform GAES on average, since tâtonnement's convergence guarantees hold for different learning rates in each market.

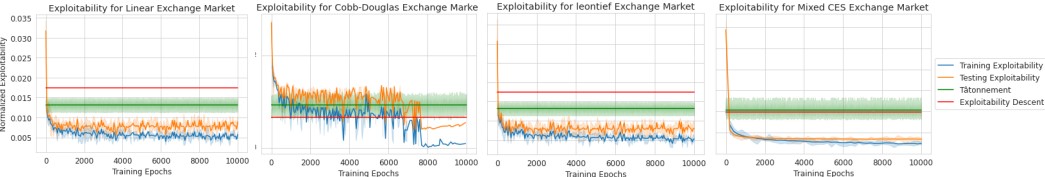

Figure 2: Training and Testing Exploitability of GAES in linear, Cobb-Douglas, Leontief, and mixed CES economies. GAES outperforms all baselines on average in all economies.

Second, we compare GAES with the performance of tâtonnment in a pathological and well-know Leontief exchange economy, *the Scarf economy* (Scarf, 1960) (Figure 3). Here, we soft start tâtonnement with the output of GAES with some added uniform noise. This additional noise ensures that the starting point of tâtonnement is distinct from the output of GAES). We see on Figure 3 that the prices generated by tâtonnement spiral out and settle into an orbit. This makes sense as, unlike pure Nash equilibria, GNE and CE are not locally stable (Flokas et al., 2020), meaning that soft-starting the algorithm with the output of GAES might be worse than using GAES alone. The success of GAES in Leontief exchange economies as well as the Scarf economy is notable, and suggests that GAES might be smoothing out the loss landscape since for these economies the exploitability is non-differentiable.

**Kyoto joint implementation mechanism.** We solve a pseudo-game model of the joint implementation mechanism proposed in the Kyoto protocols (Protocol, 1997). The *Kyoto Joint Implementation Mechanism* is a *cap-and-trade* mechanism that bounds each country that signed onto the protocol to emit anthropogenic gases below a particular emission cap. Countries bound by the mechanism can also invest in green projects in other countries, which in return increases their emission caps. Breton et al. (2006) introduce a model of the Kyoto Joint Implementation Mechanism, using this to predict the impact of the mechanism. The model that the authors propose is partially solvable analytically, that is, one can characterize equilibria qualitatively using comparative statics (Nachbar, 2002), but cannot obtain closed-form solutions for GNE. Moreover, there is no algorithm that is known to converge to the GNE of this pseudo-game, since it is not monotone.

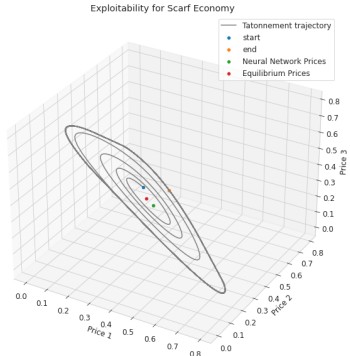

Figure 3: A phase portrait of equilibrium prices in the Scarf Economy. While the output of GAES is close to the equilibrium prices, the final prices outputted by tâtonnement prices are further than the starting prices.

Formally, the *(Kyoto) Joint Implementation Mechanism* (JI) consists of $n \in \mathbb{N}_+$ countries. Each *country* $i \in [n]$, can make decisions that result in environmentally damaging anthropogenic *emissions*, $e_i \in \mathbb{R}_+$, and can make investments $\boldsymbol{x}_i \in \mathbb{R}^n$ to offset their emissions. The investments $\boldsymbol{x}_i \in \mathbb{R}^n$ that each country $i \in [n]$ makes in another country $j \in [n]$ offsets that country's emissions by $x_{ij}\gamma_j$, where this is in proportion to an investment return rate $\gamma_j > 0$ for country $j$, with $\boldsymbol{\gamma} \in \mathbb{R}^n_+$. Each country $i$ has a revenue $r_i : \mathbb{R}^n_+ \to \mathbb{R}$, which is a function of its emissions $e_i$, a cost function $c_i : \mathbb{R}^{n \times n} \to \mathbb{R}$ that is a function of all investments, and a negative externality function, $d_i : \mathbb{R}^n \to \mathbb{R}$, that is a function of the net emissions of all countries.

Each country $i$ aims to maximize their surplus, i.e., their revenue minus costs and and negative externalities, constrained by keeping their net emissions under an emission cap, $\eta_i \in \mathbb{R}_+$. Additionally, we require the emission transfer balance to hold, i.e., $\forall i, e_i - \gamma_i \sum_{j \in [n]} x_{ji} \geq 0$, that is no country can transfer more emission reduction than they have, giving for each country, $i$:

$$\max_{\substack{(e_i, \boldsymbol{x}_i) \in \mathbb{R}_+ \times \mathbb{R}_+^n : e_i - \sum_{j \in [n]} x_{ij} \gamma_j \leq \eta_i \\ \forall i \in [n], e_i - \gamma_i \sum_{j \in [n]} x_{ji} \geq 0}} \quad r_i(e_i) \quad - \quad c_i(\boldsymbol{X}) \quad - \quad d_i\left(\left\{e_i - \sum_{j \in [n]} x_{ji} \gamma_i\right\}_i\right) \qquad \text{The}$$

literature has traditionally assumed that $r_i(e_i; \theta_i) = e_i(\theta_i - e_i/2)$, $c_i(\boldsymbol{X}) = \frac{1}{2}\left(x_{ii}^2 + \sum_{j \neq i}\left[(x_{ij} + x_{jj})^2 - x_{jj}^2\right]\right)$, and $d_i(\boldsymbol{e} - \sum_{i \in [n]} \boldsymbol{x}_i, \beta_i) = \beta_i \sum_{i \in [n]}(e_i - \sum_{j \in [n]} x_{ij})$. Fixing these functional forms, we can sample $(\boldsymbol{\theta}, \boldsymbol{\beta}, \boldsymbol{\gamma}, \boldsymbol{\eta}) \in \mathbb{R}_+^m \times \mathbb{R}^m \times \mathbb{R}^{m \times m} \times \mathbb{R}^m$ and obtain a representation of the JI mechanism.

We run two different experiments to replicate and extend the analysis of Breton et al. (2006). We first replicate their qualitative analysis of equilibria (Figure 4). Breton et al. (2006) introduce a comparative static analysis, in which they fix all parameters of JI except for $\boldsymbol{\theta}$, and characterize the kinds of GNE as $\boldsymbol{\theta}$ varies. In Figure 4, the six regions correspond to different kinds of GNE. For instance in Region 1, both countries emit strictly less than their emission cap (see Section 4 of Breton et al. (2006)). The parts of the plot that are not shaded correspond to pseudo-games for which GNE are not unique, and for which equilibria cannot be characterized analytically. We superpose on top of this plot a set of pseudo-games from an unseen test set, and color each pseudo-game by the color of the region whose condition they fulfill.

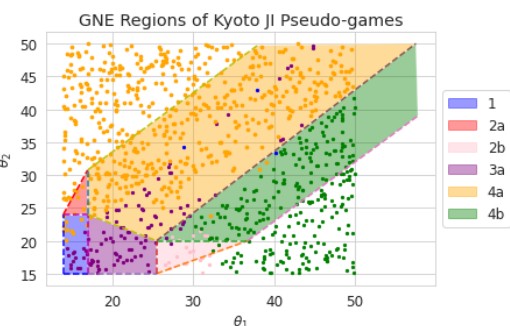

Figure 4: A taxonomy of different equilibrium types for various pseudo-games obtained by fixing all parameters, and varying $\boldsymbol{\theta}$. The $x$ and $y$ axes represent the revenue parameters $\theta_1, \theta_2$ of the countries. The colored regions (obtained analytically) correspond to different qualitative types of GNE while the dots correspond to pseudo-games in the test set colored by the GNE types that were predicted for them by GAES.

We observe that with the exception of Regions 1 and 2a, the structure of the GNE generated by GAES lines up well with the type of GNE. We believe that failure to predict Regions 1 and 2a is due to the closeness of the action profiles that fit either of these equilibrium types to other ones. For instance, GAES predicts a GNE of type 4a for pseudo-games in Region 2a but these GNE, although qualitatively different, are very close in action space: the only difference between the two regions is that for the type 2a GNE, one player emits strictly less than its cap and the other emits exactly its cap, while for the type 4a GNE, both players emit exactly their emission cap. A similar conclusion holds for GNE in Region 3a, as predicted in Region 1.

We also solve the JI pseudo-game for a distribution of JI pseudo-games (Figure 14, Appendix G), and these results confirm that the testing normalized exploitability is very low. We note that *normalized exploitability* is given as the exploitability divided by the average exploitability over the action space, which means that GAES has on average a lower exploitability than $\approx 99.5\%$ of the feasible action profiles. This confirms our hypothesis that failure to predict Regions 1 and 2a in Figure 4 arises from the proximity between GNE of these types and GNE of types 3a or 4a respectively, since according to exploitability there is very little improvement left for GAES.

## 6  CONCLUSION

We introduced GAES, a GAN that learns mappings from pseudo-games to GNE. GAES outperforms existing methods to compute GNE or CE in exchange economies, and solves even pathological examples, i.e., Scarf economies. Our approach extends the use of exploitability-based learning methods from normal-form games to continuous games, exchange markets, and beyond. GAES adds to the growing list of differentiable economics methods that aim to provide practitioners with computational tools for the study of economic properties. GAES extends the range of models for which we have approximate and reliable solvers.

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
