## A PSEUDO-GAMES VS. GAMES

To see why GNEs cannot be expressed as Nash equilibria, consider a cake cutting problem between two players, in which each agent $i = 1, 2$ receives payoff $u_i(x_i, x_{-i}) = x_i^2 - \frac{1}{1-x_{-i}^2}$ where $x_{-i}$ is the action of $i$'s opponent. In this problem, each player can request a share of the cake, i.e., for all $i$, $x_i \in [0, 1]$, and the total share of the cake demanded must be less than or equal to 1, i.e., $x_1 + x_2 \leq 1$. This is a pseudo-game where any player $i$ can take an action $x_i \in [0, x_{-i}]$. A solution can then be modelled as a GNE , i.e., $(x_1^*, x_2^*)$ s.t. $x_i^* \in \arg\max_{x_i \in [0, x_{-i}^*]} u_i(x_i, x_{-i}^*)$, which corresponds to the set $\Delta_2$, i.e., the unit simplex in $\mathbb{R}^2$. Although the cake splitting problem cannot be expressed as a game due to the joint constraint $x_1 + x_2 \leq 1$, a common first intuition of many who are familiar with games is to penalize the payoffs of the players by $-\infty$ for any $(x_1, x_2)$ such that $x_1 + x_2 \geq 1$. This then gives us the following payoff:

$$u_i(x_i, x_{-i}) = \begin{cases} x_i^2 - \frac{1}{1-x_{-i}^2} & \text{if } x_1 + x_2 \leq 1, \text{ or} \\ -\infty & \text{otherwise.} \end{cases} \tag{3}$$

The Nash equilibria of the game defined by the above payoffs are $\{(x_1, 1) \mid x_1 \in [0, 1]\} \cup \{(1, x_2) \mid x_2 \in [0, 1]\} \cup \Delta_2$, and even if the penalty term was more than $-\infty$, the set of Nash equilibria would not be equal to the set of GNE. Note that $\{(x_1, 1) \mid x_1 \in [0, 1]\}$ are all Nash equilibria since the payoff of the first player is $-\infty$ not matter what actions it chooses. A similar argument holds for $\{(1, x_2) \mid x_2 \in [0, 1]\}$.

## B GNE COMPUTATION METHODS SURVEY

Following Arrow & Debreu's introduction of GNE, Rosen (1965) initiated the study of the mathematical and computational properties of GNE in pseudo-games with jointly convex constraints, proposing a projected gradient method to compute GNE. Thirty years later, Uryas'ev & Rubinstein (1994) developed the first relaxation methods for finding GNEs, which were improved upon in subsequent works Krawczyk & Uryasev (2000); Heusinger & Kanzow (2009). Two other types of algorithms were also introduced to the literature: Newton-style methods Facchinei et al. (2009); Dreves (2017); von Heusinger et al. (2012); Izmailov & Solodov (2014); Fischer et al. (2016); Dreves et al. (2013) and interior-point potential methods Dreves et al. (2013). Many of these approaches are based on minimizing the exploitability of the pseudo-game, but others use variational inequality Facchinei et al. (2007); Nabetani et al. (2011) and Lemke methods Schiro et al. (2013).

Additional, novel methods that transform GNE problems to Nash equilibria problems have also been analyzed. These models take the form of either exact penalization methods, which lift the constraints into the objective function via a penalty term Facchinei & Lampariello (2011); Fukushima (2011); Kanzow & Steck (2018); Ba & Pang (2020); Facchinei & Kanzow (2010b), or augmented Lagrangian methods Pang & Fukushima (2005); Kanzow (2016); Kanzow & Steck (2018); Bueno et al. (2019), which do the same but augmented by dual Lagrangian variables. Using these methods, Jordan et al. (2022) provide the first convergence rates to a $\varepsilon$-GNE in monotone (resp. strongly monotone) pseudo-games with jointly affine constraints in $\tilde{O}(1/\varepsilon)$ ($\tilde{O}(1/\sqrt{\varepsilon})$) iterations. These algorithms, despite being highly efficient in theory, are numerically unstable in practice Jordan et al. (2022). Nearly all of the aforementioned approaches concern pseudo-games with jointly convex constraints. Goktas & Greenwald (2021) also introduce first-order methods to minimize exploitability in a large class of jointly convex pseudo-games in polynomial-time.

## C GNE APPLICATIONS

Some economic applications of pseudo-games and GNE solvers include energy resource allocation Hobbs & Pang (2007); Jing-Yuan & Smeers (1999), environmental protection Breton et al. (2006); Krawczyk (2005), cloud computing Ardagna et al. (2017; 2011b), ride sharing services (Jeff) Ban et al. (2019), transportation Stein & Sudermann-Merx (2018), capacity allocation in wireless and network communication Han et al. (2011); Pang et al. (2008), and applications to machine learning such as adversarial classification Bruckner et al. (2012); Bruckner & Scheffer (2009). Competitive equilibria concepts have also been used to solve many problems in resource allocation Varian (1973);

Gutman & Nisan (2012), with specific applications to college course allocation Budish (2011), pricing of cloud computing Lai et al. (2005); Zahedi et al. (2018), ad market platforms Conitzer et al. (2022a), economic forecasting Partridge & Rickman (2010), and economic policy assessment Dixon & Parmenter (1996).

One of the other motivations of research in pseudo-games is their mathematical significance for general equilibrium theory, the branch of microeconomics which models the interactions between economic agents Facchinei & Kanzow (2010a). General equilibrium theory is a cornerstone of economic theory Debreu (1996), and is also widely used in policy analysis Dixon & Parmenter (1996). The most established general equilibrium model is the Arrow-Debreu model of a competitive economy Arrow & Debreu (1954), which is an instantiation of a pseudo-game in which a seller sets prices for commodities, a set of firms choose what quantity of each commodity to produce, and a set of consumers choose what quantity of each commodity to consume in exchange for their endowment of each commodity. This model is a pseudo-game, rather than a game, because the prices set by the sellers determine the value of the consumers' endowments, i.e., their budget, which in turn determines the consumptions of goods they can afford.

The canonical solution concept for this model, and other general equilibrium models more broadly Facchinei & Kanzow (2010a), is a *competitive equilibrium (CE)* Walras (1896); Arrow & Debreu (1954). Here, there is a collection of demands, one per consumer, a collection of supplies, one per firm, and prices, one per commodity, such that given equilibrium prices: 1) no consumer can increase their utility by unilaterally deviating to a consumption they can afford, 2) no firm can increase their profit by deviating to another feasible production schedule, and 3) the aggregate demand for each commodity (i.e., the sum of the commodity's consumption across all consumers) is equal to its aggregate supply (i.e., the sum of the commodity's production and endowments across firms and consumers respectively), while the total cost of the aggregate demand is equal to the total cost of the aggregate supply . CE are intrinsically related to GNE since the set CE of an Arrow-Debreu competitive economy corresponds to the set of GNE of the corresponding pseudo-game. This approach also works for general equilibrium models more broadly: assuming local non-satiation of consumer preferences, and given a general equilibrium model, one can construct an associated pseudo-game and show that the set of CE is equal to the set of GNE of the associated pseudo-game.

**Potential Applications for GAES.** Applications of economic equilibrium concepts such as GNE and CE often require a decision maker to solve a fixed parametric model either 1) *en masse* over a distribution of parameters or 2) quickly in an iterative fashion for a sequence of different parameters.

The first common use case for the former setting occurs in internet platforms that have to price advertisers in exchange for ad impressions. One standard approach to solve this problem is to let advertisers compete in sequential first price auctions, where winning each auction gives the advertiser the right to show their ad to the website visitor associated with the auction Bigler (2019). As the number of auctions that each advertiser participates in is enormous, the bidding procedure on these platforms is automated. However, beyond certain large advertising companies, it is in general hard for advertisers to come-up with effective automated bidding strategies, as a result companies provide their own bidding strategies to advertisers. One example of these strategies are first-price pacing equilibria, in which the platform seeks a vector of pacing multipliers, one for each advertiser, and buyers bid their value times their pacing multiplier. These pacing multipliers correspond to CE Conitzer et al. (2022a;b), but for large platforms many ad markets run simultaneously, which requires these platforms to solve for CE *en masse*.

A second common use case, is in computable general equilibrium, i.e., the study of economic data through the lens of general equilibrium theory, which uses CE to make economic forecasts Dixon & Parmenter (1996). In these applications, in order to understand the takeaways from a general equilibrium model on the economy, one fixes certain parameters of the model and varies others to understand the consequences of a change in parts of the economy on the economy as whole. This practice of *comparative statics* Nachbar (2002), requires once again to solve the model for a family of parameters *en masse*.

For the setting which requires fast and iterative computation of GNE and CE, a common application is in the context of shared computational resources on platforms such as Mesos Hindman et al. (2011), Quincy Isard et al. (2009), Kubernetes Burns et al. (2016), and Yarn Vavilapalli et al. (2013). To this end, a long line of work has studied resource sharing on computing clusters Chen et al.

(2018); Ghodsi et al. (2011; 2013); Parkes et al. (2015), with many methods making use of the repeated computation of competitive equilibrium Gutman & Nisan (2012); Lai et al. (2005); Budish (2011); Zahedi et al. (2018); Varian (1973) or generalized Nash equilibria Ardagna et al. (2017; 2011b;a); Anselmi et al. (2014). In such settings, as consumers request resources from the platform, the platforms have to compute an equilibrium iteratively while handling all numerical failures within a given time frame.

Another application that requires one to solve for GNE and CE iteratively is that of state-value function-based reinforcement leaning algorithms for solving market equilibria in stochastic environments, such as *Model-based Optimistic Online Learning for Markov Exchange Economy* and *Model-based Pessimistic Online Learning for Markov Exchange Economy*, which iteratively construct state-value functions by solving a sequence of competitive equilibrium problems that have to be solved quickly Liu et al. (2022). Other related online learning algorithms such as *randomized exchange equilibrium learning* Guo et al. (2021), which compute a competitive equilibrium when agents' payoffs can be obtained from sample observations, also require solving for a competitive equilibrium iteratively and quickly.

## D  NON-LIPSCHITZ EXPLOITABILITY AND UNBOUNDED GRADIENTS

**Example 1.** *Consider a two-player pseudo-game with action space $\mathcal{A}_1 = \mathcal{A}_2 = [0, 1]$, payoffs $u_1(a_1, a_2) = a_1$ $u_2(a_1, a_2) = a_2$, and constraints $h_1(a_1, a_2) = a_2 - a_1^2$, $h_2(a_1, a_2) = a_1 - a_2^2$. The exploitability of this pseudo-game for all $\boldsymbol{a} = [0, 1]^2$ is given by*

$$\varphi(\boldsymbol{a}) = \max_{(b_1, b_2): \substack{a_2 - b_1^2 \geq 0 \\ a_1 - b_2^2 \geq 0}} b_1 + b_2 - a_1 - a_2 \tag{4}$$

$$= \max_{(b_1, b_2): \substack{\sqrt{a_2} \geq b_1 \\ \sqrt{a_1} \geq b_2}} b_1 + b_2 - a_1 - a_2 \tag{5}$$

$$= \sqrt{a_1} + \sqrt{a_2} - a_1 - a_2 \tag{6}$$

*The gradient of the exploitability, when it exists is given by $\frac{\partial \varphi}{\partial a_i}(\boldsymbol{a}) = \frac{1}{2\sqrt{a_i}} - 1$ for $i = 1, 2$. Note that the payoffs $\boldsymbol{u}$ and constraints $\boldsymbol{h}$ are both Lipschitz-continuous over $[0, 1]^2$, however, whenever $a_1 = 0$ or $a_2 = 0$, the exploitability grows unboundedly, i.e., if for some $a_i \to 0$, then $\frac{\partial \varphi}{\partial a_i}(\boldsymbol{a}) \to \infty$, and hence exploitability is not Lipschitz continuous, and its gradients cannot be bounded over the set $[0, 1]$.*

Example 1 shows that exploitability in pseudo-games behaves differently than in normal-form games, where exploitability is Lipschitz-continuous. The fact that gradients are unbounded means in turn turn that gradients can explode during training if the GNE is located near the non-differentiability (Example 1 shows this can happen, since the GNE strategy for either player can occur at 0). As a result, first-order methods can fail, not only theoretically but also in practice.

## E  ADDITIONAL PRELIMINARY DEFINITIONS

**Notation.** We use caligraphic uppercase letters to denote sets (e.g., $\mathcal{X}$); bold lowercase letters to denote vectors (e.g., $\boldsymbol{p}, \boldsymbol{\pi}$); bold uppercase letters to denote matrices (e.g., $\boldsymbol{X}, \boldsymbol{\Gamma}$), lowercase letters to denote scalar quantities (e.g., $x, \delta$). We denote the $i$th row vector of a matrix (e.g., $\boldsymbol{X}$) by the corresponding bold lowercase letter with subscript $i$ (e.g., $\boldsymbol{x}_i$). Similarly, we denote the $j$th entry of a vector (e.g., $\boldsymbol{p}$ or $\boldsymbol{x}_i$) by the corresponding Roman lowercase letter with subscript $j$ (e.g., $p_j$ or $x_{ij}$).

**Models.** An $\varepsilon$-*variational equilibrium (VE)* (or *normalized GNE*) of a pseudo-game is a strategy profile $\boldsymbol{a}^* \in \mathcal{X}$ s.t. for all $i \in [n]$ and $\boldsymbol{a} \in \mathcal{X}$, $u_i(\boldsymbol{a}^*) \geq u_i(\boldsymbol{a}_i, \boldsymbol{a}^*_{-i}) - \varepsilon$. We note that in the above definitions, one could just as well write $\boldsymbol{a}^* \in \mathcal{X}(\boldsymbol{a}^*)$ as $\boldsymbol{a}^* \in \mathcal{X}$, as any fixed point of the joint action correspondence is also a jointly feasible action profile, and vice versa. A VE is an $\epsilon$-VE with $\varepsilon = 0$. Under our assumptions, while GNE are guaranteed to exist in all pseudo-games by Arrow & Debreu's lemma on abstract economies Arrow & Debreu (1954), VE are only guaranteed to exist in

pseudo-games with jointly convex constraints Von Heusinger & Kanzow (2009). Note that the set of $\varepsilon$-VE of a pseudo-game is a subset of the set of the $\varepsilon$-GNE, as $\mathcal{X}(\boldsymbol{a}^*) \subseteq \mathcal{X}$, for all $\boldsymbol{a}^*$ which are GNE of $\mathcal{G}$. The converse, however, is not true, unless $\mathcal{A} \subseteq \mathcal{X}$. Further, when $\mathcal{G}$ is a game, GNE and VE coincide; we refer to this set simply as NE.

**Mathematical Preliminaries.** Fix any norm $\|\cdot\|$. Given $\mathcal{A} \subset \mathbb{R}^n$, the function $f : \mathcal{A} \to \mathbb{R}$ is said to be $\ell_f$-*Lipschitz-continuous* iff $\forall \boldsymbol{x}_1, \boldsymbol{x}_2 \in \mathcal{X}, \|f(\boldsymbol{x}_1) - f(\boldsymbol{x}_2)\| \leq \ell_f \|\boldsymbol{x}_1 - \boldsymbol{x}_2\|$. Consider a function $f : \mathcal{X} \to \mathcal{Y}$, we denote its Lipschitz-continuity constant by $\ell_f$. If the gradient of $f$ is $\ell_{\nabla f}$-Lipschitz-continuous, we then refer to $f$ as $\ell_{\nabla f}$-*Lipschitz-smooth*. A function $f : \mathcal{X} \to \mathbb{R}$ is said to be *convex* if $f(\boldsymbol{x}) \geq f(\boldsymbol{y}) + \nabla f(\boldsymbol{y}) \cdot (\boldsymbol{x} - \boldsymbol{y})$, for all $\boldsymbol{x}, \boldsymbol{y} \in \mathcal{X}$ and concave if $-f$ is convex. A function $f : \mathcal{A} \to \mathbb{R}$ is said to be $\mu$-*Polyak-Lojasiewicz (PL)* if for all $\boldsymbol{x} \in \mathcal{X}, 1/2 \|\nabla f(\boldsymbol{x})\|_2^2 \geq \mu(f(\boldsymbol{x}) - \min_{\boldsymbol{x} \in \mathcal{X}} f(\boldsymbol{x}))$. A function $f : \mathcal{A} \to \mathbb{R}$ is said to be $\mu$-*quadratically growing (QG)*, if for all $\boldsymbol{x} \in \mathcal{X}, f(\boldsymbol{x}) - \min_{\boldsymbol{x} \in \mathcal{X}} f(\boldsymbol{x}) \geq \mu/2 \|\boldsymbol{x}^* - \boldsymbol{x}\|^2$ where $\boldsymbol{x}^* \in \arg\min_{\boldsymbol{x} \in \mathcal{X}} f(\boldsymbol{x})$.

# F  OMMITED RESULTS AND PROOFS

We first revisit the proof of the observation that is central to GAES.

**Observation 1.** *For any* $\mathcal{D} \in \Delta(\Gamma)$, *we have* $\min_{\boldsymbol{h} \in \mathcal{X}^\Gamma} \mathbb{E}_{\mathcal{G} \sim \mathcal{D}}\left[\varphi^{\mathcal{G}}(\boldsymbol{h}(\mathcal{G}))\right] = \min_{\boldsymbol{h} \in \mathcal{X}^\Gamma} \max_{\boldsymbol{f} \in \mathcal{A}^\Gamma : \forall \mathcal{G} \in \Gamma, \boldsymbol{f}(\mathcal{G}) \in \mathcal{X}^{\mathcal{G}}(\boldsymbol{h}(\mathcal{G}))} \mathbb{E}_{\mathcal{G} \sim \mathcal{D}}\left[\psi^{\mathcal{G}}(\boldsymbol{h}(\mathcal{G}), \boldsymbol{f}(\mathcal{G}))\right]$

*Proof of Observation 1.*

$$\min_{\boldsymbol{h} \in \mathcal{X}^\Gamma} \mathbb{E}_{\mathcal{G} \sim \mathcal{D}}\left[\varphi^{\mathcal{G}}(\boldsymbol{h}(\mathcal{G}))\right] \tag{7}$$

$$= \min_{\boldsymbol{h} \in \mathcal{X}^\Gamma} \mathbb{E}_{\mathcal{G} \sim \mathcal{D}}\left[\sum_{i \in [n]} \max_{\boldsymbol{b}_i \in \mathcal{X}_i(\boldsymbol{h}_{-i}(\mathcal{G}))} \text{Regret}_i^{\mathcal{G}}(\boldsymbol{h}_i(\mathcal{G}), \boldsymbol{b}_i; \boldsymbol{h}_{-i}(\mathcal{G}))\right] \tag{8}$$

$$= \min_{\boldsymbol{h} \in \mathcal{X}^\Gamma} \mathbb{E}_{\mathcal{G} \sim \mathcal{D}}\left[\max_{\boldsymbol{b} \in \mathcal{X}(\boldsymbol{h}(\mathcal{G}))} \sum_{i \in [n]} \text{Regret}_i^{\mathcal{G}}(\boldsymbol{h}_i(\mathcal{G}), \boldsymbol{b}_i; \boldsymbol{h}_{-i}(\mathcal{G}))\right] \quad \text{(Observation 3.2. Goktas \& Greenwald (2022))}$$

$$\tag{9}$$

$$= \min_{\boldsymbol{h} \in \mathcal{X}^\Gamma} \mathbb{E}_{\mathcal{G} \sim \mathcal{D}}\left[\max_{\boldsymbol{b} \in \mathcal{X}(\boldsymbol{h}(\mathcal{G}))} \psi^{\mathcal{G}}(\boldsymbol{h}(\mathcal{G}), \boldsymbol{b})\right] \tag{10}$$

$$= \min_{\boldsymbol{h} \in \mathcal{X}^\Gamma} \mathbb{E}_{\mathcal{G} \sim \mathcal{D}}\left[\max_{\boldsymbol{f}^{\mathcal{G}} \in \mathcal{A}^\Gamma : \boldsymbol{f}^{\mathcal{G}}(\mathcal{G}) \in \mathcal{X}^{\mathcal{G}}(\boldsymbol{h}(\mathcal{G}))} \psi^{\mathcal{G}}(\boldsymbol{h}(\mathcal{G}), \boldsymbol{f}^{\mathcal{G}}(\mathcal{G}))\right] \tag{11}$$

$$= \min_{\boldsymbol{h} \in \mathcal{X}^\Gamma} \max_{\boldsymbol{f} \in \mathcal{A}^\Gamma : \forall \mathcal{G} \in \Gamma, \boldsymbol{f}(\mathcal{G}) \in \mathcal{X}^{\mathcal{G}}(\boldsymbol{h}(\mathcal{G}))} \mathbb{E}_{\mathcal{G} \sim \mathcal{D}}\left[\psi^{\mathcal{G}}(\boldsymbol{h}(\mathcal{G}), \boldsymbol{f}(\mathcal{G}))\right] \tag{12}$$

In Equation (11), for each pseudo-game we optimize over the space of functions from (pseudo-game, action profile) pairs to action profiles. Equation (12) then follows because the solution to the optimization problem over $\boldsymbol{f}^{\mathcal{G}}$ is independent for each game $\mathcal{G} \in \Gamma$, allowing us to pull it out of the expectation and optimize over $\boldsymbol{f}$ for all pseudo-games. In words, optimizing action profiles for each pseudo-game is equivalent to optimizing a function point-wise over all pseudo-games. To see this clearer, consider the following example:

$$1/2 \max_{x_1 \in [0,1]} (x_1) + 1/2 \max_{x_2 \in [0,1]} (x_2 - 1/2)^2 \tag{13}$$

The optimal value of the above problem is equal to $1/2(1) + 1/2(1/4) = 5/8$ with the solution to the first maximization being $x_1^* = 1$ and to the second $x_2^* = 1/4$.

Notice that instead of optimizing each variable individually, we can instead optimize over a function space $f : \{1, 2\} \to [0, 1]$, where $f(1)$ and $f(2)$ correspond respectively to the value of $x_1$ and $x_2$ ,

giving us:

$$\frac{1}{2} \max_{x_1 \in [0,1]} (x_1) + \frac{1}{2} \max_{x_2 \in [0,1]} (x_2 - \frac{1}{2})^2 \tag{14}$$

$$= \left\{ \frac{1}{2} \max_{f \in [0,1]^{\{1,2\}}} f(1) + \frac{1}{2} \max_{f \in [0,1]^{\{1,2\}}} (f(2) - \frac{1}{2})^2 \right\} \tag{15}$$

$$= \max_{f \in [0,1]^{\{1,2\}}} \left\{ \frac{1}{2} f(1) + \frac{1}{2} (f(2) - \frac{1}{2})^2 \right\} \tag{16}$$

where the last line follows from the fact that the maximization problem in any summand is independent from the other. Indeed, the above optimization's value is also $5/8$ and its solution is the function $f^*(1) = 1, f^*(2) = 1/4$.

$\square$

### F.1 CONVERGENCE AND GENERALIZATION BOUND

We now present the proof of Theorem 4.1. At a high-level, we first show that under our assumptions (Assumption 1) the weights of the discriminator satisfy a gradient domination condition (also known as the PL condition Lojasiewicz (1963); Bassily et al. (2018)). With this lemma in hand, our min-max optimization problem becomes a non-convex-PL optimization problem. Our proof then proceeds as follows. We first bound the euclidean distance between the gradient of the empirical cumulative regret and the empirical exploitability for the weight iterates computed by our algorithm, i.e., $\left\| \nabla \widehat{\varphi}(\boldsymbol{w}^{\boldsymbol{h},(t)}) - \nabla_{\boldsymbol{w}^{\boldsymbol{h}}} \widehat{\psi} \left( \boldsymbol{w}^{\boldsymbol{h},(t)}, \boldsymbol{w}^{\boldsymbol{f},(t)} \right) \right\|_2$, as a function of $T_{\boldsymbol{f}}$. This requires a quadratic growth condition, i.e., that our empirical exploitability satisfies w.r.r. the discriminator's weights $\boldsymbol{w}^{\boldsymbol{f}}$. We then give a progress lemma for $\boldsymbol{w}^{\boldsymbol{h}}$ for the outer loop of our algorithm, which can be telescoped to obtain our convergence theorem.

The algorithm's convergence rate $\tilde{O}(1/\epsilon^3)$ is much faster than known convergence rates for similar non-convex-concave problems $\tilde{O}(1/\epsilon^6)$ Lin et al. (2020). Moreover, we conjecture that a faster convergence rate is not achievable by simultaneous gradient descent ascent type algorithms since even for PL-PL min-max optimization, the best known convergence rate of simultaneous gradient descent ascent type algorithms is $\tilde{O}(1/\varepsilon^{10})$ Golowich et al. (2020). As such we conjecture that a nested stochastic gradient descent ascent type algorithm is computationally optimal for our problem, i.e., has a lower gradient evaluation complexity.

To start, we first restate a lemma introduced by Bassily et al. Bassily et al. (2018), which characterizes the types of compositions that preserve the PL property of a function.

**Lemma 1** (Claim 1 of Bassily et al. Bassily et al. (2018)). *Let $f : \mathbb{R}^d \to \mathbb{R}$ be $\ell_{\nabla f}$-Lipschitz-smooth and $\mu_f$-PL function for some $\ell_{\nabla f}, \mu > 0$. Let $\boldsymbol{h} : \mathbb{R}^k \to \mathbb{R}^d$ be any differentiable and Lipschitz-continuous map, such that $d \geq k$. Let $\lambda_{\min}$ and $\lambda_{\max}$ be respectively the smallest and largest eigenvalues of $\nabla \boldsymbol{h}(\boldsymbol{w})^T \nabla \boldsymbol{h}(\boldsymbol{w})$. Then, there exists $a \leq \lambda_{\min} \nabla \boldsymbol{h}(\boldsymbol{w})^T \nabla \boldsymbol{h}(\boldsymbol{w})$ and $b \geq \lambda_{\max} \nabla \boldsymbol{h}(\boldsymbol{w})^T \nabla \boldsymbol{h}(\boldsymbol{w})$ such that the function $f(\boldsymbol{h}(\cdot)) : \mathbb{R}^k \to \mathbb{R}$ is $\ell_{\nabla f \circ \boldsymbol{h}}$-smooth and $\mu_{f \circ \boldsymbol{h}}$-PL. where $\ell_{\nabla f \circ \boldsymbol{h}} \geq b\ell_{\nabla f}$ and $\mu_{f \circ \boldsymbol{h}} \doteq a\mu_f$.*

We also recall the following lemma by Karimi et al. Karimi et al. (2016), which states the relationship between PL-functions and quadratically growing functions.

**Lemma 2** (Corollary of Theorem 2 Karimi et al. (2016)). *If a function $f$ is $\mu$-PL, then $f$ is $4\mu$-quadratically-growing.*

We prove our results under the following assumptions which are weaker than Assumption 1:

**Assumption 2.** *For any player $i \in [n]$ and $\mathcal{G} \in \text{supp}(\mathcal{D})$: 1. (Lipschitz-smoothness) their payoff $u_i^{\mathcal{G}}$ is $\ell_{\nabla_u}$-Lipschitz smooth, 2. (Strong concavity) their payoff $u_i^{\mathcal{G}}$ is $\mu_{u_i}$-strongly-concave in $\boldsymbol{a}_i$, and 3. (Lipschitz-smooth hypothesis classes) For all $\boldsymbol{h} \in \mathcal{H} \subset \mathcal{X}^{\Gamma \times \mathbb{R}^\omega}$, $\boldsymbol{f} \in \mathcal{F} \subset \mathcal{A}^{\Gamma \times \mathcal{X} \times \mathbb{R}^\omega}$, $\boldsymbol{h}(\mathcal{G}; \cdot), \boldsymbol{f}(\mathcal{G}; \cdot)$ are injective, i.e., $\omega \leq nm$, and Lipschitz-continuous differentiable functions.*

While Assumption 2 holds that the hypothesis class is Lipschitz-continuous and differentiable, in Assumption 1 we assume for ease of presentation that the hypothesis class is Lipschitz-smooth. Note

that Assumption 1 is stronger and does imply the above assumption since any Lipschitz-smooth function is Lipschitz-continuous over any compact subset of the parameter space.

The following proofs and lemma statements hold in expectation over all runs of the algorithm; for notational simplicity, we omit this expectation. We first bound the Euclidean squared distance between the gradient of the empirical cumulative regret and the empirical exploitability for the weight iterates computed by our algorithm, i.e., $\left\| \nabla \widehat{\varphi}(\boldsymbol{w}^{\boldsymbol{h},(t)}) - \nabla_{\boldsymbol{w}^{\boldsymbol{h}}} \widehat{\psi}\left(\boldsymbol{w}^{\boldsymbol{h},(t)}, \boldsymbol{w}^{\boldsymbol{f},(t)}\right) \right\|_2^2$, as a function of $T_{\boldsymbol{f}}$.

**Lemma 3** (Inner Loop Error Bound). *Suppose that Assumption 2 holds. If Algorithm 1 is run with inputs that satisfy $\forall t \in T_{\boldsymbol{h}}, s \in T_{\boldsymbol{f}}, \eta_{\boldsymbol{h}}^{(t)} > 0$ and $\eta_{\boldsymbol{f}}^{(s)} = \frac{2s+1}{\mu_{\widehat{\psi}}(s+1)^2}, T_{\boldsymbol{h}} \in \mathbb{N}_{++}$, and $T_{\boldsymbol{f}} \doteq \frac{\ell_{\nabla \widehat{\psi}}^3 \ell_{\widehat{\psi}}^2}{2\mu_{\widehat{\psi}} \varepsilon}$, where $\varepsilon > 0$. Then, the outputs $(\boldsymbol{w}^{\boldsymbol{h},(t)}, \boldsymbol{w}^{\boldsymbol{f},(t)})_{t=1}^{T_{\boldsymbol{h}}}$ satisfy:*

$$\left\| \nabla \widehat{\varphi}(\boldsymbol{w}^{\boldsymbol{h}}) - \nabla_{\boldsymbol{w}^{\boldsymbol{h}}} \widehat{\psi}\left(\boldsymbol{w}^{\boldsymbol{h},(t)}, \boldsymbol{w}^{\boldsymbol{f},(t)}\right) \right\|_2^2 \leq \varepsilon.$$

*Proof of Lemma 3.* Let $\boldsymbol{w}^{\boldsymbol{f}^*}(\boldsymbol{w}^{\boldsymbol{h}}) \in \arg\max_{\boldsymbol{w}^{\boldsymbol{f}} \in \mathbb{R}^\omega : \forall \mathcal{G} \in \Gamma, \boldsymbol{f}(\mathcal{G}; \boldsymbol{w}^{\boldsymbol{f}}) \in \mathcal{X}(\boldsymbol{h}(\mathcal{G}, \boldsymbol{w}^{\boldsymbol{h}}))} \widehat{\psi}\left(\boldsymbol{w}^{\boldsymbol{h}}, \boldsymbol{w}^{\boldsymbol{f}}\right)$. For all $\boldsymbol{w}^{\boldsymbol{h}} \in \mathbb{R}^\omega$, $\widehat{\psi}\left(\boldsymbol{w}^{\boldsymbol{h}}, \cdot\right)$ is the composition of $\mathbb{E}\left[\psi(\boldsymbol{a}, \cdot)\right]$ for all $\boldsymbol{a} \in \mathcal{X}$, a $\ell_{\nabla\psi}$-Lipschitz-smooth and $\mu_\psi$-strongly-concave function (Assumption 1), with an injective Lipschitz-continuous differentiable function, $\boldsymbol{f}(\mathcal{G}, \cdot)$, which means that for all $\boldsymbol{w}^{\boldsymbol{h}} \in \mathbb{R}^\omega$, $\widehat{\psi}\left(\boldsymbol{w}^{\boldsymbol{h}}, \cdot\right)$ is a $\mu_{\widehat{\psi}}$-PL function and $\ell_{\widehat{\psi}}$-Lipschitz-smooth function (Lemma 1), and the following convergence bound holds if $\eta_{\boldsymbol{f}}^{(t)} \doteq \frac{2t+1}{2\mu_{\widehat{\psi}}(t+1)^2}$ for the inner loop iterates as a corollary of of convergence results on stochastic gradient ascent for PL objectives (Theorem 4, Karimi et al. (2016)), we have:

$$\widehat{\varphi}(\boldsymbol{w}^{\boldsymbol{h},(t)}) - \widehat{\psi}\left(\boldsymbol{w}^{\boldsymbol{h},(t)}, \boldsymbol{w}^{\boldsymbol{f},(t)}\right) \leq \frac{\ell_{\nabla\widehat{\psi}} \ell_{\widehat{\psi}}^2}{2\mu_{\widehat{\psi}} T_{\boldsymbol{f}}} \tag{17}$$

Since for all $\boldsymbol{w}^{\boldsymbol{h}} \in \mathbb{R}^\omega$, is $\mu_{\widehat{\psi}}$-PL, by Lemma 2, we have:

$$\widehat{\varphi}(\boldsymbol{w}^{\boldsymbol{h},(t)}) - \widehat{\psi}\left(\boldsymbol{w}^{\boldsymbol{h},(t)}, \boldsymbol{w}^{\boldsymbol{f},(t)}\right) \geq 4\mu_{\widehat{\psi}} \left\| \boldsymbol{w}^{\boldsymbol{f}^*}(\boldsymbol{w}^{\boldsymbol{h},(t)}) - \boldsymbol{w}^{\boldsymbol{f},(t)} \right\|_2^2 , \tag{18}$$

Combining the two previous inequalities, we get:

$$4\mu_{\widehat{\psi}} \left\| \boldsymbol{w}^{\boldsymbol{f}^*}(\boldsymbol{w}^{\boldsymbol{h},(t)}) - \boldsymbol{w}^{\boldsymbol{f},(t)} \right\|_2^2 \leq \frac{\ell_{\nabla\widehat{\psi}} \ell_{\widehat{\psi}}^2}{2\mu_{\widehat{\psi}} T_{\boldsymbol{f}}} \tag{19}$$

$$\left\| \boldsymbol{w}^{\boldsymbol{f}^*}(\boldsymbol{w}^{\boldsymbol{h},(t)}) - \boldsymbol{w}^{\boldsymbol{f},(t)} \right\|_2^2 \leq \frac{\ell_{\nabla\widehat{\psi}} \ell_{\widehat{\psi}}^2}{8\mu_{\widehat{\psi}}^2 T_{\boldsymbol{f}}} \tag{20}$$

$$\left\| \boldsymbol{w}^{\boldsymbol{f}^*}(\boldsymbol{w}^{\boldsymbol{h},(t)}) - \boldsymbol{w}^{\boldsymbol{f},(t)} \right\|_2 \leq \frac{\ell_{\widehat{\psi}}}{2\mu_{\widehat{\psi}}} \sqrt{\frac{\ell_{\nabla\widehat{\psi}}}{2T_{\boldsymbol{f}}}} \tag{21}$$

Finally, we bound the error between the approximate gradient computed by Algorithm 1 and the true gradient $\nabla \widehat{\varphi}(\boldsymbol{w}^{\boldsymbol{h}})$ at each iteration $t \in \mathbb{N}_{++}$. Note that $\nabla \psi$ is Lipschitz-continuous in $(\boldsymbol{w}^{\boldsymbol{h}}, \boldsymbol{w}^{\boldsymbol{f}})$ since the composition of Lipschitz continuous functions is Lipschitz. Hence, $\nabla \widehat{\psi}$ is also Lipschitz

and we have:

$$= \left\| \nabla_{\boldsymbol{w^h}} \widehat{\varphi} \left( \boldsymbol{w^{h,(t)}} \right) - \nabla_{\boldsymbol{w^h}} \widehat{\psi} \left( \boldsymbol{w^{h,(t)}}, \boldsymbol{w^{f,(t)}} \right) \right\|_2 \tag{22}$$

$$\leq \left\| \nabla_{(\boldsymbol{w^h}, \boldsymbol{w^f})} \widehat{\psi} \left( \boldsymbol{w^{h,(t)}}, \boldsymbol{w^f}^*(\boldsymbol{w^h}) \right) - \nabla_{(\boldsymbol{w^h}, \boldsymbol{w^f})} \widehat{\psi} \left( \boldsymbol{w^{h,(t)}}, \boldsymbol{w^{f,(t)}} \right) \right\|_2 \tag{23}$$

$$\leq \ell_{\nabla \widehat{\psi}} \left\| (\boldsymbol{w^{h,(t)}}, \boldsymbol{w^f}^*(\boldsymbol{w^{h,(t)}})) - (\boldsymbol{w^{h,(t)}}, \boldsymbol{w^{f,(t)}}) \right\|_2 \tag{24}$$

$$\leq \ell_{\nabla \widehat{\psi}} \left( \left\| \boldsymbol{w^{h,(t)}} - \boldsymbol{w^{h,(t)}} \right\|_2 + \left\| \boldsymbol{w^f}^*(\boldsymbol{w^{h,(t)}}) - \boldsymbol{w^{f,(t)}} \right\|_2 \right) \tag{25}$$

$$= \ell_{\nabla \widehat{\psi}} \left\| \boldsymbol{w^f}^*(\boldsymbol{w^{h,(t)}}) - \boldsymbol{w^{f,(t)}} \right\|_2 \tag{26}$$

$$\leq \frac{\ell_{\nabla \widehat{\psi}}^{3/2} \ell_{\widehat{\psi}}}{2\mu_{\widehat{\psi}} \sqrt{2T_{\boldsymbol{f}}}} \qquad \qquad \text{(Equation (21))}$$

$$\tag{27}$$

Then, given $\varepsilon > 0$, for any number of inner loop iterations such that $T_{\boldsymbol{f}} \geq \frac{\ell_{\nabla \widehat{\psi}}^3 \ell_{\widehat{\psi}}^2}{2\mu_{\widehat{\psi}} \varepsilon}$, for all $t \in [T_{\boldsymbol{h}}]$, we have:

$$\left\| \nabla \widehat{\varphi}(\boldsymbol{w^h}) - \nabla_{\boldsymbol{w^h}} \widehat{\psi} \left( \boldsymbol{w^{h,(t)}}, \boldsymbol{w^{f,(t)}} \right) \right\|_2^2 \leq \varepsilon \tag{28}$$

$$\square$$

Using the above gradient error bound we derive a progress bound for the outer loop of our algorithm.

**Lemma 4** (Progress Lemma for Approximate Iterate). *Suppose that Assumption 2 holds. If Algorithm 1 is run with inputs satisfying* $\forall t \in [T_{\boldsymbol{h}}], s \in [T_{\boldsymbol{f}}], \eta_{\boldsymbol{h}}^{(t)} > 0$ *and* $\eta_{\boldsymbol{f}}^{(s)} \doteq \frac{2s+1}{2\mu_{\widehat{\psi}}(s+1)^2}$, $T_{\boldsymbol{h}} \in \mathbb{N}_{++}$, *and for* $T_{\boldsymbol{f}} \geq \frac{\ell_{\nabla \widehat{\psi}}^3 \ell_{\widehat{\psi}}^2}{2\mu_{\widehat{\psi}} \varepsilon}$, *where* $\varepsilon > 0$. *Then, the outputs* $(\boldsymbol{w^{h,(t)}}, \boldsymbol{w^{f,(t)}})_{t=1}^{T_{\boldsymbol{h}}}$ *satisfy:*

$$\widehat{\varphi}(\boldsymbol{w^{h,(t+1)}}) \leq \widehat{\varphi}(\boldsymbol{w^{h,(t)}}) - \eta_{\boldsymbol{h}}^{(t)} \left\| \nabla_{\boldsymbol{h}} \widehat{\varphi}(\boldsymbol{w^{h,(t)}}) \right\|_2^2 + \eta_{\boldsymbol{h}}^{(t)} \varepsilon + \frac{(\eta_{\boldsymbol{h}}^{(t)})^2 \ell_{\widehat{\psi}} \ell_{\nabla \widehat{\varphi}}^2}{2} \tag{29}$$

*Proof of Lemma 4.* Fix $t \in T_{\boldsymbol{h}}$. Define $\mathbf{err}^{(t)} \doteq \nabla \widehat{\varphi}(\boldsymbol{w^{h,(t)}}) - \nabla_{\boldsymbol{w^h}} \widehat{\psi}(\boldsymbol{w^{h,(t)}}, \boldsymbol{w^{f,(t)}})$. Since $\widehat{\psi}$ is Lipschitz-smooth, $\widehat{\varphi}$ is weakly-convex Lin et al. (2020), and as such we have:

$$\widehat{\varphi}(\boldsymbol{w^{h,(t+1)}}) \tag{30}$$

$$\leq \widehat{\varphi}(\boldsymbol{w^{h,(t)}}) + \left\langle \nabla_{\boldsymbol{h}} \widehat{\varphi}(\boldsymbol{w^{h,(t)}}), \boldsymbol{w^{h,(t+1)}} - \boldsymbol{w^{h,(t)}} \right\rangle + \ell_{\nabla \widehat{\varphi}}/2 \left\| \boldsymbol{w^{h,(t+1)}} - \boldsymbol{w^{h,(t)}} \right\|_2^2 \tag{31}$$

$$\leq \widehat{\varphi}(\boldsymbol{w^{h,(t)}}) + \left\langle \nabla \widehat{\varphi}(\boldsymbol{w^{h,(t)}}), -\eta_{\boldsymbol{h}}^{(t)} \nabla_{\boldsymbol{w^h}} 1/\mathcal{B}_{\boldsymbol{h}}^{(t)} \sum_{\mathcal{G} \in \mathcal{B}_{\boldsymbol{h}}^{(t)}} \psi_{\mathcal{G}}(\boldsymbol{w^{h,(t)}}, \boldsymbol{w^{f,(t)}}) \right\rangle$$

$$+ \ell_{\nabla \widehat{\varphi}}/2 \left\| \eta_{\boldsymbol{h}}^{(t)} \nabla_{\boldsymbol{w^h}} 1/\mathcal{B}_{\boldsymbol{h}}^{(t)} \sum_{\mathcal{G} \in \mathcal{B}_{\boldsymbol{h}}^{(t)}} \psi_{\mathcal{G}}(\boldsymbol{w^{h,(t)}}, \boldsymbol{w^{f,(t)}}) \right\|_2^2 \tag{32}$$

$$\leq \widehat{\varphi}(\boldsymbol{w^{h,(t)}}) - \eta_{\boldsymbol{h}}^{(t)} \left\langle \nabla \widehat{\varphi}(\boldsymbol{w^{h,(t)}}), \nabla_{\boldsymbol{w^h}} 1/\mathcal{B}_{\boldsymbol{h}}^{(t)} \sum_{\mathcal{G} \in \mathcal{B}_{\boldsymbol{h}}^{(t)}} \psi_{\mathcal{G}}(\boldsymbol{w^{h,(t)}}, \boldsymbol{w^{f,(t)}}) \right\rangle + \frac{(\eta_{\boldsymbol{h}}^{(t)})^2 \ell_{\widehat{\psi}} \ell_{\nabla \widehat{\varphi}}^2}{2} \tag{33}$$

where the last line follows from $\psi_{\mathcal{G}}$ being $\ell_{\widehat{\psi}}$-Lipschitz continuous. Taking the expectation w.r.t $\mathcal{B}_{\boldsymbol{h}}^{(t)}$ conditioned on $(\boldsymbol{w^{h,(t)}}, \boldsymbol{w^{f,(t)}})$, we get:

$$\leq \widehat{\varphi}(\boldsymbol{w^{h,(t)}}) - \eta_{\boldsymbol{h}}^{(t)} \left\| \nabla_{\boldsymbol{w^h}} \widehat{\psi}(\boldsymbol{w^{h,(t)}}, \boldsymbol{w^{f,(t)}}) \right\|_2^2 + \frac{(\eta_{\boldsymbol{h}}^{(t)})^2 \ell_{\widehat{\psi}} \ell_{\nabla \widehat{\varphi}}^2}{2} \tag{34}$$

$$= \widehat{\varphi}(\boldsymbol{w^{h,(t)}}) - \eta_{\boldsymbol{h}}^{(t)} \left\| \nabla_{\boldsymbol{h}} \widehat{\varphi}(\boldsymbol{w^{h,(t)}}) - \nabla_{\boldsymbol{h}} \widehat{\varphi}(\boldsymbol{w^{h,(t)}}) + \nabla_{\boldsymbol{w^h}} \widehat{\psi}(\boldsymbol{w^{h,(t)}}, \boldsymbol{w^{f,(t)}}) \right\|_2^2 + \frac{(\eta_{\boldsymbol{h}}^{(t)})^2 \ell_{\widehat{\psi}} \ell_{\nabla \widehat{\varphi}}^2}{2} \tag{35}$$

$$\leq \widehat{\varphi}(\boldsymbol{w^{h,(t)}}) - \eta_{\boldsymbol{h}}^{(t)} \left\| \nabla_{\boldsymbol{h}} \widehat{\varphi}(\boldsymbol{w^{h,(t)}}) \right\|_2^2 + \eta_{\boldsymbol{h}}^{(t)} \left\| \nabla_{\boldsymbol{h}} \widehat{\varphi}(\boldsymbol{w^{h,(t)}}) - \nabla_{\boldsymbol{w^h}} \widehat{\psi}(\boldsymbol{w^{h,(t)}}, \boldsymbol{w^{f,(t)}}) \right\|_2^2 + \frac{(\eta_{\boldsymbol{h}}^{(t)})^2 \ell_{\widehat{\psi}} \ell_{\nabla \widehat{\varphi}}^2}{2} \tag{36}$$

$$\leq \widehat{\varphi}(\boldsymbol{w^{h,(t)}}) - \eta_{\boldsymbol{h}}^{(t)} \left\| \nabla_{\boldsymbol{a}} \widehat{\varphi}(\boldsymbol{w^{h,(t)}}) \right\|_2^2 + \eta_{\boldsymbol{h}}^{(t)} (\mathbf{err}^{(t)})^2 + \frac{(\eta_{\boldsymbol{h}}^{(t)})^2 \ell_{\widehat{\psi}} \ell_{\nabla \widehat{\varphi}}^2}{2} \tag{37}$$

By Lemma 3, we then have $(\mathbf{err}^{(t)})^2 \leq \varepsilon$, which gives us for all $t \in T_{\boldsymbol{h}}$:

$$\leq \widehat{\varphi}(\boldsymbol{w^{h,(t)}}) - \eta_{\boldsymbol{h}}^{(t)} \left\| \nabla_{\boldsymbol{a}} \widehat{\varphi}(\boldsymbol{w^{h,(t)}}) \right\|_2^2 + \eta_{\boldsymbol{h}}^{(t)} \varepsilon + \frac{(\eta_{\boldsymbol{h}}^{(t)})^2 \ell_{\widehat{\psi}} \ell_{\nabla \widehat{\varphi}}^2}{2} \tag{38}$$

$\square$

Finally, telescoping the the inequality given in the above lemma we can obtain our convergence result.

**Theorem 4.1** (Convergence to Stationary Point). *Suppose that Assumption 1 holds. Let $\varepsilon > 0$. If Algorithm 1 is run with inputs satisfying $\eta_{\boldsymbol{h}}^{(t)} \in \Theta(1/\sqrt{t})$, $\eta_{\boldsymbol{f}}^{(s)} \in \Theta(1/s)$, $T_{\boldsymbol{h}} \in \mathbb{N}_{++}$, and $T_{\boldsymbol{f}} \in O(1/\varepsilon)$, for all $t \in [T_{\boldsymbol{h}}]$, $s \in [T_{\boldsymbol{f}}]$. Then, the outputs $(\boldsymbol{w^{h,(t)}}, \boldsymbol{w^{f,(t)}})_{t=0}^{T_{\boldsymbol{h}}}$ satisfy*
$$\mathbb{E}\left[ \min_{t=0,\dots,T_{\boldsymbol{h}}-1} \left\| \nabla_{\boldsymbol{a}} \widehat{\varphi}(\boldsymbol{w^{h,(t)}}) \right\|_2^2 \right] \in O\left( \frac{\log(T_{\boldsymbol{h}})}{\sqrt{T_{\boldsymbol{h}}}} + \varepsilon \right).$$

*Proof of Theorem 4.1.* By Lemma 4, we have:

$$\widehat{\varphi}(\boldsymbol{w^{h,(t+1)}}) \leq \widehat{\varphi}(\boldsymbol{w^{h,(t)}}) - \eta_{\boldsymbol{h}}^{(t)} \left\| \nabla_{\boldsymbol{a}} \widehat{\varphi}(\boldsymbol{w^{h,(t)}}) \right\|_2^2 + \eta_{\boldsymbol{h}}^{(t)} \varepsilon + \frac{(\eta_{\boldsymbol{h}}^{(t)})^2 \ell_{\widehat{\psi}} \ell_{\nabla \widehat{\varphi}}^2}{2} \tag{39}$$

Summing up the inequalities for $t = 0, \dots, T_{\boldsymbol{h}} - 1$:

$$\sum_{t=1}^{T_{\boldsymbol{h}}} \eta_{\boldsymbol{h}}^{(t)} \left\| \nabla_{\boldsymbol{a}} \widehat{\varphi}(\boldsymbol{w^{h,(t)}}) \right\|_2^2 \leq \widehat{\varphi}(\boldsymbol{w^{h,(0)}}) - \widehat{\varphi}(\boldsymbol{w^{h,(T_{\boldsymbol{h}})}}) + \sum_{t=1}^{T_{\boldsymbol{h}}} \eta_{\boldsymbol{h}}^{(t)} \varepsilon + \sum_{t=1}^{T_{\boldsymbol{h}}} (\eta_{\boldsymbol{h}}^{(t)})^2 \frac{\ell_{\widehat{\psi}} \ell_{\nabla \widehat{\varphi}}^2}{2} \tag{40}$$

Taking the minimum of $\left\| \nabla_{\boldsymbol{a}} \widehat{\varphi}(\boldsymbol{w^{h,(t)}}) \right\|_2^2$ across all $t \in [T_{\boldsymbol{h}}]$, to obtain:

$$\left( \min_{t=0,\dots,T_{\boldsymbol{h}}-1} \left\| \nabla_{\boldsymbol{a}} \widehat{\varphi}(\boldsymbol{w^{h,(t)}}) \right\|_2^2 \right) \sum_{t=1}^{T_{\boldsymbol{h}}} \eta_{\boldsymbol{h}}^{(t)} \leq \widehat{\varphi}(\boldsymbol{w^{h,(0)}}) - \widehat{\varphi}(\boldsymbol{w^{h,(T_{\boldsymbol{h}})}}) + \sum_{t=1}^{T_{\boldsymbol{h}}} \eta_{\boldsymbol{h}}^{(t)} \varepsilon + \sum_{t=1}^{T_{\boldsymbol{h}}} (\eta_{\boldsymbol{h}}^{(t)})^2 \frac{\ell_{\widehat{\psi}} \ell_{\nabla \widehat{\varphi}}^2}{2} \tag{41}$$

$$\left( \min_{t=0,\dots,T_{\boldsymbol{h}}-1} \left\| \nabla_{\boldsymbol{a}} \widehat{\varphi}(\boldsymbol{w^{h,(t)}}) \right\|_2^2 \right) \leq \frac{\widehat{\varphi}(\boldsymbol{w^{h,(0)}}) - \widehat{\varphi}(\boldsymbol{w^{h,(T_{\boldsymbol{h}})}}) + \sum_{t=1}^{T_{\boldsymbol{h}}} \eta_{\boldsymbol{h}}^{(t)} \varepsilon + \sum_{t=1}^{T_{\boldsymbol{h}}} (\eta_{\boldsymbol{h}}^{(t)})^2 \frac{\ell_{\widehat{\psi}} \ell_{\nabla \widehat{\varphi}}^2}{2}}{\sum_{t=1}^{T_{\boldsymbol{h}}} \eta_{\boldsymbol{h}}^{(t)}} \tag{42}$$

$$\left( \min_{t=0,\dots,T_{\boldsymbol{h}}-1} \left\| \nabla_{\boldsymbol{a}} \widehat{\varphi}(\boldsymbol{w^{h,(t)}}) \right\|_2^2 \right) \leq \frac{\widehat{\varphi}(\boldsymbol{w^{h,(0)}}) - \widehat{\varphi}(\boldsymbol{w^{h,(T_{\boldsymbol{h}})}})}{\sum_{t=1}^{T_{\boldsymbol{h}}} \eta_{\boldsymbol{h}}^{(t)}} + \frac{\ell_{\widehat{\psi}} \ell_{\nabla \widehat{\varphi}}^2}{2} \frac{\sum_{t=1}^{T_{\boldsymbol{h}}} (\eta_{\boldsymbol{h}}^{(t)})^2}{\sum_{t=1}^{T_{\boldsymbol{h}}} \eta_{\boldsymbol{h}}^{(t)}} + \varepsilon \tag{43}$$

Suppose that $\eta_{\boldsymbol{h}}^{(t)} = \frac{1}{\sqrt{t}}$, we then have:

$$\left( \min_{t=0,\ldots,T_{\boldsymbol{h}}-1} \left\| \nabla_{\boldsymbol{a}} \widehat{\varphi}(\boldsymbol{w}^{\boldsymbol{h},(t)}) \right\|_2^2 \right) \in O\left( \frac{\log(T_{\boldsymbol{h}})}{\sqrt{T_{\boldsymbol{h}}}} + \varepsilon \right) \tag{44}$$

Taking an expectation over all runs of the algorithms and using Jensen's inequality, we obtain the result. $\qquad\square$

This result characterizes the generalization capacity of the generator and discriminator as a function of the covering number of the hypothesis classes. The proof relies on the following lemma, which states that the distance of any generator and discriminator from the hypothesis class to their $r$-cover is bounded in payoff space.

**Lemma 5** (Bounded Expected Cumulative Regret). *Suppose that part 1. of Assumption 1 holds. For any hypothesis classes $\mathcal{H} \subset \mathcal{X}^\Gamma$, $\mathcal{F} \subset \mathcal{A}^{\mathcal{X} \times \Gamma}$ and hypotheses $\boldsymbol{h} \in \mathcal{H}$ and $\boldsymbol{f} \in \mathcal{F}$, any pseudo-game distribution $\mathcal{D} \in \Delta(\Gamma)$:*

$$\left| \mathbb{E}_{\mathcal{G} \sim \mathcal{D}} \left[ \psi^{\mathcal{G}}(\boldsymbol{h}^r(\mathcal{G}), \boldsymbol{f}^r(\mathcal{G}, \boldsymbol{h}^r(\mathcal{G}))) \right] - \mathbb{E}_{\mathcal{G} \sim \mathcal{D}} \left[ \psi^{\mathcal{G}}(\boldsymbol{h}(\mathcal{G}), \boldsymbol{f}(\mathcal{G}, \boldsymbol{h}(\mathcal{G}))) \right] \right| \leq 2\ell_\psi r \tag{45}$$

*Proof of Lemma 5.* Since part 1. of Assumption 1 holds, we have that $\psi^{\mathcal{G}}$ is $\ell_{\psi^{\mathcal{G}}}$-Lipschitz continuous with $\ell_{\psi^{\mathcal{G}}} = \max_{(\boldsymbol{a},\boldsymbol{b}) \in \mathcal{A} \times \mathcal{A}} \left| \nabla \psi^{\mathcal{G}}(\boldsymbol{a}, \boldsymbol{b}) \right|$ and let $\ell_\psi = \max_{\mathcal{G} \in \Gamma} \max_{(\boldsymbol{a},\boldsymbol{b}) \in \mathcal{A} \times \mathcal{A}} \left| \nabla \psi^{\mathcal{G}}(\boldsymbol{a}, \boldsymbol{b}) \right|$.

$$\left| \mathbb{E}_{\mathcal{G} \sim \mathcal{D}} \left[ \psi^{\mathcal{G}}(\boldsymbol{h}^r(\mathcal{G}), \boldsymbol{f}^r(\mathcal{G}, \boldsymbol{h}^r(\mathcal{G}))) \right] - \mathbb{E}_{\mathcal{G} \sim \mathcal{D}} \left[ \psi^{\mathcal{G}}(\boldsymbol{h}(\mathcal{G}), \boldsymbol{f}(\mathcal{G}, \boldsymbol{h}(\mathcal{G}))) \right] \right| \tag{46}$$

$$= \left| \mathbb{E}_{\mathcal{G} \sim \mathcal{D}} \left[ \psi^{\mathcal{G}}(\boldsymbol{h}^r(\mathcal{G}), \boldsymbol{f}^r(\mathcal{G}, \boldsymbol{h}^r(\mathcal{G}))) - \psi^{\mathcal{G}}(\boldsymbol{h}(\mathcal{G}), \boldsymbol{f}(\mathcal{G}, \boldsymbol{h}(\mathcal{G}))) \right] \right| \tag{47}$$

$$\leq \max_{\mathcal{G} \in \Gamma} \left| \psi^{\mathcal{G}}(\boldsymbol{h}^r(\mathcal{G}), \boldsymbol{f}^r(\mathcal{G}, \boldsymbol{h}^r(\mathcal{G}))) - \psi^{\mathcal{G}}(\boldsymbol{h}(\mathcal{G}), \boldsymbol{f}(\mathcal{G}, \boldsymbol{h}(\mathcal{G}))) \right| \tag{48}$$

$$\leq \max_{\mathcal{G} \in \Gamma} \ell_{\psi^{\mathcal{G}}} \left\| (\boldsymbol{h}^r(\mathcal{G}), \boldsymbol{f}^r(\mathcal{G}, \boldsymbol{h}^r(\mathcal{G}))) - (\boldsymbol{h}(\mathcal{G}), \boldsymbol{f}(\mathcal{G}, \boldsymbol{h}(\mathcal{G}))) \right\| \tag{49}$$

$$= \ell_\psi \left\| (\boldsymbol{h}^r(\mathcal{G}), \boldsymbol{f}^r(\mathcal{G}, \boldsymbol{h}^r(\mathcal{G}))) - (\boldsymbol{h}(\mathcal{G}), \boldsymbol{f}(\mathcal{G}, \boldsymbol{h}(\mathcal{G}))) \right\| \tag{50}$$

$$= \ell_\psi \left( \left\| \boldsymbol{h}^r(\mathcal{G}) - \boldsymbol{h}(\mathcal{G}) \right\| + \left\| \boldsymbol{f}^r(\mathcal{G}, \boldsymbol{h}^r(\mathcal{G})) - \boldsymbol{f}(\mathcal{G}, \boldsymbol{h}(\mathcal{G})) \right\| \right) \tag{51}$$

$$\leq 2\ell_\psi r \tag{52}$$

$$\qquad\square$$

**Theorem 4.2** (Sample Complexity of Expected Cumulative Regret). *Suppose that part 1. of Assumption 1 holds. Let $\varepsilon, \delta \in (0,1)$, $r \in O\left( \varepsilon/\ell_\psi \right)$ and consider hypothesis classes $\mathcal{H}, \mathcal{F}$ given by the minimum $r$-covering sets of $\mathcal{X}^\Gamma$, $\mathcal{A}^{\mathcal{X} \times \Gamma}$ respectively. For any $\boldsymbol{h} \in \mathcal{X}^\Gamma$ and $\boldsymbol{f} \in \mathcal{A}^{\mathcal{X} \times \Gamma}$, take the closest hypotheses $\boldsymbol{h}^r \in \arg\min_{\boldsymbol{h}' \in \mathcal{H}} \|\boldsymbol{h} - \boldsymbol{h}'\|$ and $\boldsymbol{f}^r \in \arg\min_{\boldsymbol{f}' \in \mathcal{F}} \|\boldsymbol{f} - \boldsymbol{f}'\|$ to respectively. Then, for any pseudo-game distribution $\mathcal{D} \in \Delta(\Gamma)$ with compact support, with probability at least $1 - \delta$ over draws of the training set $\mathcal{S} \sim \mathcal{D}^k$ with $k \in \Omega\left( 1/\varepsilon^2 \log\left( \delta^{-1} \rho(\mathcal{H}, \varepsilon/\ell_\psi) \rho(\mathcal{F}, \varepsilon/\ell_\psi) \right) \right)$:*

$$\left| \mathbb{E}_{\mathcal{G} \sim \mathrm{unif}(\mathcal{S})} \left[ \psi^{\mathcal{G}}(\boldsymbol{h}^r(\mathcal{G}), \boldsymbol{f}^r(\mathcal{G}, \boldsymbol{h}^r(\mathcal{G}))) \right] - \mathbb{E}_{\mathcal{G} \sim \mathcal{D}} \left[ \psi^{\mathcal{G}}(\boldsymbol{h}(\mathcal{G}), \boldsymbol{f}(\mathcal{G}, \boldsymbol{h}(\mathcal{G}))) \right] \right| \leq \varepsilon \ .$$

*Proof of Theorem 4.2.* Since part 1. of Assumption 1 holds, we have that $\psi^{\mathcal{G}}$ is $\ell_{\psi^{\mathcal{G}}}$-Lipschitz continuous with $\ell_{\psi^{\mathcal{G}}} = \max_{(\boldsymbol{a},\boldsymbol{b}) \in \mathcal{A} \times \mathcal{A}} \left| \nabla \psi^{\mathcal{G}}(\boldsymbol{a}, \boldsymbol{b}) \right|$. Let $\ell_\psi = \max_{\mathcal{G} \in \Gamma} \max_{(\boldsymbol{a},\boldsymbol{b}) \in \mathcal{A} \times \mathcal{A}} \left| \nabla \psi^{\mathcal{G}}(\boldsymbol{a}, \boldsymbol{b}) \right|$.

$$\mathbb{P}_{\mathcal{S} \sim \mathcal{D}^k} \left[ \exists (\boldsymbol{h}, \boldsymbol{f}) \in \mathcal{X}^\Gamma \times \mathcal{A}^{\Gamma \times \mathcal{X}}, \left| \mathbb{E}_{\mathcal{G} \sim \mathrm{unif}(\mathcal{S})} \left[ \psi^{\mathcal{G}}(\boldsymbol{h}^r(\mathcal{G}), \boldsymbol{f}^r(\mathcal{G}, \boldsymbol{h}^r(\mathcal{G}))) \right] - \mathbb{E}_{\mathcal{G} \sim \mathcal{D}} \left[ \psi^{\mathcal{G}}(\boldsymbol{h}(\mathcal{G}), \boldsymbol{f}(\mathcal{G}, \boldsymbol{h}(\mathcal{G}))) \right] \right| \geq \varepsilon \right]$$

$$\tag{53}$$

$$\leq \mathbb{P}_{\mathcal{S} \sim \mathcal{D}^k} \left[ \exists (\boldsymbol{h}, \boldsymbol{f}) \in \mathcal{X}^\Gamma \times \mathcal{A}^{\Gamma \times \mathcal{X}}, \underbrace{\left| \mathbb{E}_{\mathcal{G} \sim \mathrm{unif}(\mathcal{S})} \left[ \psi^{\mathcal{G}}(\boldsymbol{h}^r(\mathcal{G}), \boldsymbol{f}^r(\mathcal{G}, \boldsymbol{h}^r(\mathcal{G}))) \right] - \mathbb{E}_{\mathcal{G} \sim \mathcal{D}} \left[ \psi^{\mathcal{G}}(\boldsymbol{h}^r(\mathcal{G}), \boldsymbol{f}^r(\mathcal{G}, \boldsymbol{h}^r(\mathcal{G}))) \right] \right|}_{\text{Variance: Error due to sample size}} \right.$$

$$\left. + \underbrace{\left| \mathbb{E}_{\mathcal{G} \sim \mathcal{D}} \left[ \psi^{\mathcal{G}}(\boldsymbol{h}^r(\mathcal{G}), \boldsymbol{f}^r(\mathcal{G}, \boldsymbol{h}^r(\mathcal{G}))) \right] - \mathbb{E}_{\mathcal{G} \sim \mathcal{D}} \left[ \psi^{\mathcal{G}}(\boldsymbol{h}(\mathcal{G}), \boldsymbol{f}(\mathcal{G}, \boldsymbol{h}(\mathcal{G}))) \right] \right|}_{\text{Bias: Hypothesis class approximation error}} \geq \varepsilon \right] \tag{54}$$

Let $r = \frac{\varepsilon}{6\ell_\psi}$, then using Lemma 5, we then have:

$$\leq \mathbb{P}_{\mathcal{S}\sim\mathcal{D}^k}\left[\exists(\boldsymbol{h},\boldsymbol{f})\in\mathcal{X}^\Gamma\times\mathcal{A}^{\Gamma\times\mathcal{X}}, \left|\mathbb{E}_{\mathcal{G}\sim\mathrm{unif}(\mathcal{S})}\left[\psi^\mathcal{G}(\boldsymbol{h}^r(\mathcal{G}),\boldsymbol{f}^r(\mathcal{G},\boldsymbol{h}^r(\mathcal{G})))\right]-\mathbb{E}_{\mathcal{G}\sim\mathcal{D}}\left[\psi^\mathcal{G}(\boldsymbol{h}^r(\mathcal{G}),\boldsymbol{f}^r(\mathcal{G},\boldsymbol{h}^r(\mathcal{G})))\right]\right|+2\ell_\psi r\geq\varepsilon\right]$$
(55)

$$= \mathbb{P}_{\mathcal{S}\sim\mathcal{D}^k}\left[\exists(\boldsymbol{h}^r,\boldsymbol{f}^r)\in\mathcal{H}\times\mathcal{F}, \left|\mathbb{E}_{\mathcal{G}\sim\mathrm{unif}(\mathcal{S})}\left[\psi^\mathcal{G}(\boldsymbol{h}^r(\mathcal{G}),\boldsymbol{f}^r(\mathcal{G},\boldsymbol{h}^r(\mathcal{G})))\right]-\mathbb{E}_{\mathcal{G}\sim\mathcal{D}}\left[\psi^\mathcal{G}(\boldsymbol{h}^r(\mathcal{G}),\boldsymbol{f}^r(\mathcal{G},\boldsymbol{h}^r(\mathcal{G})))\right]\right|\geq\frac{2\varepsilon}{3}\right]$$
(56)

where the penultimate line follow from the Lipschitz continuity of the Nash approximation error, and the final line from $r = \frac{\varepsilon}{6\ell_\psi}$. Applying a union bound we then get:

$$\mathbb{P}_{\mathcal{S}\sim\mathcal{D}^k}\left[\exists(\boldsymbol{h}^r,\boldsymbol{f}^r)\in\mathcal{H}\times\mathcal{F}, \left|\mathbb{E}_{\mathcal{G}\sim\mathrm{unif}(\mathcal{S})}\left[\psi^\mathcal{G}(\boldsymbol{h}^r(\mathcal{G}),\boldsymbol{f}^r(\mathcal{G},\boldsymbol{h}^r(\mathcal{G})))\right]-\mathbb{E}_{\mathcal{G}\sim\mathcal{D}}\left[\psi^\mathcal{G}(\boldsymbol{h}^r(\mathcal{G}),\boldsymbol{f}^r(\mathcal{G},\boldsymbol{h}^r(\mathcal{G})))\right]\right|\geq\frac{2\varepsilon}{3}\right]$$
(57)

$$\leq \rho(\mathcal{H},\varepsilon/6\ell_\psi)\rho(\mathcal{F},\varepsilon/6\ell_\psi)\mathbb{P}_{\mathcal{S}\sim\mathcal{D}^k}\left[\left|\mathbb{E}_{\mathcal{G}\sim\mathrm{unif}(\mathcal{S})}\left[\psi^\mathcal{G}(\boldsymbol{h}^r(\mathcal{G}),\boldsymbol{f}^r(\mathcal{G},\boldsymbol{h}^r(\mathcal{G})))\right]-\mathbb{E}_{\mathcal{G}\sim\mathcal{D}}\left[\psi^\mathcal{G}(\boldsymbol{h}^r(\mathcal{G}),\boldsymbol{f}^r(\mathcal{G},\boldsymbol{h}^r(\mathcal{G})))\right]\right|\geq\frac{2\varepsilon}{3}\right]$$
(58)

Note that for all $\mathcal{G}\in\Gamma$, $\psi_\mathcal{G}$ is bounded above by $\psi_{\max}=\max_{\mathcal{G}\in\mathrm{supp}(\mathcal{D})}\max_{(\boldsymbol{a},\boldsymbol{b})\in\mathcal{A}\times\mathcal{A}}\psi^\mathcal{G}(\boldsymbol{a},\boldsymbol{b})$ and bounded below by $\psi_{\min}=\min_{\mathcal{G}\in\mathrm{supp}(\mathcal{D})}\min_{(\boldsymbol{a},\boldsymbol{b})\in\mathcal{A}\times\mathcal{A}}\psi^\mathcal{G}(\boldsymbol{a},\boldsymbol{b})$ where $\psi_{\max}$ and $\psi_{\min}$ are well-defined since $\mathcal{A}$ and $\mathrm{supp}(\mathcal{D})$ are compact and $\psi^\mathcal{G}$ is continuous. Hence, by Hoeffding's inequality, we have:

$$\leq 2\rho(\mathcal{H},\varepsilon/6\ell_\psi)\rho(\mathcal{F},\varepsilon/6\ell_\psi)\exp\left\{-2k\left(\frac{2\varepsilon}{3(\psi_{\max}-\psi_{\min})}\right)^2\right\}$$
(59)

Re-organizing expressions, in order to get that:

$$\mathbb{P}_{\mathcal{S}\sim\mathcal{D}^k}\left[\exists(\boldsymbol{h},\boldsymbol{f})\in\mathcal{X}^\Gamma\times\mathcal{A}^{\Gamma\times\mathcal{X}}, \left|\mathbb{E}_{\mathcal{G}\sim\mathrm{unif}(\mathcal{S})}\left[\psi^\mathcal{G}(\boldsymbol{h}^r(\mathcal{G}),\boldsymbol{f}^r(\mathcal{G},\boldsymbol{h}^r(\mathcal{G})))\right]-\mathbb{E}_{\mathcal{G}\sim\mathcal{D}}\left[\psi^\mathcal{G}(\boldsymbol{h}(\mathcal{G}),\boldsymbol{f}(\mathcal{G},\boldsymbol{h}(\mathcal{G})))\right]\right|\geq\varepsilon\right]=\delta$$

we obtain that the sample size $k$ should be set as follows to obtain the result:

$$k\geq\frac{9(\psi_{\max}-\psi_{\min})^2}{8\varepsilon^2}\left[\log\left(\frac{2}{\delta}\right)+\log\left(\rho(\mathcal{H},\varepsilon/6\ell_\psi)\right)+\log\left(\rho(\mathcal{F},\varepsilon/6\ell_\psi)\right)\right]$$
(60)

$\square$

## G EXPERIMENTS

We run three sets of experiments in which we train GAES in three different pseudo-game settings. All experiments are run with 5 randomly selected different seeds ($\{5, 10, 25, 30, 42\}$), with hyperparameter selection being done over all 5 seeds. Unless otherwise mentioned, all results correspond to an average over these 5 seeds, with confidence intervals reported across these seeds as appropriate. In all of our experiments, we adopt the update rule in ADAM for the gradient step, making use of the ADAM implementation in the OPTAX library. We use JaxOPT's projected gradient method to compute best-responses and thus the exploitability of an action profile when a closed form is not available for the best-response. For all of the networks used in our experiments, if BatchNorm is used, it is applied before the activation layer. We describe whether if BatchNorm is used in the architecture of each network individually in the following sections.

**Computational Resources.** Our normal-form game experiments were run on GPUs while our other experiments were run on CPUs.

**Programming Languages, Packages, and Licensing.** We ran our experiments in Python 3.7 Van Rossum & Drake Jr (1995), using NumPy Harris et al. (2020), Jax Bradbury et al. (2018), OPTAX Bradbury et al. (2018), Haiku Hennigan et al. (2020), JaxOPT Blondel et al. (2021), and pycdd

Troffaes (2020). All figures were graphed using Matplotlib Hunter (2007) and Seaborn Waskom (2021).

Numpy and Seaborn are distributed under a liberal BSD license. Matplotlib only uses BSD compatible code, and its license is based on the PSF license. CVXPY is licensed under an APACHE license. Jax and Haiku are licensed under the Apache 2.0 License. Pycdd is distributed under a GNU license.

### G.1 NORMAL-FORM GAMES

Our first set of experiments aims to explore the impact of the accuracy of the discriminator on the performance of the generator. To this end, we consider normal-form games in which there exists a closed form solution for the discriminator. We observe that with an accurate enough discriminator, GAES achieves a performance similar to the neural architecture proposed by Duan et al. (2021a) when using the same equilibrium mapping architecture for the generator.

In our experiments, we replicate the setup proposed by Duan et al. (2021a), and we try to solve five games from the GAMUT library, namely, Traveller's Dilemma, Bertrand Oligopoly, Grab the Dollar, War of Attrition, and Majority Voting. We give a description of each game, as presented by Duan et al. (2021a):

- *Traveler's dilemma*: Each player simultaneously requests an amount of money and receives the lowest of the requests submitted by all players.

- *Grab the dollar*: A price is up for grabs, and both players have to decide when to grab the price. The action of each player is the chosen times. If both players grab it simultaneously, they will rip the price and receive a low payoff. If one chooses a time earlier than the other, they will receive the high payoff, and the opposing player will receive a payoff between the high and the low.

- *War of attrition*: In this game, both players compete for a single object, and each chooses a time to concede the object to the other player. If both concede at the same time, they share the object. Each player has a valuation of the object, and each player's utility is decremented at every time step.

- *Bertrand oligopoly*: All players in this game are producing the same item and are expected to set a price at which to sell the item. The player with the lowest price gets all the demand for the item and produces enough items to meet the demand to obtain the corresponding payoff.

- *Majority Voting*: This is an $n$-player symmetric game. All players vote for one of the other players. Players' utilities for each candidate being declared the winner are arbitrary. If there is a tie, the winner is the candidate with the lowest number. There may be multiple Nash equilibria in this game.

**Game Generation.** We use the *GAMUT library* Nudelman et al. (2004), which is a normal-form game generation library designated for testing game-theoretic algorithms, to generate a data set of games. Following Duan et al. (2021a), the games were generated so that payoffs were normalized between $[0, 1]$, with all other parameters drawn randomly. We generate 2000 games with 2 players and 300 actions for both players and for each game category, setting aside 200 for validation and 100 for testing.

**Network Architecture.** In all our experiments, we use a generator for GAES that has the same architecture as the equilibrium solver proposed by Duan et al. (2021a). Namely, we use a neural network with 4 hidden layers each with 1024 nodes and ReLU activations. The final layers of the generator have the same dimension as an action profile and each action in the profile is passed through a player-wise softmax activation. We augment the entire network with BatchNorm layers with non-trainable parameters after each activation layer. The total number of parameters for this generator is 20,855.

To explore how the accuracy of GAES's generator degrades as one uses more and more approximate discriminators, we consider 4 types of discriminators: a true discriminator that takes as input every player's expected payoffs for all actions and outputs the action with the highest payoff (this discriminator recovers exactly the Duan et al. (2021a) network architecture); a softmax discriminator with

a scaling parameter of 100, i.e., $\text{softmax}(\boldsymbol{u}) = \frac{e^{100u_i}}{\sum_i e^{100u_i}}$, that takes as input the expected payoffs of each player at the given equilibrium actions predicted by the generator and outputs its softmax; a less precise softmax discriminator with a scaling parameter of 10; and, a neural network with one linear layer with 1024 nodes and a softmax activation layer with scaling 1 (the total number of parameters for this discriminator is 7,115).

**Training and Hyperparameters.** We run our algorithm with no inner loop iterations and 10,000 outer loop iterations for the non-neural network discriminators, since they require no training. We adopt for the gradient step in our algorithm the ADAM algorithm. We use a learning rate of 0.001 for the optimizer step on each of the generator and discriminator (when appropriate), and use the default settings for the other hyperparameters of ADAM as given in the OPTAX implementation. We process the training data in batches of size 50.

**Experimental Results.** In Figure 5 (train) and Figure 6 (test), we observe that as the quality of the discriminator becomes more approximate the quality of the generator degrades significantly in certain games, underscoring the importance of designing optimal discriminators for GAES in certain settings. Perhaps more interestingly, we observe that GAES with a linear layer discriminator has a hard time in the Bertrand oligopoly and Traveller's Dilemma games. The reason for this is that the discriminator has a hard time learning a pure strategy best-response action, and this seems crucial for our training algorithm to find an optimal generator in these two games. This is further justified by GAES's near perfect performance when coupled with the true best-response discriminator. In contrast, for other games the neural network's performance suggests that an approximate best-response is enough for our training algorithm to find the optimal discriminator. Future work can investigate the relationship between game classes, and the precision-level w.r.t. the discriminator that is required for our training algorithms to perform well. We note that *normalized exploitability* is given as the exploitability divided by the average exploitability over the action space.

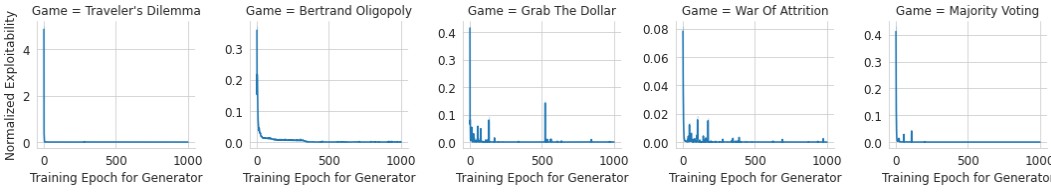

(a) GAES with discriminator that outputs the true best-response ($\arg\max$ of the expected payoffs for each player).

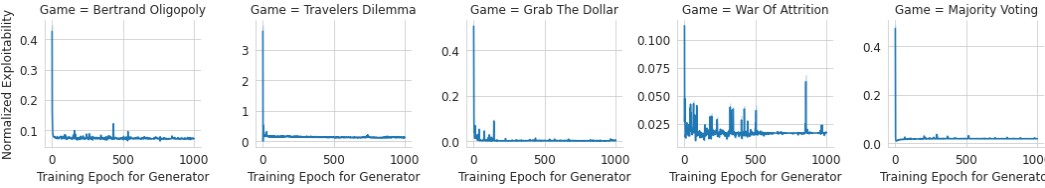

(b) GAES with discriminator that outputs the softmax of the expected payoffs for each player (scaling parameter 100).

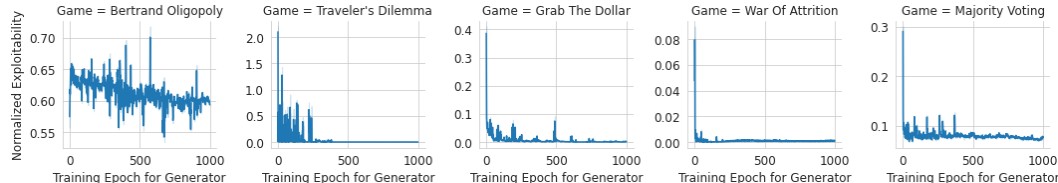

(c) GAES with discriminator that outputs the softmax of the expected payoffs for each player (scaling parameter 10).

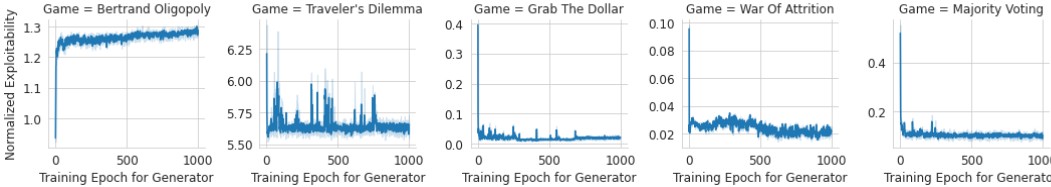

(d) GAES with a neural network discriminator.

Figure 5: Training Exploitability of GAES on five classes of GAMUT games. We observe that the performance of GAES degrades significantly in Bertrand oligopoly and traveller's dilemma when the discriminator is not precise enough.

### G.2 ARROW-DEBREU EXCHANGE ECONOMIES

**Additional Preliminaries.** A *Competitive equilibrium (CE)* is a tuple which consists of allocations $\boldsymbol{X} \in \mathbb{R}_+^{m \times n}$, and prices $\boldsymbol{p} \in \mathbb{R}_+^m$ such that 1. all traders $i \in [n]$ maximize utility constrained by their budget: $\boldsymbol{x}_i^* \in \arg\max_{\boldsymbol{x}_i \in \mathcal{X}_i : \boldsymbol{x}_i \cdot \boldsymbol{p} \leq \boldsymbol{e}_i \cdot \boldsymbol{p}} u_i(\boldsymbol{x}_i)$, 2. the markets clear and goods that are not demanded are priced at 0, i.e., $\sum_{i=1}^m \boldsymbol{x}_i^* \leq \sum_{i=1}^m \boldsymbol{e}_i$ and $\boldsymbol{p}^* \cdot \left(\sum_{i=1}^m \boldsymbol{x}_i^* - \sum_{i=1}^m \boldsymbol{e}_i\right) = 0$.

A *Scarf economy*, denoted $(\boldsymbol{E}, \boldsymbol{V})$, is a Leontief exchange economy with 3 buyers and 3 goods, where the valuation $\boldsymbol{V}$ and endowment $\boldsymbol{E}$ matrices are given as follows:

$$\boldsymbol{E} = \begin{pmatrix} 1 & 0 & 0 \\ 0 & 1 & 0 \\ 0 & 0 & 1 \end{pmatrix} \qquad\qquad \boldsymbol{V} = \begin{pmatrix} 0 & 1 & 0 \\ 0 & 0 & 1 \\ 1 & 0 & 0 \end{pmatrix} \qquad (61)$$

**Related Work.** Exchange economies can be solved in polynomial-time via tâtonnement for CES utilties with $\rho \in [0, 1)$ Bei et al. (2015). However, tâtonnement is not guaranteed to converge beyond these domains. There exists a convex programs to compute CE in linear exchange markets in polynomial time Devanur et al. (2016). The computation of CE is PPAD-complete for Leontief Codenotti et al. (2006), piecewise-linear concave, and additively seperable concave Chen et al. (2009),

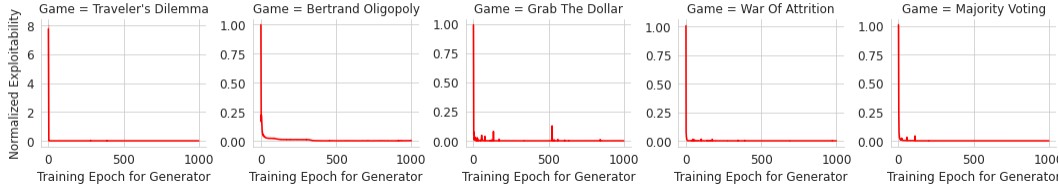

(a) GAES with discriminator that outputs the true best-response ($\arg\max$ of the expected payoffs for each player).

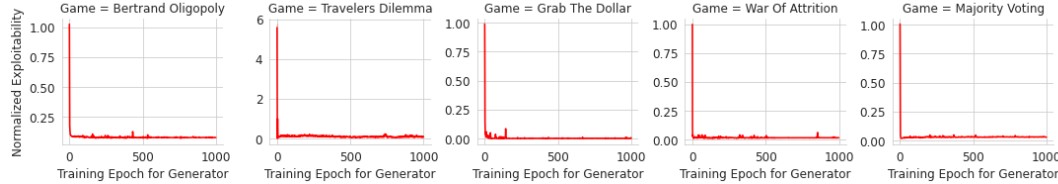

(b) GAES with discriminator that outputs the softmax of the expected payoffs for each player (scaling parameter 100).

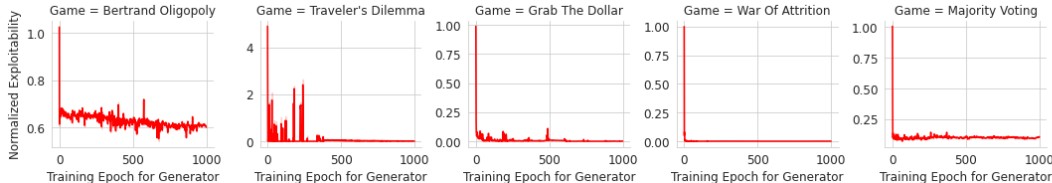

(c) GAES with discriminator that outputs the softmax of the expected payoffs for each player (scaling parameter 10).

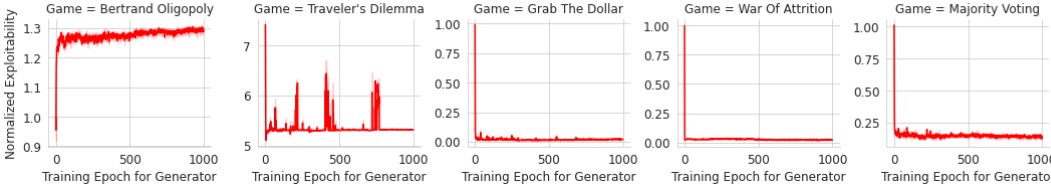

(d) GAES with a neural network discriminator.

Figure 6: Testing Exploitability of GAES on five classes of GAMUT games. We observe that the performance of GAES degrades significantly in Bertrand oligopoly and traveller's dilemma when the discriminator is not precise enough.

and exchange markets Vazirani & Yannakakis (2011); Chen & Teng (2009). The complexity of CES markets for $\rho \in (\infty, 0)$ is unknown and remains an open question.

**Experimental Setup.** We consider experiments on six different exchange economies, each with 3 buyers and 5 goods, and with each economy defined by the utility functions of the players: 1) linear, 2) Cobb-Douglas, 3) Leontief, 4) gross substitutes CES where for all buyers $i \in [n]$, $\rho_i \in [0.5, 1]$, 5) gross complements CES where for all buyers $i \in [n]$, $\rho_i \in [-1.25, -0.75]$, and 6) mixed CES markets where for all buyers $i \in [n]$, $\rho_i \in [-1.25, -0.75] \cap [0.5, 1]$.[7] In addition to these settings, we also consider a Leontief economy with 3 buyers and 3 goods with the goal of solving Scarf economies.

**Baselines.** We benchmark our algorithm to the most well-known algorithm to solve Arrow-Debreu markets, *tâtonnement*, which is an auction-like algorithm that is guaranteed to converge for $\rho \in (1, 0]$ Bei et al. (2015), and thus including the Cobb-Douglas cases, and the *exploitability descent*

---

[7]We reduce the range of the $\rho$ parameter to avoid numerical instability in the computation of utilities.

algorithm[8]. Tâtonnement is defined by the following sequence of prices:

$$\boldsymbol{p}^{(t+1)} = \boldsymbol{p}^{(t)} + \eta\sqrt{t}\left(\sum_{i\in[n]}\boldsymbol{x}_i^{(t)} - \sum_{i\in[n]}\boldsymbol{e}_i^{(t)}\right) \qquad t = 0, 1, \dots \qquad (62)$$

$$\boldsymbol{x}_i^{(t)} \in \argmax_{\boldsymbol{x}_i:\boldsymbol{x}_i\cdot\boldsymbol{p}^{(t)}\leq\boldsymbol{e}_i\cdot\boldsymbol{p}^{(t)}} u_i(\boldsymbol{x}_i) \qquad i \in [n], t = 0, 1, \dots, \qquad (63)$$

while exploitability descent is defined by the following sequence of prices:

$$(\boldsymbol{p}^{(t+1)}, \boldsymbol{X}^{(t+1)}) = (\boldsymbol{p}^{(t)}, \boldsymbol{X}^{(t)}) - \eta\sqrt{t}\left(\nabla\varphi(\boldsymbol{p}^{(t)}, \boldsymbol{X}^{(t)})\right) \qquad t = 0, 1, \dots \qquad (64)$$

where $\varphi(\boldsymbol{p}^{(t)}, \boldsymbol{X}^{(t)})$ is the exploitability associated with the exchange economy pseudo-game. For both of these baselines, we use a decreasing learning rate of $\eta\sqrt{t}$ as a function of the number of iterations $t$, and run the baselines until convergences is observed (we observe that experiments all converge in less than $\leq 200$ iterations). We note that the use of a learning rate schedule is necessary for these algorithms to converge Goktas & Greenwald (2022). We run an extensive grid search for $\eta$ over the set $\{1.0, 0.5, 0.1, 0.05, 0.01, 0.005, 0.001, 0.0005, 0.0001\}$, selecting $\eta$ to minimize exploitability over the validation set. We then evaluate these baselines on the test set with the selected hyperparameters.

**Economy Generation.** For all experiments, we generate 5,000 exchange economy instances, and set aside 500 markets for each of validation and test. To generate a market instance, for all buyers $i \in [n]$ and goods $j \in [m]$, we sample each endowment $e_{ij} \sim \text{Unif}[10^{-9}, 1]$, valuations $v_{ij} \sim \text{Unif}[10^{-9}, 1]$, and when appropriate we sample the substitution parameters $\boldsymbol{\rho}$ from the ranges mentioned above. A competitive equilibrium is guaranteed to exist for exchange markets in all the exchange markets we sample by Arrow-Debreu's first existence theorem Arrow & Debreu (1954) since buyers are endowed with a non-zero amount of each good.

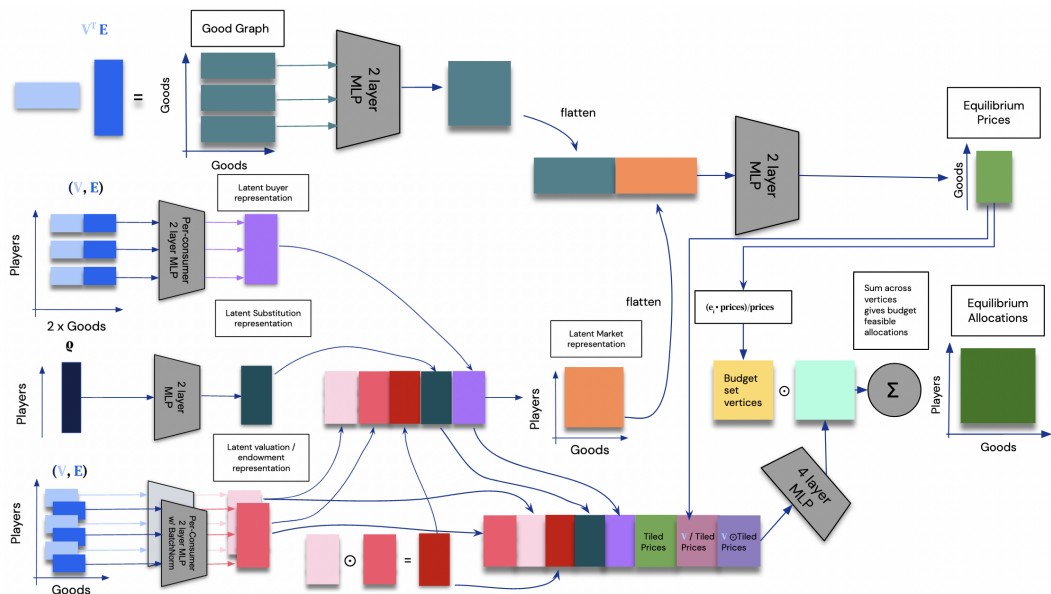

Figure 7: Architecture of the generator of GAES for exchange economies.

**Generator Architecture.** We summarize our generator's architecture in Figure 7. As a reminder, in this setting, the generator takes as input an exchange economy $(\boldsymbol{E}, \boldsymbol{V}, \boldsymbol{\rho})$[9] and outputs equilibrium prices $\hat{\boldsymbol{p}}$ and allocations $\hat{\boldsymbol{X}}$. We use the same generator architecture in each of our experiments.

---

[8]This algorithm simply corresponds to running gradient descent on the exploitability. More information on this algorithm can be found in Goktas & Greenwald Goktas & Greenwald (2022).

[9]If the economy is not a CES economy $\boldsymbol{\rho}$ is drawn uniformly at random from $[0.25, 0.75]^n$.

This generator takes as input the economy matrix $(\boldsymbol{E}, \boldsymbol{V})$, and passes it through two fully connected layers with ReLU activations and 20 and 10 nodes respectively, to obtain a *latent buyer representation*. The valuations $\boldsymbol{V}$, and endowments $\boldsymbol{E}$ are also separately fed through a network with the same architecture. The network for valuations are augmented with a BatchNorm layer with trainable parameters. This gives us a *latent valuation representation* and a *latent endowment representation*. Each latent representation as well as the element-wise product of the latent valuations and latent endowments are then concatenated and fed through two fully connected layers with 20 and $m$ (number of goods) hidden nodes, respectively, followed by ReLU activations at each layer. This gives us a *latent market representation*. We then pass the matrix $\boldsymbol{V}^T \boldsymbol{E}$, which we call the *good graph*, through two fully connected layers, each with BatchNorm with trainable parameters and ReLU activation. These layers have 20 and 10 nodes respectively. We refer to the output of this network as the *latent good graph*. We then concatenate the flattened, latent good graph and latent market representations and feed them through three fully connected layers with 40, 20, and $m$ (number of goods) nodes, respectively. The outputs of the first two layers are passed through ReLU activations, while the last layer is passed through a softmax. The output of this final layer is the generator's predicted prices $\hat{\boldsymbol{p}}$.

Given these prices, we then build an *allocation coefficient matrix* of dimensions $n \times m$, where the $(i,j)th$ entry is given by $\frac{e_i \cdot \hat{\boldsymbol{p}}}{\hat{p}_j}$. We then calculate the budgets $\boldsymbol{E}\hat{\boldsymbol{p}}$ of the buyers at prices $\hat{\boldsymbol{p}}$, and feed them through two fully connected layers, with 30 and 20 nodes respectively and ReLU activations, to obtain a *latent budget representation*. Define the *tiled prices* as $(\hat{\boldsymbol{p}}, \ldots, \hat{\boldsymbol{p}})^T \in \mathbb{R}^{n \times m}$, i.e., a matrix whose rows consists of the vector $\hat{\boldsymbol{p}}$ repeated $n$ times so as to obtain "$n$ tiles of prices". We concatenate the latent representation of buyers, the tiled prices, the latent endowment representation, the valuations element-wise divided by the tiled prices, the valuations element-wise multiplied by the tiled prices, and the latent budget on the last dimension, i.e., we append each matrix horizontally so as to preserve the number of rows $n$. We pass the obtained matrix through 3 fully connected layers, with 100, 50, and 20 nodes respectively and ReLU activations. Finally, the output of this network is passed through a fully connected layer with $m$ nodes and a softmax activation, leaving us with a matrix of dimension $n \times m$ whose rows sum up to 1. We multiply this matrix element-wise with the allocation coefficient matrix, which gives us the allocation $\hat{\boldsymbol{X}}$ for the generator. Notice that $\hat{\boldsymbol{X}}$ is budget feasible at price $\hat{\boldsymbol{p}}$. The total number of parameters of this generator is 20,855.

**Discriminator Architectures.** We summarize the architecture of our network in Figure 8. As a reminder, in this setting, the generator takes as input an exchange economy $(\boldsymbol{E}, \boldsymbol{V}, \boldsymbol{\rho})$ as well as an equilibrium $(\hat{\boldsymbol{p}}, \hat{\boldsymbol{X}})$ and outputs best-responses $(\boldsymbol{p}^*, \boldsymbol{X}^*)$. We build different, modular discriminator architectures for each of the linear, Cobb-Douglas, Leontief, and CES markets. These networks take as input the market matrix $(\boldsymbol{E}, \boldsymbol{V}, \boldsymbol{\rho})$ and the equilibrium $(\hat{\boldsymbol{X}}, \hat{\boldsymbol{p}})$ predicted by the generator. For all four discriminators, the discriminator outputs price $\boldsymbol{p}^*$ such that $p_j^* = 1$ if $j \in \arg\max_{j \in [m]} \sum_{i \in [n]} (\hat{x}_{ij} - e_{ij})$ and $p_j^* = 0$ otherwise. In regard to the allocations, we build a modular allocation network, which takes as input a latent representation of each consumer as a matrix of dimension $n \times p$, and outputs an allocation. It is in this latent representation that the discriminators for each economy differ. We describe the latent representation associated with different exchange economies below. We first build an *allocation coefficient matrix*, where the $(i,j)th$ entry is given by $\frac{e_i \cdot \hat{\boldsymbol{p}}}{\hat{p}_j}$. We then pass the matrix of latent consumer representations through three fully connected layers with 100, 50, 20 nodes respectively and ReLU activations. We take this output and pass it through a final fully connected layer with $m$ (num goods) nodes and softmax activations to obtain a matrix of dimension $n \times m$ whose rows sum up to 1. We multiply this matrix element-wise with the allocation coefficient matrix, which gives us *allocations* $\boldsymbol{X}^*$ that are budget feasible at prices $\hat{\boldsymbol{p}}$.

With the main architecture out of the way, we can now present the different latent consumer representations that we use. For the linear and Cobb-Douglas markets, the latent representation is simply the matrix that is given by the valuations matrix $\boldsymbol{V}$ whose rows are divided by $\hat{\boldsymbol{p}}$. For Leontief, the latent representation is the matrix whose $(i,j)th$ entries are given as $\frac{v_{ij}}{\sum_{j \in [m]} p_j v_{ij}}$. Finally, for CES, the latent representation is given as follows: First we obtain latent representations of each of $\boldsymbol{\rho}$, $\hat{\boldsymbol{p}}$, $(e_i \bigodot p)_{i \in [n]}^T$, and $\boldsymbol{V}$ by passing them through separate but identical, two fully connected BatchNorm layers with ReLU activations, and 20 and 10 nodes respectively. The concatenation of

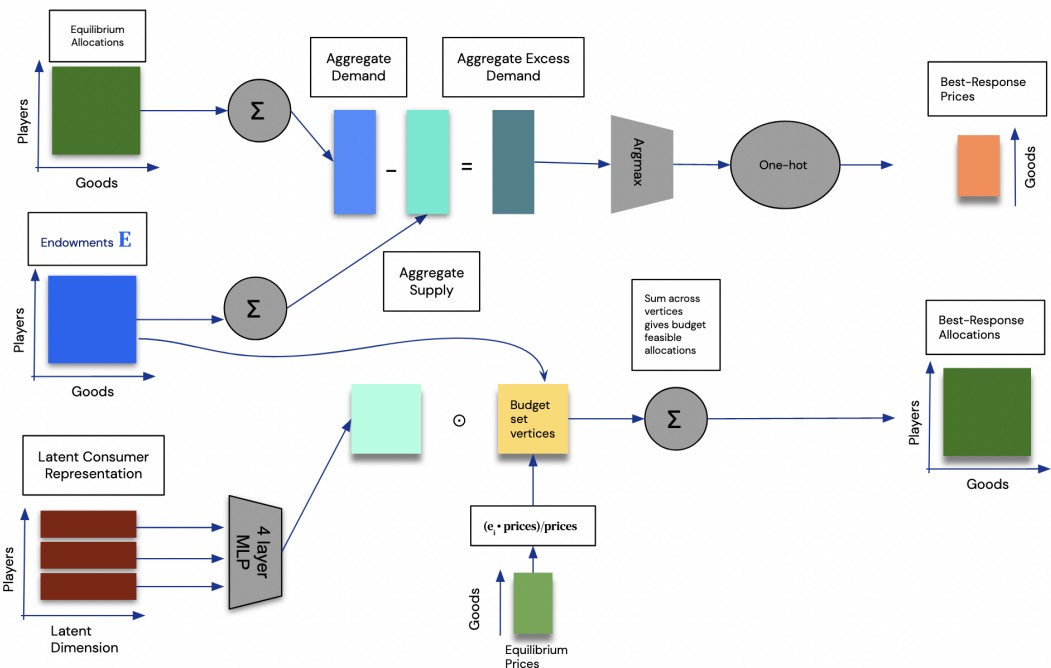

Figure 8: Architecture of the generator of GAES for exchange economies. The latent consumer representation associated with each exchange economy is described in the description of the discriminator.

all of these latent representations on the last dimension so as to obtain a matrix with $n$ rows and 40 columns, where each row is the concatenation of the latent representations for that consumer gives us the *latent consumer representation*.

The number of parameters for the discriminator for each of the linear and Cobb-Douglas markets is $6,775$, for Leontief is $7,115$, and for CES is $14,125$.

**Training Hyperparameters.** We run our algorithm with an initial warm-up of 10,000 iterations for the discriminator. This warm-up procedure follows exactly the inner loop of Algorithm 1 but instead uses randomly sampled economies, and randomly sampled action profiles. After the warmup, we use only one step of inner loop iteration for running Algorithm 1 and run the outer loop for 10,000 iterations. Together with the warmup, the small number of inner loop iterations allow us to significantly speed up the training process. The gradient step is provided by the ADAM algorithm. For the discriminator, we use the same learning rate for the warm-up and regular training. The learning rates used for ADAM in different markets can be found in Table 1. For all other hyperparameters, we use the default settings of ADAM as given in the OPTAX implementation. We process the training data in batches of size 200. We found the learning rates for our algorithm by performing grid search on the validation set for all economies. For the Scarf economy the grid search values were sampled from a standard lognormal distribution.

We present additional results, missing from the main body of the paper, in Figure 2 and Figure 11. We see that GAES outperforms all baselines in all economies except Gross Substitutes CES economies for which tâtonnement performs best. This makes sense since for Gross Substitutes CES economies tâtonnement is guaranteed to converge to a competitive equilibrium. Even the, we note that the performance of GAES is remarkable since it achieves a testing normalized exploitability of 0.005, meaning that GAES finds an action profile which is closer than 99.95% of allocations and prices to a competitive equilibrium.

| Economy Type | Generator | Discriminator |
|---|---|---|
| Linear | 0.0001 | 0.001 |
| Cobb-Douglas | 0.0001 | 0.00001 |
| Leontief | 0.00001 | 0.01 |
| GS CES | 0.00001 | 0.0001 |
| GC CES | 0.0001 | 0.0001 |
| Mixed CES | 0.0001 | 0.00001 |
| Scarf | 0.000003297599624930109 | 0.0000014820835507051155 |

Table 1: Learning rates used for ADAM to train GAES in different markets. These learning rates are found via grid search on the validation set for all economies.

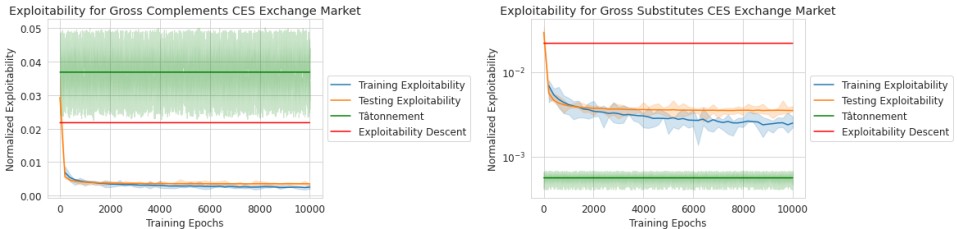

Figure 9: Training and Testing Exploitability of GAES in GS CES and, GC CES, CES economies.

## G.3  KYOTO JOINT IMPLEMENTATION MECHANISM

**Experimental Setup**   For this experiment, we focus on computing a refinement of the GNE known as VE (see Appendix E), which are guaranteed to exist for this jointly convex pseudo-games.[10] This does not change the structure of the generator of GAES or the training algorithm. However, it allows us to consider discriminators that output best-responses that are in the space of jointly feasible actions rather than in the space of individually feasible action spaces, greatly simplifying the architecture of our discriminator. We aim to first replicate the results provided in section 4 of Breton et al. Breton et al. (2006). To do so, we first consider a 2 country Kyoto JI mechanism, with all parameters of the Kyoto JI mechanism except $\theta$ fixed, and compute equilibria for different values of $\theta$ (Figure 4). We then also consider a 2 country Kyoto JI mechanism where we sample all parameters randomly (Figure 14).

**Pseudo-Game Generation.**   We sample 12,000 pseudo-games, putting aside 1,000 for validation and 1,000 for testing. We sample the payoff and constraint parameters of all the players ($\theta$, $\gamma$, $\eta$, $\beta$), uniformly in the range $[0.5, 50]$ to produce the pseudo-games. For each of these pseudo-games, since the set of jointly feasible actions is a polytope, we also generate the vertices associated with the set of jointly feasible actions. To do so, we use the pycdd library Troffaes (2020), and store a matrix of vertices for each pseudo-game, where the rows correspond to the maximum number of vertices, denoted $\mathrm{MaxNumVertex}$, for any pseudo-game in the training set, and the columns correspond to the dimension of the action space, i.e., $n * (n + 1)$ (the first row corresponds to emissions, the last $n$ rows correspond to the investment matrix). For the experiments, and replicating the comparative static analysis of Breton et al. (2006), we randomly sample and fix all parameters of the game except $\theta$, and sample $\theta$ from the range $[0.5, 50]^n$ as stated above.

**Hyperparameters.**   We run our algorithm with an initial warm-up of 10,000 iterations for the discriminator. This warm-up procedure follows exactly the inner loop of Algorithm 1 but instead uses randomly sampled pseudo-games and randomly sampled action profiles. After the warmup, we use only one step of inner loop iteration for running Algorithm 1 and run the outer loop for 10,000 iterations. Together with the warmup, the small number of inner loop iterations allow us to significantly speed up the training process. The gradient step in our algorithm is a step of the ADAM algorithm. For the discriminator, we use the same learning rate for the warm-up and regular

---

[10]We recall that a *variational equilibrium (VE)* (or *normalized GNE*) of a pseudo-game is a strategy profile $\boldsymbol{a}^* \in \mathcal{X}$ s.t. for all $i \in [n]$ and $\boldsymbol{a} \in \mathcal{X}$, $u_i(\boldsymbol{a}^*) \geq u_i(\boldsymbol{a}_i, \boldsymbol{a}_{-i}^*)$. See Appendix E for more details.

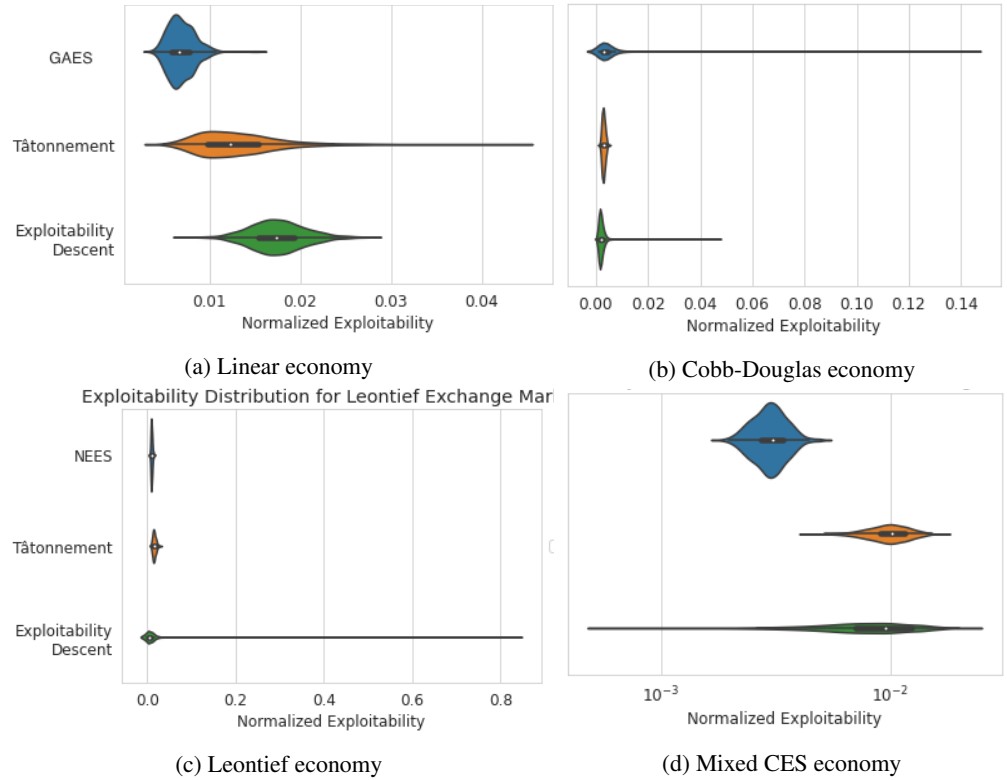

(a) Linear economy            (b) Cobb-Douglas economy

(c) Leontief economy            (d) Mixed CES economy

Figure 10: The distribution of test exploitability on pseudo-games. For Leontief, exploitability descent is not shown, as it performed much worse than either methods. GAES outperforms all baselines on average in all markets and in distribution in all markets except Cobb-Douglas.

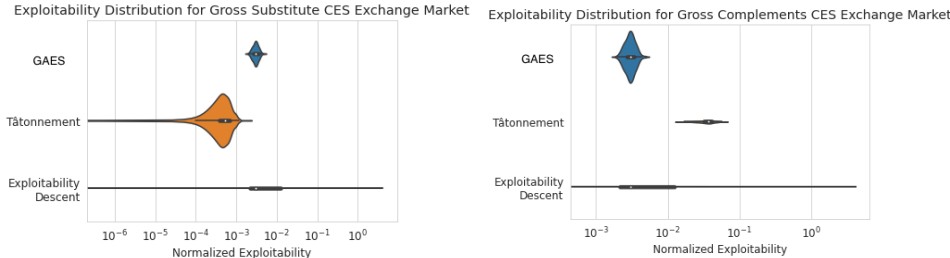

Figure 11: Distribution of test exploitability on exchange economies/pseudo-games. GAES outperforms all baselines on average in all markets, and in distribution in all markets except Cobb-Douglas.

training, following a grid search which is a learning rate of $0.0001$, while for the generator we use a learning rate of $0.001$.

**Generator Architecture.** We summarize the architecture of GAES's generator for Kyoto JI mechanisms in Figure 12. The generator for the Kyoto setting takes as input the game matrix ($\theta$, $\gamma$, $\eta$, $\beta$), and feeds these inputs through a neural network with two hidden layers, each with 20 and 30 nodes respectively and ReLU activations. The output of each layer is also passed through a ReLU activation, as well as a BatchNorm Ioffe & Szegedy (2015) layer with trainable parameters, and with default parameters as implemented by Haiku. The output layer of the neural network consists of a fully connected layer with softmax activation with output dimension equal to $\mathrm{MaxNumVertex}$. The output of this final layer is multiplied with the matrix of vertices associated with the pseudo-game ($\theta$, $\gamma$, $\eta$, $\beta$) across its rows, i.e., each vertex associated with pseudo-game's constraint is multiplied

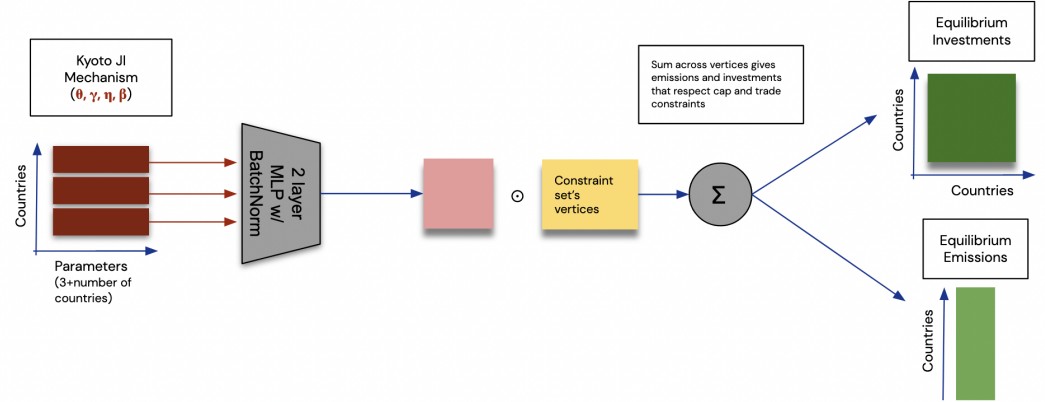

Figure 12: Architecture of the generator of GAES for Kyoto JI mechanisms.

by some probability. The obtained matrix is then summed up across the rows and output by the generator after setting the first column to be $e$, and the matrix formed by the remaining columns to be $X$. Since the neural network outputs a convex combination of the vertices associated with the constraints of the game, the action profile outputted by the neural network is always jointly feasible. The number of parameters for the generator is 2,824.

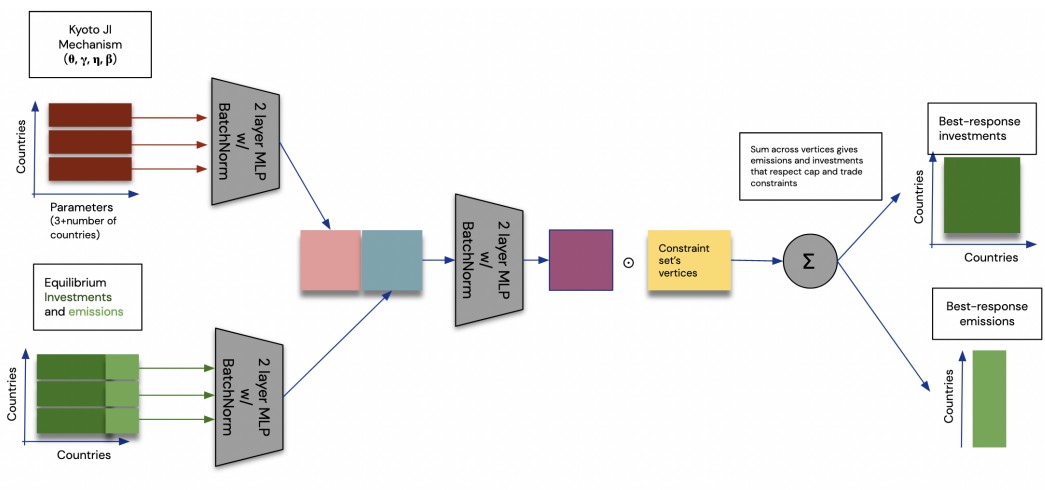

Figure 13: Architecture of the generator of GAES for Kyoto JI mechanisms.

**Discriminator Architecture** We summarize the architecture of GAES's discriminator for Kyoto JI mechanisms in Figure 13. Our discriminator takes as input the matrix $(e, X)$, the output of the generator, and the pseudo-game matrix $(\theta, \gamma, \eta, \beta)$. The equilibrium $(e, X)$ is first passed through a neural network with two fully connected trainable BatchNorm layers, each with 500 nodes. Similarly, the pseudo-game $(\theta, \gamma, \eta, \beta)$ is passed through a network with the same architecture. The output of both networks are then concatenated over the last dimension, i.e., the matrices outputted by both networks are appended horizontally so as to preserve the number of rows $n$. and passed through a neural network with two fully connected trainable BatchNorm layers, each with 500 nodes. For each of these layers, the output is passed through a ReLU activation. The output of the last neural network is then flattened and passed through a final fully connected layer with a softmax activation. The output of this final layer is multiplied with the matrix $(\theta, \gamma, \eta, \beta)$ of vertices associated with the pseudo-game across its rows. The obtained matrix is then summed up across the rows and output by the discriminator after setting the first column to be $e$, and with the matrix formed by the remaining columns of the output adopted as $X$. Since the neural network outputs a convex combination of the vertices associated with the constraints of the game, the action profile is always jointly feasible,

meaning that the neural network outputs a best-response (for a VE). The number of parameters for the discriminator is 1,302,544.

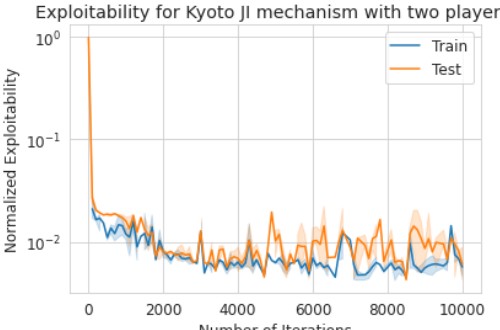

Figure 14: Normalized exploitability achieved by GAES throughout training for a two country JI mechanism.