# OpenReview forum: "Generative Adversarial Equilibrium Solvers"
_ICLR.cc/2024/Conference — ICLR 2024 poster_

### Official Review · Reviewer_pH5J · 2023-10-24

**Soundness:** 4 excellent
**Presentation:** 2 fair
**Contribution:** 3 good
**Rating:** 6
**Confidence:** 4

**Summary:**

In this paper, the authors present a novel neural network-based method for approximating the generalized Nash equilibrium (GNE) within pseudo-games derived from a specific distribution. These pseudo-games consist of players operating within compact and convex action spaces, where the choices made by each player can influence the feasible action space of others.

To facilitate the training of the GNE neural network solver, the authors introduce exploitability as the loss function.  Exploitability quantifies the total utility gains that all players would achieve by deviating to their own best responses. However, calculating exploitability poses a challenge due to the potentially infinite action space. To tackle this issue, an adversarial network is employed to approximate each player's best response.

The authors also establish a theoretical framework by providing a generalization bound for this neural solver. Furthermore, in practical experiments, they apply this approach to identify Nash equilibria in normal-form games, compute competitive equilibria in Arrow-Debreu economic models, and determine GNE in an environmental economic model involving the Kyoto mechanism.

**Strengths:**

- The versatility of this method is evident because it can be applied to a wide range of games, thanks to the inherent generality of pseudo-games.
- The methodology looks strong. The use of two neural networks (a generator and a discriminator) and adversarial training is intriguing to me.
- The concept of employing a neural network as a function approximator to compute GNE is innovative. I believe it has the potential to expedite equilibrium computation in practice.

**Weaknesses:**

- The title in the PDF is still the template title.
- The paper concentrates on finding a single equilibrium, yet many games have multiple equilibria. Incorporating a discussion on equilibrium selection would enhance the work.
- The overall presentation of this paper would benefit from further refinement. The figures within the paper are small and appear blurry due to the absence of vector graphics formats such as .pdf or .svg. Upgrading the figures to vector graphics would improve their clarity and overall visual impact.

**Questions:**

- How does the performance of GAES degrade with approximate discriminators in practice? Is there a way to quantify the required discriminator accuracy?
- Could you apply GAES to find other solution concepts like correlated equilibria? How would the formulation need to change?

---

> ### Author Response · Authors · 2023-11-16
>
> Thank you for your review!
>
> **Regarding the weaknesses you mention.**
>
> - A clash in macro definitions seems to have caused the title to be missing from the paper, this will be fixed for the camera-ready version.
> - We will make sure to include a discussion on equilibrium selection, but we note that a strength of our method is that it does not require practitioners to have to select an equilibrium selection criteria which might be hard to decide.
> - We will make sure to include graphics which are clearer.
>
> **Regarding your questions.**
>
> - **On the effectiveness in approximating the best response.** As we show in additional experiments in Appendix G.1., the accuracy of the discriminator directly impacts the accuracy of the generator, however these two architectures can be tuned independently. We find it interesting to see that 3 of the 5 games can be solved accurately, even when the discriminator is not optimized. This seems to suggest that the accuracy needed for the discriminator depends on the equilibrium structure induced by the payoff structure of the pseudo-game, a direction which future work can explore.
>
> - **On generalizing GAES for correlated equilibria.** Yes, GAES can be directly extended to compute correlated equilibria, and in fact to any equilibrium notion which can be expressed as a $\Phi$-equilibrium [1] where $\Phi$ is a set of action deviations. To do so, one can replace the cumulative regret with the $\Phi$-cumulative regret, i.e., the sum of unilateral improvements by deviations within a set $\Phi$-action deviations. In this more general setting, the discriminator would then simply output action deviations within $\Phi$.
>
> **References**
>
> [1] Greenwald, Amy, and Amir Jafari. "A general class of no-regret learning algorithms and game-theoretic equilibria." Learning Theory and Kernel Machines: 16th Annual Conference on Learning Theory and 7th Kernel Workshop, COLT/Kernel 2003, Washington, DC, USA, August 24-27, 2003. Proceedings. Berlin, Heidelberg: Springer Berlin Heidelberg, 2003.

---

> > ### Comment · Reviewer_pH5J · 2023-11-20
> >
> > Thank you for your response. I have some further questions.
> >
> > * **About the effectiveness in approximating the best response.**
> >
> > I have observed from Figure 5 and Figure 6 that the performance of GAES degrades significantly in Bertrand Oligopoly and Traveller’s Dilemma when the discriminator is not precise enough. However, you only applied a one-layer neural network with a softmax activation function as the neural discriminator. So, how does a more complicated architecture, such as a multi-layer neural network discriminator, perform in the five normal-form games?
> >
> > Furthermore, could you provide some insight into how sensitive the performance of GAES would be with respect to the precision of the discriminator?
> >
> >
> > * **About your general comments.**
> > > We provide an affirmative answer to the open question posed by Marris et al. [1] and Duan et al. [2] ...
> >
> > I did not see the title of Duan et al. [2] in your general comments.
> > It seems to be an AAMAS paper, however, I still not find it in the references of your paper.
> > The references that seem most similar are Duan et al. [2021a] and Duan et al. [2021b].
> > However, I noticed that these two references are both duplicated and contain inaccuracies, such as differences in author order and conference details compared to what you listed in the general comments.
> > It would be good if you could check the references in your paper.

---

> > > ### Author Response · Authors · 2023-11-20
> > >
> > > Thank you for your questions!
> > >
> > > > I have observed from Figure 5 and Figure 6 that the performance of GAES degrades significantly in Bertrand Oligopoly and Traveller’s Dilemma when the discriminator is not precise enough. However, you only applied a one-layer neural network with a softmax activation function as the neural discriminator. So, how does a more complicated architecture, such as a multi-layer neural network discriminator, perform in the five normal-form games?
> > >
> > > From anecdotal experience, a four layer feedforward neural network discriminator with ReLU activations and a softmax final layer performed well in all games except Bertrand Oligopoly and Traveller’s Dilemma. We believe that for a discriminator to perform well for these games, it has to be endowed with a more special architecture since the best-response in these games is only in discrete action which the softmax final activation hinders from representing correctly.  Future work that seeks to build foundation models to solve all pseudo-games can seek to devise optimal generator and discriminator architectures.
> > >
> > > > Furthermore, could you provide some insight into how sensitive the performance of GAES would be with respect to the precision of the discriminator?
> > >
> > > The sensitivity of the generator’s accuracy to the accuracy of the discriminator depends highly on the (pseudo-)game in question. For discrete action (pseudo-)games where the best-response is always discrete the accuracy of the discriminator (i.e., the discriminator outputting a pure action rather than a mixed action) seems to matter a lot as our experiments show, while for (pseudo-)games which do not have this property the performance of the trained generator seems highly robust to the accuracy of the discriminator.
> > >
> > > > I did not see the title of Duan et al. [2] in your general comments. It seems to be an AAMAS paper, however, I still not find it in the references of your paper. The references that seem most similar are Duan et al. [2021a] and Duan et al. [2021b]. However, I noticed that these two references are both duplicated and contain inaccuracies, such as differences in author order and conference details compared to what you listed in the general comments. It would be good if you could check the references in your paper.
> > >
> > > Thank you for pointing this out, references [1] and [2] seem to have not gotten copied over from our draft rebuttal answer. You can find the references mentioned below. We will update the references in the paper to reflect the updated/published versions of the papers and their author list.
> > >
> > > [1] Marris, Luke, et al. "Turbocharging Solution Concepts: Solving NEs, CEs and CCEs with Neural Equilibrium Solvers." Advances in Neural Information Processing Systems.
> > >
> > > [2] Zhijian Duan, Wenhan Huang, Dinghuai Zhang, Yali Du, Jun Wang, Yaodong Yang, Xiaotie Deng. International Conference on Autonomous Agents and Multi-Agent Systems 2023

---

> > > > ### Comment · Reviewer_pH5J · 2023-11-21
> > > >
> > > > > For discrete action (pseudo-)games where the best-response is always discrete the accuracy of the discriminator (i.e., the discriminator outputting a pure action rather than a mixed action) seems to matter a lot as our experiments show.
> > > >
> > > > I believe that for such discrete action pseudo-games, we still need the discriminator to output mixed actions to better approximate the best response. This is because the discrete best response is discontinuous and could change differently with a slight modification of the payoffs.
> > > >
> > > > >  (pseudo-)games which do not have this property the performance of the trained generator seems highly robust to the accuracy of the discriminator.
> > > >
> > > > It is counterintuitive that the generator is more robust with respect to the discriminator in pseudo-games than in normal-form games. As far as I know, pseudo-games are generalized versions of normal-form games; thus, they are more difficult to solve. Do you mean that when the best response is smooth, the generator would be robust to the discriminator?
> > > >
> > > >
> > > > > [2] Zhijian Duan, Wenhan Huang, Dinghuai Zhang, Yali Du, Jun Wang, Yaodong Yang, Xiaotie Deng. International Conference on Autonomous Agents and Multi-Agent Systems 2023
> > > >
> > > > I still haven't seen the title of this reference, and I also cannot find it in the references of your paper.
> > > > Are you referring to the paper `Duan, Zhijian, et al. "Is Nash Equilibrium Approximator Learnable?." Proceedings of the 2023 International Conference on Autonomous Agents and Multiagent Systems. 2023.` ?

---

> ### Author Response · Authors · 2023-11-21
>
> Thank you so much for your quick answer!
>
> > I believe that for such discrete action pseudo-games, we still need the discriminator to output mixed actions to better approximate the best response. This is because the discrete best response is discontinuous and could change differently with a slight modification of the payoffs.
>
> Prior work seems to corroborate to our point. Namely, Duan et al. [1] show that expected exploitability in normal-form games (i.e., when the discriminator in our setting outputs the exact best-response) can be effectively minimized. This is because while the best-response might be discontinuous (assuming it is unique) in the candidate equilibrium, the exploitability itself is still continuous in the candidate equilibrium.
>
> > It is counterintuitive that the generator is more robust with respect to the discriminator in pseudo-games than in normal-form games. As far as I know, pseudo-games are generalized versions of normal-form games; thus, they are more difficult to solve. Do you mean that when the best response is smooth, the generator would be robust to the discriminator?
>
> Our statement was only regarding regarding pseudo-games with **discrete** action spaces which also includes normal-form games. The intuition for our answer is that if a mixed action best-response always exists, then it is more likely that the players' best-responses change smoothly as a function of the other players' actions. Once again, these claims are empirical anecdotes and not theoretical claims, since in general, without strong concavity and Lipschitz-smoothness of payoffs it is not possible to guarantee a connection between the best-response of each player and their opponents' actions (e.g., Lipschitz-continuity of the best-responses w.r.t. the opponents' actions). Beyond this, we prefer to refrain ourselves from speculating a theory.
>
>
> > I still haven't seen the title of this reference, and I also cannot find it in the references of your paper. Are you referring to the paper Duan, Zhijian, et al. "Is Nash Equilibrium Approximator Learnable?." Proceedings of the 2023 International Conference on Autonomous Agents and Multiagent Systems. 2023. ?
>
> Thank you for noticing this! You can find the correct ACM citation in [1]. The version of the paper cited in our paper is [2]. We consulted [2] while writing our paper, however, the contents align with the most recent version [1].
>
> [1] Zhijian Duan, Wenhan Huang, Dinghuai Zhang, Yali Du, Jun Wang, Yaodong Yang, and Xiaotie Deng. 2023. Is Nash Equilibrium Approximator Learnable? In Proceedings of the 2023 International Conference on Autonomous Agents and Multiagent Systems (AAMAS '23). International Foundation for Autonomous Agents and Multiagent Systems, Richland, SC, 233–241.
>
> [2] Zhijian Duan, Dinghuai Zhang, Wenhan Huang, Yali Du, Jun Wang, Yaodong Yang, and Xiaotie
> Deng. Towards the pac learnability of nash equilibrium, 2021b. URL https://arxiv.org/abs/2108.07472.

---

### Official Review · Reviewer_Wcqa · 2023-10-29

**Soundness:** 2 fair
**Presentation:** 3 good
**Contribution:** 3 good
**Rating:** 6
**Confidence:** 4

**Summary:**

The paper introduces generative adversarial equilibrium solvers (GAES), a GAN that learns to map pseudo-games to their generalized Nash equilibria from a sample of problem instances. In particular, they provide a formulation that makes the problem amenable to standard stochastic first-order methods. They use GAES to compute in a scalable way competitive equilibria in exchange economies and an environmental economic model of the Kyoto mechanism, outperforming earlier methods.

**Strengths:**

Pseudo-games are very general game-theoretic models with a number of applications, most notably Arrow-Debreu competitive economies. Yet, there is a lack of scalable techniques for computing generalized Nash equilibria in such settings. This paper makes a concrete contribution in that direction by providing a method with promising performance across a number of benchmarks. The proposed method is natural and the experimental results are overall convincing and quite thorough. Indeed, the paper appears to attain state of the art performance in a number of important applications, and could have significant impact in this area.

**Weaknesses:**

There are some soundness issues that the authors have to address. First, there appears to be a significant gap between the theoretical analysis and the experimental settings. Specifically, it is not clear how a stationary point in the sense of Theorem 4.1 translates to a GNE. If stationary points are not necessarily GNE, the narrative of the paper has to be restructured. In particular, it is often claimed that the method maps pseudo-games to GNE, and it is not clear whether that claim is theoretically sound. Of course, computing GNE is intractable, but it is alluded (for example in the abstract) that under a distribution over problem instances the problem could be easier. Theorem 4.1 also makes a strong concavity assumption which appears to be violated in all settings of interest. It should be the case that a "small" regularizer can always be incorporated without affecting the equilibria by much, but I think this should be discussed in more detail.

I am also confused about Footnote 4. It is claimed that the method obtains the state of the art $O(1/\epsilon^3)$ complexity, a major improvement over $O(1/\epsilon^6)$, which the authors claim was the previous state of the art. The authors have to explain more precisely the class of problems this applies to; there are many variants of the PL condition studied in the literature. In particular, the following papers seem to obtain a much better dependency: "Faster single-loop algorithms for minimax optimization without strong concavity," "faster stochastic algoritms for minimax optimization under Polyak-Lojasiewicz Conditions" and "Doubly smoothed GDA for constrained nonconvex-nonconcave minimax optimization."

Besides the issues above, the algorithmic approach is very close to the paper "Exploitability minimization in games and beyond," which limits to some extent the algorithmic contribution of the present paper. The authors have to highlight the comparisons in more detail.

**Questions:**

Some minor comments for the authors:

1. The title of the submission document is the default one.
2. The references have to be polished. There are many papers that are published many years ago and only the arXiv version is cited. There is also an issue with consistency: sometimes URLs are included, sometimes they are not. Please fix those issues.
3. There are underfull equations in Observation 1 and immediately below.
4. I don't understand how the paper of Daskalakis et al. (2009) is relevant in the context of Footnote 4 about min-max optimization.
5. The appendix has many overfull equations that need to be formatted properly.

---

> ### Author Response · Authors · 2023-11-16
>
> Thank you for your review!
>
> **Regarding the weaknesses you mention.**
>
> **On stationary points of the exploitability.** Stationary points of exploitability do not always coincide with GNE, however, our convergence result to stationary points of the expected exploitability is still in line with known theoretical results, since the computation of GNE under Assumption 1 is PPAD-complete [1] and converging to a GNE would imply polynomial-time computation for problems in PPAD. That said, stationary points of the exploitability correspond to $\varepsilon$-GNE, where $\varepsilon$ is equal to the exploitability at the stationary point, and as at a stationary point the exploitability cannot be decreased via first-order information, this Theorem 4.1 suggests that our approach is sound to learn equilibrium mappings via first-order methods, the most prevalent training methods in deep learning.
>
>
> **On the strong concavity assumption.** Our theorems directly generalize to payoff functions that only satisfy concavity. This extension follows from the discussion at lines 235-239 and is obtained by regularizing the exploitability. Note that this regularization does not modify the optimal solutions of the optimization problem. However, while running our experiments for exchange economies we have tried adding this regularization to the cumulative regret, but we observed that this regularized cumulative regret performed worse than the original cumulative regret most likely due to the fact that although regularization makes the cumulative regret strongly concave, it also increases the non-convexity of the exploitability. As such, we have decided to present the experiments without regularization. We will make sure to include a more extensive discussion to our paper on this issue.
>
> **On the differences between our paper and Goktas and Greenwald’s “Exploitability Minimization in Games and Beyond”.**
> While Goktas and Greenwald observe that the exploitability minimization problem for *only one* pseudo-game can be expressed as a min-max optimization, it is not obviously true that one can minimize the expected exploitability via a min-max optimization problem. The novelty in our method lies in noticing that we can compute the expected exploitability after computing the expected cumulative regret, by optimizing over the space of best-response functions from pseudo-games to actions, rather than the space of actions individually for every pseudo-game.
>
> **On convergence complexity.** Thank you for pointing out these papers to us, especially "Doubly smoothed GDA for constrained nonconvex-nonconcave minimax optimization." which seems to going to appear at NeurIPS seems very relevant and we were not aware of them. At a high-level, in all three papers, ignoring acceleration-type methods–-which are rarely used in the training of neural networks in practice due to their complexity—the best convergence complexity achieved is $O(\varepsilon^{-4})$ (see for instance [2]), while our algorithm—which makes use of no acceleration techniques—achieves a complexity of $O(\varepsilon^{-3})$. As a result, in the light of these more recent (and some unpublished) results, our algorithm still achieves a faster convergence rate than single-loop algorithms with no-acceleration.
>
>
> **Regarding your questions.**
>
> 1) A clash in macro definitions seems to have caused the title to be missing from the paper, this will be fixed for the camera-ready version.
> 2) We chose to cite the Arxiv versions of the paper to allude to the most recent version of the papers. That said, to address your concern we will cite the published versions of the papers when the Arxiv versions we have consulted match the published versions.
> 3) We will fix these equation formatting issues for the camera-ready.
> 4) Thank you for pointing this citation out. It seems that we accidentally cited the wrong paper from Daskalakis et al., while we sought to cite [2].
> 5) Once again, we will fix these equation formatting issues for the camera-ready.
>
> **References**
>
> [1] Yang, Junchi, et al. "Faster single-loop algorithms for minimax optimization without strong concavity." International Conference on Artificial Intelligence and Statistics. PMLR, 2022.
>
> [2] Daskalakis, Constantinos, Dylan J. Foster, and Noah Golowich. "Independent policy gradient methods for competitive reinforcement learning." Advances in neural information processing systems 33 (2020): 5527-5540.

---

> > ### Comment · Reviewer_Wcqa · 2023-11-19
> > **Thank you for the Response**
> >
> > I thank the authors for their response. Regarding the first point, computing GNE is of course intractable in general, but your abstract gives the misleading idea that your setting is more benign; namely, you are saying that "Although the computation of GNE and CE is intractable in the worst-case, i.e., PPAD-hard, in practice, many applications only require solutions with high accuracy in expectation over a distribution of problem instances." That is, in the setting where one is given a distribution over games, the PPAD-hardness results do not necessarily kick in. Now you are claiming in the abstract "We introduce Generative Adversarial Equilibrium Solvers (GAES): a family of generative adversarial neural networks that can learn GNE and CE from only a sample of problem instances." If I understand correctly this is wrong: you only compute stationary points. If the focus of the paper is on computing GNE, then the method is not theoretically sound. At the very least the authors have to revise the presentation of the paper to account for this discrepancy. On a similar note, I am not sure whether it is fair to compare your algorithm with other methods that are theoretically guaranteed to converge to competitive equilibria.
> >
> > Regarding the iteration complexity, there is also a paper appearing at this year's NeurIPS, namely "A Single-Loop Accelerated Extra-Gradient Difference Algorithm with Improved Complexity Bounds for Constrained Minimax Optimization," which seems to subsume the complexity bounds of this paper. This is of course concurrent work, so it does not have any bearing on the evaluation of this paper, but it would be good to mention that in the revised version.

---

> > > ### Author Response · Authors · 2023-11-20
> > >
> > > Thank you for the clarifications!
> > >
> > > > Regarding the first point, computing GNE is of course intractable in general, but your abstract gives the misleading idea that your setting is more benign; namely, you are saying that "Although the computation of GNE and CE is intractable in the worst-case, i.e., PPAD-hard, in practice, many applications only require solutions with high accuracy in expectation over a distribution of problem instances." That is, in the setting where one is given a distribution over games, the PPAD-hardness results do not necessarily kick in. Now you are claiming in the abstract "We introduce Generative Adversarial Equilibrium Solvers (GAES): a family of generative adversarial neural networks that can learn GNE and CE from only a sample of problem instances." If I understand correctly this is wrong: you only compute stationary points. If the focus of the paper is on computing GNE, then the method is not theoretically sound. At the very least the authors have to revise the presentation of the paper to account for this discrepancy.
> > >
> > > We want to emphasize that the main goal of our method is indeed to compute GNE, and the reason our convergence results are only to a stationary point of the expected exploitability (rather than its minimum) is simply due to our desire to propose broad computational results that apply to nearly all pseudo-games, including those which are PPAD-complete to solve (e.g., the pseudo-games studied in our experiments). If we make stronger assumptions, for instance if in addition to Assumption 1, we assume that the pseudo-game is zero-sum, and the generator is affine in its parameters, then our convergence result implies convergence to a GNE since the expected exploitability then becomes convex. *All this said, we understand your concern and will make the necessary adjustments to our paper to clarify these subtleties.*
> > >
> > > > On a similar note, I am not sure whether it is fair to compare your algorithm with other methods that are theoretically guaranteed to converge to competitive equilibria.
> > >
> > > As one of our motivations was to show that GAES can allow practitioners to solve pseudo-games repeatedly or en masse without having to tune algorithm hyperparameters for every single pseudo-game *individually*, we deemed it necessary to compare the performance of GAES against known baseline algorithms.
> > >
> > > > Regarding the iteration complexity, there is also a paper appearing at this year's NeurIPS, namely "A Single-Loop Accelerated Extra-Gradient Difference Algorithm with Improved Complexity Bounds for Constrained Minimax Optimization," which seems to subsume the complexity bounds of this paper. This is of course concurrent work, so it does not have any bearing on the evaluation of this paper, but it would be good to mention that in the revised version.
> > >
> > > Thank you for providing us with additional references which we will add to our related works section. We could not find a version of the paper online so we will refrain ourselves from speculating on the differences between our result and the authors’ results. That said, we would like to note that our convergence result is not only a complexity result on min-max optimization but also a result on the complexity of training generative neural networks (e.g., we derive the necessary assumptions on the neural networks to obtain our convergence results). Additionally, our convergence result, takes advantage of the special structure of the discriminator (namely, Lipschitz-smoothness of discriminator in its inputs, which guarantees that the expected exploitability is also Lipschitz-smooth in the parameters of the generator) to obtain our convergence rate for our algorithms, and as such cannot be directly obtained from results on min-max optimization without ignoring this special structure.

---

> ### Comment · Reviewer_Wcqa · 2023-11-21
> **Thank you for the Response**
>
> I thank the authors for their time and their response. I have no further questions at the moment.

---

### Official Review · Reviewer_Db1F · 2023-11-01

**Soundness:** 2 fair
**Presentation:** 2 fair
**Contribution:** 3 good
**Rating:** 5
**Confidence:** 4

**Summary:**

This paper study the generalized Nash equilibrium of pseudo games where a player’s action not only affects his utility, but also other players’ action sets. The authors use GAN and employ exploitability as the loss function. The solver is applied to compute the GNE of Arrow-Debreu competitive economies and the Kyoto mechanism.

**Strengths:**

1. Introduce a novel method to compute GNE by GAN.
2. Provide theoretical guarantee on convergence and generalization bounds.
3. The performance is better according to the experiments.
4. The literature review in the appendix summarizes the current methods to solve GNE and the application of pseudo games.

**Weaknesses:**

1. Assume strong concavity in assumption 1, however, the utility function is not strong convex in Arrow-Debreu competitive economy.
2. Do not provide guarantee for the performance on the training set.
3. Use different network architecture in two experiments which means GAES is not a general solver for GNE.

**Questions:**

1. In which paper was the name "pseudo game" and "GNE" made? It seems that the cited paper by Arrow Debreu mentioned the game first, but did not name it.
2. Do you measure the difference between the results and the optimal action in the feasible set? Notice that the results “is on average better than at least 99% of the action profiles” in the experiments.
3. Is there any guarantee when the utility function only satisfy convexity?
4. What is the title of this paper?

---

> ### Author Response · Authors · 2023-11-16
>
> Thank you for your review!
>
> **Regarding the weaknesses you mention.**
> 1) Our theorems directly generalize to the concave game settings and as such apply to Arrow-Debreu markets as well; see part 3) of the below answer to your questions and the discussion in lines 235-240 for more information.
> 2) Our results provide computational complexity guarantees on the training dataset (Theorem 4.1), whose average exploitability achieved on the training set generalizes to the test set with a sample complexity result (Theorem 4.2). Even though Theorem 4.1 does not provide a bound on the value of the average exploitability of the training set, it tells us that average exploitability on the training set, cannot be minimized via first-order deviations, hence suggesting the correctness of the algorithm we use. We note that lack of a full characterization of the average exploitability on the training set is a common occurrence in the literature for learning equilibrium mappings (see for instance [4] and [5]).
> 3) Just like any neural network, GAES also faces an architecture search problem, and for it to solve the task at hand a search over optimal architectures has to be done. However, this does not affect the generality of GAES. To give a theoretical explanation, according to our sample complexity bounds, the accuracy of GAES depends on the choice of the generator and discriminator hypothesis classes which appear as a "bias" term corresponding to the difference in performance between the optimal generator and discriminator in the class of all possible functions and the optimal ones in the hypothesis class. That is, the choice of a hypothesis class will affect the bias term in performance. As the selection of a neural network architecture effectively determines the hypothesis space we chose, if the architecture of the network is not well-chosen, then the bias of GAES will go up, hence degrading accuracy. This, however, does not take away the fact that with an appropriate neural network architecture GAES is guaranteed to be ``general’’ as the bias term will go down to 0.
>
>
> **Regarding your questions.**
> 1) Pseudo-games were introduced by Kenneth Arrow and Gerard Debreu in 1954 [1] under the name of Abstract Economies, who studied GNE under the name of “equilibrium points”. More recently, Francisco Facchinei and Christian Kanzow’s works has re-popularized Arrow and Debreu’s model under the name of pseudo-games, calling equilibrium points GNE [2]. To the best of our knowledge the first use of the term GNE is due to Harker [3].
>
> 2) The distance between the candidate equilibrium actions outputted by GAES and any GNE is measured in payoff space in terms of the exploitability.  As such the phrasing “is on average better than at least 99% of the action profiles” means that the GAES achieves an exploitability lower than 99% of all other action profiles in the feasible action space.
>
> 3) Yes, our theoretical guarantees apply more broadly to any concave game—that is we can drop the strong concavity condition. As mentioned in lines 235-240, our theorems generalize to the concave game settings by replacing the exploitability with the regularized exploitability, however for consistency and simplicity we chose to present our results under these stronger assumptions, since in experiments non-regularized exploitability performed better.
>
> 4) The paper is called “Generative Adversarial Equilibrium Solvers”, a clash in macro definitions seems to have caused the title to be missing from the paper.
>
> **References**
>
> [1] Arrow, Kenneth J., and Gerard Debreu. "Existence of an equilibrium for a competitive economy." Econometrica: Journal of the Econometric Society (1954): 265-290.
>
> [2] Facchinei, Francisco, and Christian Kanzow. "Generalized Nash equilibrium problems." Annals of Operations Research 175.1 (2010): 177-211.
>
> [3] Harker, P. T. (1991). Generalized Nash games and quasi-variational inequalities. European Journal of Operational Research, 54, 81–94.
>
> [4] Marris, Luke, et al. "Turbocharging Solution Concepts: Solving NEs, CEs and CCEs with Neural Equilibrium Solvers." Advances in Neural Information Processing Systems.
>
> [5] Zhijian Duan, Wenhan Huang, Dinghuai Zhang, Yali Du, Jun Wang, Yaodong Yang, Xiaotie Deng. International Conference on Autonomous Agents and Multi-Agent Systems 2023

---

> > ### Comment · Reviewer_Db1F · 2023-11-20
> >
> > Thanks for your response. I have several addtional comments.
> >
> > > Our theorems directly generalize to the concave game settings and as such apply to Arrow-Debreu markets as well; see part 3) of the below answer to your questions and the discussion in lines 235-240 for more information.
> >
> > I checked with the cited paper Von Heusinger & Kanzow, 2009. It seems not a trivial result that under the regularized utility function, the GNE is still the same, letting alone the non-uniqueness issue of GNE. It would be better provide a detailed explanation.
> >
> > > Do not provide guarantee for the performance on the training set.
> >
> > Theorem 4.1 is about the convergence, but there is no result about how far is the stationary point to GNE, which is the "performance" I mean.
> >
> > > Just like any neural network, GAES also faces an architecture search problem, and for it to solve the task at hand a search over optimal architectures has to be done.
> >
> > This is not what a "solver" should be like. Anyway, it does not matter much.
> >
> > > The distance between the candidate equilibrium actions outputted by GAES and any GNE is measured in payoff space in terms of the exploitability. As such the phrasing “is on average better than at least 99% of the action profiles” means that the GAES achieves an exploitability lower than 99% of all other action profiles in the feasible action space.
> >
> > "better than at least 99% of the action profiles" is really strange. Literally, the rest 1% action profile may be much better than what the approach learned.

---

> > > ### Author Response · Authors · 2023-11-20
> > >
> > > Thank you for your clarifications!
> > >
> > > > I checked with the cited paper Von Heusinger & Kanzow, 2009. It seems not a trivial result that under the regularized utility function, the GNE is still the same, letting alone the non-uniqueness issue of GNE. It would be better provide a detailed explanation.
> > >
> > > We will add additional clarification to the camera-ready on this point, and add a forward reference to an appendix section where we will expand on the regularized exploitability as introduced by Von Heusinger & Kanzow.
> > >
> > > > Theorem 4.1 is about the convergence, but there is no result about how far is the stationary point to GNE, which is the "performance" I mean.
> > >
> > > We agree with you. That said, the reason our convergence results are only to a stationary point of the expected exploitability (rather than its minimum) is simply due to our desire to propose broad computational results that apply to nearly all pseudo-games, including those which are PPAD-complete to solve (e.g., the pseudo-games studied in our experiments). If we make stronger assumptions, for instance if in addition to Assumption 1, we assume that the pseudo-game is zero-sum, and the generator is affine in its parameters, then our convergence result implies convergence to a minimum of the expected exploitability since the expected exploitability then becomes convex.
> > >
> > > > This is not what a "solver" should be like. Anyway, it does not matter much.
> > >
> > > We understand your concern regarding the language. We are open to alternative terminology suggestions.
> > >
> > > > "better than at least 99% of the action profiles" is really strange. Literally, the rest 1% action profile may be much better than what the approach learned.
> > >
> > > That is perhaps possible, meaning that maybe 1% of the action profiles could qualitatively be much closer (e.g., closer in action space as opposed to in payoff space) to a GNE. However, our experiments on the Kyoto joint implementation mechanism suggests that GAES not only performs well in payoff space but also in action space, replicating closely the qualitative aspects of an exact GNE. This said, we would like to add that most, if not all, of the computational literature on GNE measures the quality of GNE in payoff space (i.e., by looking at $\varepsilon$-GNE), and not in action space (see for instance Jordan et al. [1]).
> > >
> > > **References**
> > >
> > > [1] Jordan, Michael I., Tianyi Lin, and Manolis Zampetakis. "First-order algorithms for nonlinear generalized nash equilibrium problems." Journal of Machine Learning Research 24.38 (2023): 1-46.

---

### Official Review · Reviewer_ni1H · 2023-11-10

**Soundness:** 4 excellent
**Presentation:** 4 excellent
**Contribution:** 4 excellent
**Rating:** 10
**Confidence:** 5

**Summary:**

This work proposes a novel algorithm, General Adversarial Equilibrium Solvers, for training general GNE solvers.

The goal of equilibrium solvers is, given a strategic game between multiple players, to find a (generalized) Nash equilibrium of the game.

While there has been a few previous work that proposes algorithms to train equilibrium solvers, they all suffers from three technical challenges:
- the gradient of the exploitability requires solving a concave maximization problem
- the exploitability of pseudo-games is in general not Lipschitz-continuous
- the gradients cannot be bounded in general

The authors formulates equilibrium solver training as training a generative adversarial networks, where the generator takes a pseudo-game representation, and outputs a tuple of actions (one per player), and the discriminator takes both the pseudo-game, and the output of the generator, and outputs a best-response for each player.

The goal of discriminator is to output a best-response actions that produces the exploitability, and the goal of generator is to output actions that minimizes the exploitability, i.e., GNE.

**Strengths:**

# Presentation
- The paper is well-organized and easy-to-follow: Sec.1 motivates the readers by illustrating the possible applications of GNE solvers, including network communication, cloud computing, and economic models (e.g., Arrow-Debreu exchange economy, Kyoto joint implementation mechanism)

# Novelty, Technical Contribution
- The formulation of GAES establishes a novel, efficient, simple, and scalable algorithm to train generic GNE solvers.
    - To the best of my knowledge, most of the previous work relied on supervised learning, and suffered from a few technical challenges in terms of computational tractability and stability.
    - GAES beautifully solves these problems, and provides a simple yet powerful framework for training GNE solvers.
- The formulation is strongly backed up by theoretical guarantees; convergence of the networks towards a stationary point of exploitability, and sample complexity.
- The experiments are conducted on non-trivial games, namely Arrow-Debreu exchange economies and Kyoto joint implementation mechanism — which are non-monotone or non-jointly convex. Strong empirical results on these games verifies the efficiency of GAES.

**Weaknesses:**

# Weaknesses
- I don’t see any special weakness in this paper. The authors establish a simple yet powerful framework for training GNE solvers, and backs up their algorithm both with strong theoretical guarantees and empirical results.

**Questions:**

- Would it be possible to scale GAES to modern games that consists of multiple neural networks (e.g., GANs, multi-agent RL problems, etc.)? If not, what would be the main technical challenges to do so?

---

> ### Author Response · Authors · 2023-11-16
>
> Thank you so much for your encouraging and thorough review! It means a lot for us!
>
> Regarding your question. GAES extends directly to games $\mathcal{G} \in \Gamma$ in which players are $n \in \mathbb{N}$ “neural players” with each player $i \in \mathbb{N}$ “choosing” an action (i.e., network weights) from an action space (i.e., a subset of $\mathbb{R}^d$ where $d$ is the number of parameters) which maximizes some desired payoff function $u_i^{\mathcal{G}}$. In such settings, GAES can be seen a meta-learning method, where any game $\mathcal{G} \in \Gamma$ is described as parameters of the problem.
>
> For instance, for generative adversarial neural networks for image classification, we have two neural players consisting of a generator and a discriminator for whom the payoff functions are respectively given as the expected negated cross-entropy and expected cross-entropy over a data set of images. As such, each game is characterized by the data set of images, and GAES can be trained over *distributions* of image datasets, so as to output optimal network weights for the generator and discriminator for a given image dataset.

---

### Author Response · Authors · 2023-11-16
**Common Answer**

**Summary of our contributions:** We provide an affirmative answer to the open question posed by Marris et al. [1] and Duan et al. [2] on whether unsupervised learning methods to learn mappings from games to their equilibria can be extended to more general game-theoretic models, while making significant progress on an important problem in computational economics, namely the computation of competitive equilibrium. Indeed, the problem of devising general-purpose algorithms to compute competitive equilibrium for Arrow-Debreu markets is a problem that was first posed by Herbert Scarf in the 1970s [3] and to this day there exists no known efficient algorithms to compute competitive equilibrium in exchange economies in which consumers have Leontief (or more generally gross complements) preferences. GAES, as shown in experiments, can effectively approximate competitive equilibrium in these economies via a general purpose method to approximate generalized Nash equilibrium (GNE) over a distribution of pseudo-games and does so with extensive theoretical analysis (Theorem 4.3 and 4.5).

**Summary of common concern:** Reviewers Db1F and Wcqa were concerned that our theory  applies only to payoff functions which are strongly concave and as such does not apply to our experimental setups

**Answer**: Our theorems directly generalize to payoff functions that only satisfy concavity. This extension follows from the discussion at lines 235-239 and is obtained by regularizing the exploitability. Note that this regularization does not modify the optimal solutions of the optimization problem (see, for instance, Theorem 3.3. of Von Heusinger and Kanzow [4]) . However, while running our experiments for exchange economies we have tried adding this regularization to the cumulative regret, but we observed that this regularized cumulative regret performed worse than the original cumulative regret most likely due to the fact that although regularization makes the cumulative regret strongly concave, it also increases the non-convexity of the exploitability. As such, we have decided to present the experiments without regularization. We will make sure to include a more extensive discussion to our paper on this issue.

**References**

[1] Marris, Luke, et al. "Turbocharging Solution Concepts: Solving NEs, CEs and CCEs with Neural Equilibrium Solvers." Advances in Neural Information Processing Systems.

[2] Zhijian Duan, Wenhan Huang, Dinghuai Zhang, Yali Du, Jun Wang, Yaodong Yang, Xiaotie Deng. International Conference on Autonomous Agents and Multi-Agent Systems 2023

[3] Scarf, H.E., 1967b, “On the computation of equilibrium prices” in Fellner, W.J. (ed.), Ten Economic Studies in the tradition of Irving Fischer, New York, NY: Wiley

[4] Anna Von Heusinger and Christian Kanzow. “Optimization reformulations of the generalized Nash equilibrium problem using Nikaido-Isoda-type functions”. In: Computational Optimization and Applications 43.3 (2009), pp. 353–377.

---

### Meta-Review · Area_Chair_VxbV · 2023-12-06

**Metareview:**

The paper proposes a GAN mapping pseudogames to their Nash equilibria and use it to scalably compute equilibria certain economic applications, beating prior baselines.

Overall, while reviewer ni1H gave a high-confidence score of 10 with little substantiation, they failed to engage in any of the follow up conversation; so, I'm going to somewhat discount their review as an outlier.

Overall, while the results appear a bit incremental given existing paper (for example, Goktas and Greenwald), I think the authors did a good job at responding to the different concerns. I believe, and most reviewers agree, that the paper has merits. I also found the empirical evaluation interesting. For all these reasons, I'm recommending weak acceptance.

**Justification For Why Not Higher Score:**

I think the paper is perhaps slightly too incremental to justify bumping up the score.

**Justification For Why Not Lower Score:**

I think that the paper has merits and deserves acceptance.

---

### Decision · Program_Chairs · 2024-01-16

Accept (poster)